EMBO
Molecular Medicine

# A viral glycoprotein targets IgG⁺ memory B cells to mediate humoral immune evasion

Precious Cramer [1,2,3,15,16], Stefan F H Neys[4,16], Manuela Fiedler [5,6], Raquel Lorenzetti[4], Henrike Reinhard[5], Iga Janowska[4], Julian Staniek[4], Ann-Katrin Kohl[1], Petra Hadlova[7,8], Magdalena Huber [1], Bodo Plachter [9], Clarissa Read [10,11], Valeria Falcone [1], Jens von Einem[9], Katja Hoffmann[1], Tihana Lenac Rovis[12], Stipan Jonjic[12], Philipp Kolb [1], Marta Rizzi [4,7,13,14,17 ✉] & Hartmut Hengel [1,17 ✉]

## Abstract

**Virus infections elicit long-term IgG antibody and memory responses. Human cytomegalovirus (HCMV) is widespread in humans and disseminates despite the presence of virus-specific antibodies. Here, we report that the HCMV Fcγ-binding glycoprotein 34 modulates humoral immunity by binding to IgG⁺ memory B cells. gp34–B cell receptor (BCR) interaction initiates activation of the PDK1/AKT/mTOR/S6 pathway and BCR internalization in a SYK-independent manner. Prolonged stimulation also induces B-cell activation via upregulation of CD69 and CD86. In a T-cell-dependent response, however, interaction with gp34 blocks B-cell proliferation, differentiation into plasmablasts, and soluble IgG production, while stimulating TNF-α secretion. Through gp34 stimulation on IgG⁺ B cells, neighboring IgM⁺ and IgA⁺ B cells are likewise impaired in proliferation, plasmablast formation, and immunoglobulin secretion. In summary, gp34 specifically interacts with IgG⁺ memory B cells, inducing a hyporesponsive state across the B-cell compartment through direct and indirect regulation. This reveals a novel mode of viral evasion from B-cell responses by suppressing secondary immunity.**

**Keywords** Human Cytomegalovirus (HCMV); Glycoprotein 34 (gp34); Humoral Immunity; Plasmablasts; Fcγ
**Subject Categories** Immunology; Microbiology, Virology & Host Pathogen Interaction

## Introduction

Human cytomegalovirus (HCMV) is a beta-herpesvirus which causes lifelong infection in humans with recurring episodes of virus production and shedding (Boeckh and Geballe, 2011). In the immunocompetent host, primary and secondary infections are usually asymptomatic. In immunosuppressed individuals, such as transplant recipients (Boeckh et al, 2003) and HIV-infected patients (Deayton et al, 2004), HCMV reactivation is associated with increased morbidity and mortality. Due to transplacental transmission, HCMV is the most common viral infection acquired in utero, resulting in miscarriage or neurological disorders in newborns (Britt, 2017).

Control of HCMV infection and dissemination is mediated by T and NK cell-driven immune responses (Polic et al, 1998). In addition, B lymphocytes and neutralizing antibodies play a role in HCMV control. In fact, B cell-deficient mice produce 100–1000 times more infectious progeny in tissues during recurrent mouse CMV (MCMV) infection compared to control mice (Jonjić et al, 1994), and adoptive transfer of memory B cells (MBCs) into Rag⁻/⁻ mice is effective to provide long-term protection against an ongoing lethal MCMV infection (Klenovsek et al, 2007). In the MCMV infection model of congenital HCMV infection or infection in transplant recipients, passive immunization with MCMV-specific IgGs results in reduced pathology in newborn mice (Golemac et al, 2008), and prevents viral reactivation (Martins et al, 2019). In humans, the role of HCMV-specific IgGs in protection is debated, despite the fact that neutralizing antibodies can block infection of endothelial cells in vitro (Gerna et al, 2008) and very high doses of HCMV IgG-positive IVIG can prevent fetal infection in pregnancies (Kagan et al, 2021).

[1]Institute of Virology, Medical Center and Faculty of Medicine, University of Freiburg, Freiburg 79104, Germany. [2]Spemann Graduate School of Biology and Medicine (SGBM), University of Freiburg, Freiburg 79104, Germany. [3]Faculty of Biology, University of Freiburg, Freiburg 79104, Germany. [4]Department of Rheumatology and Clinical Immunology, Medical Center and Faculty of Medicine, University of Freiburg, Freiburg 79104, Germany. [5]Institute of Virology, Heinrich-Heine-University, University Hospital of Düsseldorf, Universitätsstr. 1, Düsseldorf 40225, Germany. [6]Berlin Institute of Health (BIH) at Charité - Universitätsmedizin Berlin, Berlin, Germany. [7]Division of Clinical and Experimental Immunology, Institute of Immunology, Center for Pathophysiology, Infectiology and Immunology, Medical University of Vienna, Vienna, Austria. [8]CLIP-Cytometry, Department of Paediatric Hematology and Oncology, 2nd Medical School, Charles University, V Uvalu 84, 150 06, Prague 5, Czech Republic. [9]Institute for Virology, University Medical Center of the Johannes Gutenberg-University Mainz, Mainz D-55131, Germany. [10]Institute of Virology, Ulm University Medical Center, Ulm 89081, Germany. [11]Central Facility for Electron Microscopy, Ulm University, Ulm 89081, Germany. [12]Center for Proteomics University of Rijeka Faculty of Medicine Brace Branchetta 20, Rijeka 51000, Croatia. [13]Center for Chronic Immunodeficiency, University Medical Center Freiburg, Faculty of Medicine, University of Freiburg, Freiburg, Germany. [14]CIBSS – Centre for Integrative Biological Signalling Studies, University of Freiburg, Freiburg, Germany. [15]Present address: Center for Virology and Vaccine Research, Beth Israel Deaconess Medical Center, Harvard Medical School, Boston, MA, USA. [16]These authors contributed equally as first authors: Precious Cramer, Stefan F H Neys. [17]These authors contributed equally as senior authors: Marta Rizzi, Hartmut Hengel. ✉E-mail: marta.rizzi@uniklinik-freiburg.de; hartmut.hengel@uniklinik-freiburg.de

B cells are clonally diverse cells that express surface immunoglobulin (Ig) receptors (BCR), which recognize antigenic epitopes of pathogens. Following antigen re-exposure, MBCs, predominantly IgG$^+$, mount a rapid and robust humoral immune response (Kurosaki et al, 2015; Palm and Henry, 2019). Such MBCs may differentiate into plasmablasts, capable of secreting affinity-matured Igs, crucial for clearing infections (Mesin et al, 2020; Victora and Nussenzweig, 2012). MBCs, together with class-switched plasma cells, form the basis for a strong humoral immune response to antigens upon recall.

HCMV has developed strategies to evade humoral immunity, as inferred from frequent super-infection with new HCMV strains in healthy individuals (Ross et al, 2010), virus transmission in seropositive pregnant women (Lilleri et al, 2013), and low efficacy of HCMV IgG prophylaxis in solid organ and hematopoietic stem cell transplant recipients (Hodson et al, 2007; Raanani et al, 2009). HCMV may escape humoral immunity because it disseminates by cell-to-cell spread (Falk et al, 2018), whereby critical immunogenic envelope glycoproteins are not available for recognition by neutralizing antibodies. This mechanism, though plausible, does not completely explain how the virus is able to disseminate in infected healthy individuals.

HCMV is characterized by an extensive proteome of approximately 750 translational open reading frames (ORFs), identified by ribosome profiling and transcript analysis (Stern-Ginossar et al, 2012). These include several ORFs encoding IgG-Fc (Fcγ) binding glycoproteins which antagonize host Fcγ receptor activation by interacting with specific IgGs bound to HCMV-infected cells (Corrales-Aguilar et al, 2014). The Fcγ-binding glycoproteins gp34, gp68, gp95, and gpRL13 are encoded by independent genes, *RL11, UL119-UL118, RL12*, and *RL13*, respectively (Atalay et al, 2002; Corrales-Aguilar et al, 2014; Cortese et al, 2012). gp34 and gp68 are the best characterized and bind all IgG subclasses (Atalay et al, 2002; Sprague et al, 2008), while *RL13* expression is lost in HCMV laboratory strains due to rapid mutations in the gene upon virus culture (Dargan et al, 2010; Stanton et al, 2010). gp34 and gp68 synergize by recognizing separate sites of human Fcγ. By simultaneously binding to IgG immune complexes, gp34 and gp68 lead to inhibition of FcγRIII-mediated NK cell degranulation (Kolb et al, 2021).

Here, we used purified HCMV virions and soluble recombinant gp34 (gp34$_{1-179}$) as a tool to explore HCMV´s interaction with B cells. First, we demonstrate that gp34$_{1-179}$ specifically interacts with the BCR of IgG$^+$ MBCs. We show that gp34$_{1-179}$ binding to membrane (m)IgG results in activation of the PDK1/AKT/mTOR/S6 pathway and BCR internalization, in a SYK-independent manner, and subsequently induces expression of early activation markers. In the context of T-dependent responses, however, gp34$_{1-179}$ impairs IgG$^+$ B cell proliferation, plasmablast formation, and IgG secretion. We further show that gp34$_{1-179}$-primed IgG$^+$ B cells indirectly suppress IgM$^+$ and IgA$^+$ B cells, leading to a total impaired plasmablast formation and antibody secretion. Collectively, these interactions establish a state of global functional hyporesponsiveness across the B cell compartment and highlight a novel strategy of viral immune evasion directed against memory B cells, suggesting a potential virus-induced defect in recall responses.

# Results

## gp34 is a constituent of the HCMV envelope

Prior studies have reported that HCMV Fcγ-binding glycoproteins are present in the tegument (Stannard and Hardie, 1991; Varnum

et al, 2004), a region that spans the space between the envelope and nucleocapsid. To determine the exact localization of gp34 within the virion, HCMV particles purified from HB5-infected MRC5 cells were subjected to negative staining transmission electron microscopy (TEM). Immunogold-labeling with an anti-gp34 monoclonal antibody (α-gp34 mtrp.04) indicated that gp34 is located in the envelope of HB5. The triple knockout mutant *ΔUL119-118/ΔRL11/ΔRL12* (TKO) lacking all three Fcγ proteins showed no evidence of α-gp34 mtrp.04 binding (Fig. 1). The presence of gp34 in the HCMV virion was independently confirmed by recent studies determining the complete repertoire of proteins associated with the HCMV virion of strains AD169 and Merlin by mass spectrometry (Bentley et al, 2025; Reyda et al, 2014). Together, we show that gp34 is present in the virion as a part of the envelope.

## The ectodomain of gp34 is sufficient to maintain Fcγ binding-mediated functions

Despite the fact that gp34 is expressed as a type I transmembrane glycoprotein on the HCMV virion as well as infected cells, we found that membrane residency of gp34 is dispensable for the gp34-mediated abrogation of FcγR activation (Corrales-Aguilar et al, 2014). Hence, we generated recombinant soluble variants of epitope-tagged gp34 (Fig. EV1A, left) as a tool to identify potential target cells that might interact with gp34. Variants lacking the signal peptide (gp34$_{\Delta S-180}$ and gp34$_{\Delta S-156}$) as well as variant gp34$_{1-124}$ lost the ability to bind Fcγ (Fig. EV1A, right). Variant gp34$_{1-179}$ was the minimum ectodomain region required to maintain Fcγ binding of gp34 as well as retain the ability to antagonize host FcγRIII activation and was therefore suitable for further studies (Fig. EV1A,B,D,E). Importantly, the Flag-tagged point mutant of gp34$_{1-179}$ with a tryptophan to phenylalanine substitution at position 65 (gp34$_{1-179W65F}$) (Fig. EV1B) was unable to bind Fcγ and antagonize host FcγR (Fig. EV1B,D,E). Hence, gp34$_{1-179W65F}$ together with the ectodomain region of an Fcγ–nonrelated protein, ICOS-L$_{1-268}$, were used as controls. Modification of the proteins did not affect maturation and glycosylation (Fig. EV1C). Soluble gp34 variants were V5/His6-tagged for detection and immune-affinity purification (Fig. EV1C).

## Soluble gp34$_{1-179}$ binds to IgG$^+$ B cells

Incubation of gp34$_{1-179}$ with freshly isolated PBMCs from healthy donors resulted in binding of gp34$_{1-179}$ but not gp34$_{1-179W65F}$ to CD19$^+$ B cells and CD16$^+$ NK cells but not to CD3$^+$ T cells (Appendix Fig. S1A). Physical interaction between gp34$_{1-179}$ and NK cells was found to be mediated by cytophilic IgGs already bound to FcγRIIIA (CD16) on primary NK cells (Appendix Fig. S2B,C).

All B cells express the low-affinity Fcγ receptor IIb (CD32b) (Rankin et al, 2006; Veri et al, 2007). Since gp34 is known to antagonize antibody-mediated host FcγR activation (Corrales-Aguilar et al, 2014), we first hypothesized that gp34$_{1-179}$ might modulate B cell function by interacting with IgGs bound to CD32b (Dickler and Kunkel, 1972). Thus, we incubated isolated human B cells with gp34$_{1-179}$, ICOS-L$_{1-268}$, and gp34$_{1-179W65F}$ and assessed their binding by flow cytometry. gp34$_{1-179}$ was bound to 15% of the B cells while no binding was observed for control proteins gp34$_{1-179W65F}$ and ICOS-L$_{1-268}$ (Fig. 2A; Appendix Fig. S2A).

## HB5

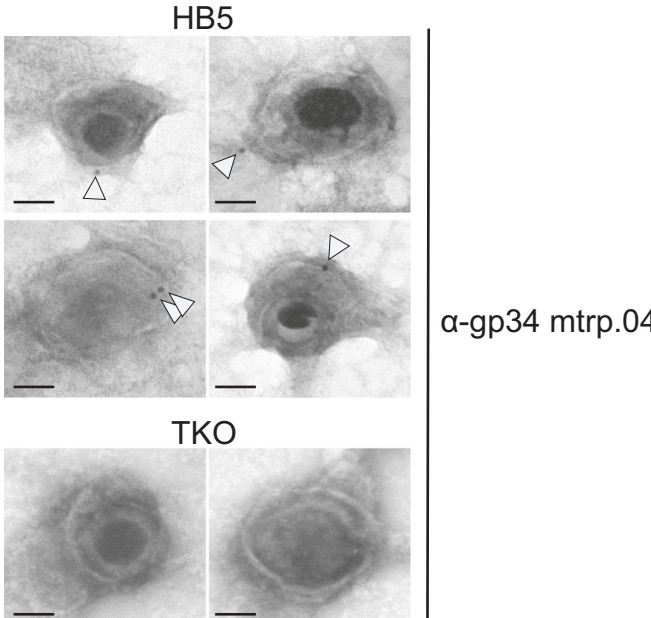

α-gp34 mtrp.04

## TKO

**Figure 1.  HCMV gp34 is a constituent of the virion envelope.**

Purified HB5 and HB5 *UL119-118/ΔRL11/ΔRL12* (TKO) virions were analyzed by negative staining TEM following immuno-gold labeling with α-gp34 mtrp.04 antibody and secondary antibodies conjugated to 10 nm gold particles (arrowheads). The top four panels show virions from wild-type HB5, and the bottom 2 panels represent virions from triple knockout (TKO) HB5. Scale bars indicate 100 nm.

gp34$_{1-179}$-bound cells were exclusively IgG$^+$ B cells (Fig. 2B,C; Appendix Fig. S2B). Supporting these data, binding of gp34$_{1-179}$ was observed in IgM$^-$/IgG$^+$/CD32b$^-$ RPMI 8866 cells, but not in IgM$^+$/IgG$^-$/CD32b$^+$ Daudi cells (Fig. 2D). Hence, CD32b expression did not mediate binding of gp34$_{1-179}$ to IgG$^+$ B cells. The specific interaction of gp34 with IgG$^+$ B cells was confirmed by immuno-precipitation using gp34$_{1-179}$ as bait in lysates from EBV-transformed B cell clones expressing either IgG or IgD BCR (Appendix Fig. S2C,D). These results corroborated that gp34$_{1-179}$ interacts exclusively with IgG$^+$ B cells via the BCR.

### SYK-independent activation of IgG$^+$ B cells by gp34$_{1-179}$

Crosslinking of the BCR triggers a signaling cascade resulting in B-cell activation (Fig. EV2A). B cell fate following activation, however, depends largely on the type of antigen triggering the BCR, the activation of co-stimulatory pathways, and the presence of T-cell help (Batista and Harwood, 2009; Kurosaki et al, 2010). Depending on the environment, BCR stimulation may thereby result in either B cell survival, activation, and expansion, or in inhibition, anergy, and even elimination (Elizabeth Franks and Cambier 2018; Goodnow, 1996; Noelle and Erickson, 2005). Hence, we first tested if binding of gp34$_{1-179}$ to IgG$^+$ B cells elicited signaling responses. We used α-BCR and gp34$_{1-179W65F}$ as positive and negative controls, respectively. In primary B cells, gp34$_{1-179}$ triggered Lyn phosphorylation to a similar extent as BCR stimulation by α-BCR (Fig. 3A,B). gp34$_{1-179}$, however, failed to

induce p-SYK (Fig. 3B), which was not the result of a delayed response (Fig. EV3A). Importantly, simultaneous stimulation with gp34$_{1-179}$ and α-BCR resulted in a 50% reduction of SYK phosphorylation after 5 min (Fig. 3C), indicating that gp34$_{1-179}$ actively modulates proximal BCR signaling outcome when in competition with α-BCR stimulation. Active SYK recruits the adapter protein BLNK/SLP-65, which provides a docking site enabling SYK to phosphorylate effector enzymes such as PLCγ2, at tyrosine residue Y759 (Hashimoto et al, 1999; Kim et al, 2004; Kurosaki and Tsukada, 2000). We found that in line with the absence of detectable p-SYK induction, gp34$_{1-179}$ stimulation was accompanied by unaltered PLCγ2 (Y579) phosphorylation levels (Fig. 3B).

Docking protein 3 (DOK3) associates with Lyn to restrict SYK activation and calcium signaling (Lösing et al, 2013; Stork et al, 2007). We observed an increase in DOK3 synthesis in gp34$_{1-179}$-stimulated B cells after 1 h (Fig. EV3B). Prolonged binding of gp34$_{1-179}$ may thereby result in more frequent association of DOK3 with Lyn, reducing SYK activity over time.

BCR activation leads to receptor internalization, thereby mediating antigen processing and presentation, key in building a proper immune response (Chu et al, 1984). gp34$_{1-179}$ altered the dynamic of membrane IgG (mIgG) internalization that was significantly delayed within the first hour but became similar to α-BCR stimulation after 6 h of stimulation (Fig. 3D). The level of BCR internalization is highly dependent on BCR signaling strength (Hou et al, 2006), although BCR internalization can occur independently of active SYK (Caballero et al, 2006; H. Ma et al, 2001). Therefore, a delayed BCR internalization in gp34$_{1-179}$-treated B cells could be the result of alterations or delays in the activation of different pathways downstream of the BCR.

Stimulation via the BCR results in upregulation of co-stimulatory molecules CD69, MHC-II, CD86, and CD80. gp34$_{1-179}$ stimulation induced the expression of CD69 and CD86, and to a lesser extent MHC-II, but failed to upregulate CD80 in IgG$^+$ B cells compared to α-BCR stimulation (Fig. 3E). Next, we examined whether HCMV virion-resident gp34 could interact with B cells in a similar manner as soluble gp34 (gp34$_{1-179}$). Like in the soluble form, virion-associated gp34 activated B cells by upregulating CD69 and CD86 after 24 and 48 h, respectively (Fig. EV4A).

Whereas BCR stimulation with α-BCR induces crosslinking of the BCR through both heavy and light chains, gp34$_{1-179}$ engages only the IgG heavy chain (Anderson et al, 1967; Kolb et al, 2021; Sprague et al, 2008). To determine whether these distinct signaling outcomes between gp34$_{1-179}$ and α-BCR were due to differences in heavy and/or light chain binding, we compared BCR signaling responses between α-IgG (H + L), α-IgG (H), and total BCR (α-Ig) stimulation. Similar signaling responses were observed between these different stimuli (Appendix Fig. S3A), indicating that the altered responses detected for gp34$_{1-179}$ are not attributable to variations in BCR binding sites. Furthermore, we could show that these differences were not caused by changes in total protein levels of Syk, Lyn, or PLCγ2 during stimulation (Appendix Fig. S3B). In summary, our data indicate that both virion-resident and soluble gp34$_{1-179}$ activated IgG$^+$ B cells in an unconventional manner, impacting downstream IgG BCR signaling as well as effector functions.

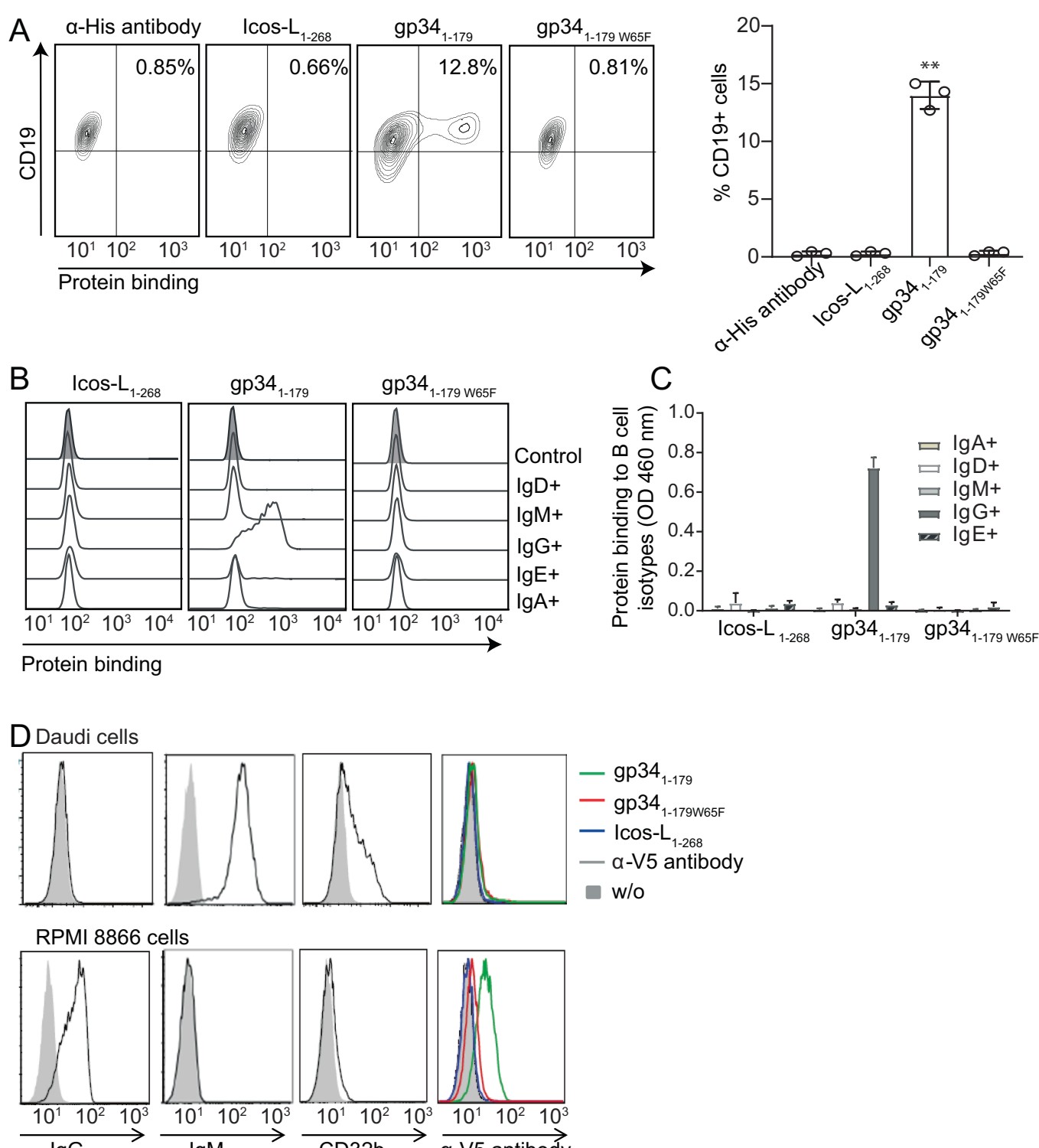

## gp34$_{1-179}$ activates mTORC1 through the PI3K signaling axis

As gp34$_{1-179}$ failed to induce SYK phosphorylation in IgG$^+$ B cells, we assessed SYK-independent BCR activation pathways downstream of Lyn (Fig. EV2A). Lyn can phosphorylate CD19 in a SYK-

independent manner (Fujimoto et al, 1998; Hasegawa et al, 2001), subsequently activating phosphatidylinositol 3-kinase (PI3K) (Fruman et al, 2017; Greaves et al, 2019; Manning and Toker, 2017). As a readout for PI3K activation, we measured phosphorylation of phosphoinositide-dependent kinase 1 (PDK1), which in turn is responsible for protein kinase B (AKT) phosphorylation at T308.

**Figure 2.  gp34$_{1-179}$ specifically binds to IgG$^+$ B cells.**

(A) Binding of 10 µg/ml each of histidine-tagged gp34$_{1-179}$, gp34$_{1-179W65F}$, or ICOS-L$_{1-268}$ to isolated human B cells was assessed by flow cytometry. Bar graphs depict the percentage of CD19$^+$ cells bound by each recombinant protein. Three independent experiments were performed and analyzed by a Kruskal–Wallis test (**$P = 0.0417$). The error bars are standard deviations of the mean. (B) Specific B-cell receptor isotypes binding to histidine-tagged recombinant proteins were assessed by flow cytometry. (C) Binding of gp34$_{1-179}$, gp34$_{1-179W65F}$, or ICOS-L$_{1-268}$ to different immunoglobulin isotypes was studied by ELISA. α-IgG, α-IgA, α-IgM, α-IgD, or α-IgE antibodies-coated plates were incubated with isolated B cells. Indicated Histidine-tagged recombinant proteins were then co-incubated on the ELISA plate. Recombinant proteins bound by each B-cell isotype were detected with peroxidase-conjugated anti-His antibodies. The error bars are the standard deviation of the mean of two biological replicates ($n = 2$). (D) Daudi and RPMI 8866 cell lines were analyzed by flow cytometry for surface expression of IgG, IgM, and CD32b. The cell lines were examined for gp34$_{1-179}$, gp34$_{1-179W65F}$, or ICOS-L$_{1-268}$ (all V5 tagged) binding with an anti-V5 antibody.

Similar to α-BCR stimulation, we observed a two-step activation of AKT at both positions T308 and S473 by gp34$_{1-179}$ (Fig. 4A), implying a maximally active and stable protein (Manning and Toker, 2017). These data suggest that gp34$_{1-179}$ can trigger the metabolic PI3K/PDK1/AKT signaling axis and activate the mechanistic target of rapamycin complex 1/2 (mTORC1/2) pathway to the same extent as classical BCR ligation.

Complete phosphorylation of AKT depends on conformational changes which occur after activation of PI3K leading to the initial phosphorylation at residue T308 thereby exposing residue S473. To evaluate if AKT T308 phosphorylation was truly dependent on PI3K activation in gp34$_{1-179}$-activated cells, we blocked the catalytic subunit of PI3K with Wortmannin (Arcaro and Wymann, 1993; Wymann et al, 1996). As expected, B cells stimulated with α-BCR in the presence of Wortmannin showed a decreased induction of AKT S473 phosphorylation, although this did not return to unstimulated levels (Fig. 4B). Conversely, in gp34$_{1-179}$ stimulated cells, Wortmannin completely blocked p-AKT S473 induction (Fig. 4B). These data show that where α-BCR stimulation depends only partially on PI3K activation for the phosphorylation of AKT, gp34$_{1-179}$ depends exclusively on the Lyn-activated PI3K pathway for AKT activation.

mTORC1 and FOXO1 (Foxhead box protein 1) are two major downstream targets of AKT, and are responsible for protein synthesis, B-cell survival, and proliferation (Manning and Toker, 2017). Therefore, we analyzed S6 phosphorylation as a downstream target of mTORC1. α-BCR stimulation resulted in S6 phosphorylation in all B-cell subsets, including IgD$^+$ cells (Fig EV3C,D). gp34$_{1-179}$ also induced p-S6 in IgG$^+$ B cells, although to a significantly lower extent (Fig. 4C,D). We again confirmed that these distinct signaling outcomes were not attributable to the difference in binding site to the BCR, nor to alterations in protein expression of AKT and S6 during stimulation (Appendix Fig. S3A,B). Together, we conclude that binding of gp34$_{1-179}$ to IgG$^+$ B cells activates the PI3K/AKT/mTOR/S6 pathway in a SYK-independent manner.

## gp34$_{1-179}$ blocks plasmablast formation and immunoglobulin secretion

Because gp34 induced B cell receptor signaling and early B cell activation events, we next studied the role of gp34 in T-dependent B cell responses. To this end, we made use of an in vitro model by stimulating isolated B cells with CD40L and IL-21. We first examined whether HCMV virion-resident gp34 could be used for these experiments. These results indicated that for these culture assays spanning over multiple days, HCMV virions were not suitable, as they induced apoptosis after three days (Fig. EV4B,C).

We therefore continued all experiments with the soluble gp34 (gp34$_{1-179}$) variant.

T-dependent B-cell stimulation with CD40L and IL-21 induced a strong burst in B cell proliferation and differentiation into plasmablasts (Fig. 5A,B). The presence of gp34$_{1-179}$ in this setting, however, led to a marked reduction in B cell proliferation compared to all three control groups (CD40L/IL-21 alone, or with the addition of α-BCR or gp34$_{1-179W65F}$; Fig. 5A; Appendix Fig. S4C). Where the addition of BCR stimulation (α-BCR) even enhanced plasmablast formation in CD40L/IL-21-cultured B cells, the presence of gp34$_{1-179}$ impaired plasmablast formation (Fig. 5B; Appendix Fig. S4B). This was accompanied by a complete block in the production of soluble (s)IgG (Fig. 5C, left). Strikingly, sIgM and sIgA were also absent in the culture supernatant (Fig. 5C, middle and right).

When B cells were stimulated with a different Fcγ-binding protein, *Staphylococcus aureus* protein A (SpA), CD69 and CD86 were upregulated to a similar degree as compared to α-BCR (Fig EV5A). In contrast to gp34$_{1-179}$, however, SpA could efficiently induce B-cell proliferation and plasmablast formation in B cells cultured in the presence of CD40L/IL-21 (Fig. EV5B,C). These findings indicate the use of a novel mechanism in which HCMV-derived gp34 regulates B-cell effector function. From the total peripheral blood B-cell population, 15 to 20% are IgG$^+$ (Fecteau et al, 2006; Seifert and Küppers, 2016). To exclude that gp34$_{1-179}$ had a direct impact on non-IgG$^+$ B cells, we selectively depleted IgG$^+$ B cells prior to stimulation. In this context, proliferation and plasmablast differentiation of IgG$^-$ B cells were unaffected in the presence of gp34$_{1-179}$ (Fig. 5D,E). We therefore conclude that gp34$_{1-179}$ impairs proliferation, plasmablast differentiation, and subsequent immunoglobulin secretion in IgM$^+$ and IgA$^+$ B cells by reprogramming IgG$^+$ B cells.

We hypothesized that paracrine signals from IgG$^+$ B cells may have caused the effects observed on IgM$^+$ and IgA$^+$ B cells in gp34$_{1-179}$-treated cultures. We therefore screened the supernatant of these different B cell cultures for alterations in cytokine profiles. B cell cultures stimulated with CD40L and IL-21 in the presence of gp34$_{1-179}$, but not α-BCR or gp34$_{1-179W65F}$, contained increased concentrations of the pro-inflammatory cytokine tumor necrosis factor alpha (TNF-α; Fig. 5F). B cells treated with TNF-α prior to LPS stimulation become unresponsive (Frasca et al, 2012), and in HCMV-seropositive pregnant women, MBCs produce high amounts of TNF-α when polyclonally stimulated (Dauby et al, 2014). Hence, through TNF-α production, gp34-bound IgG$^+$ B cells could suppress neighboring non-IgG$^+$ B cells. A definitive causal relationship, however, could not be established (Appendix Fig. S5).

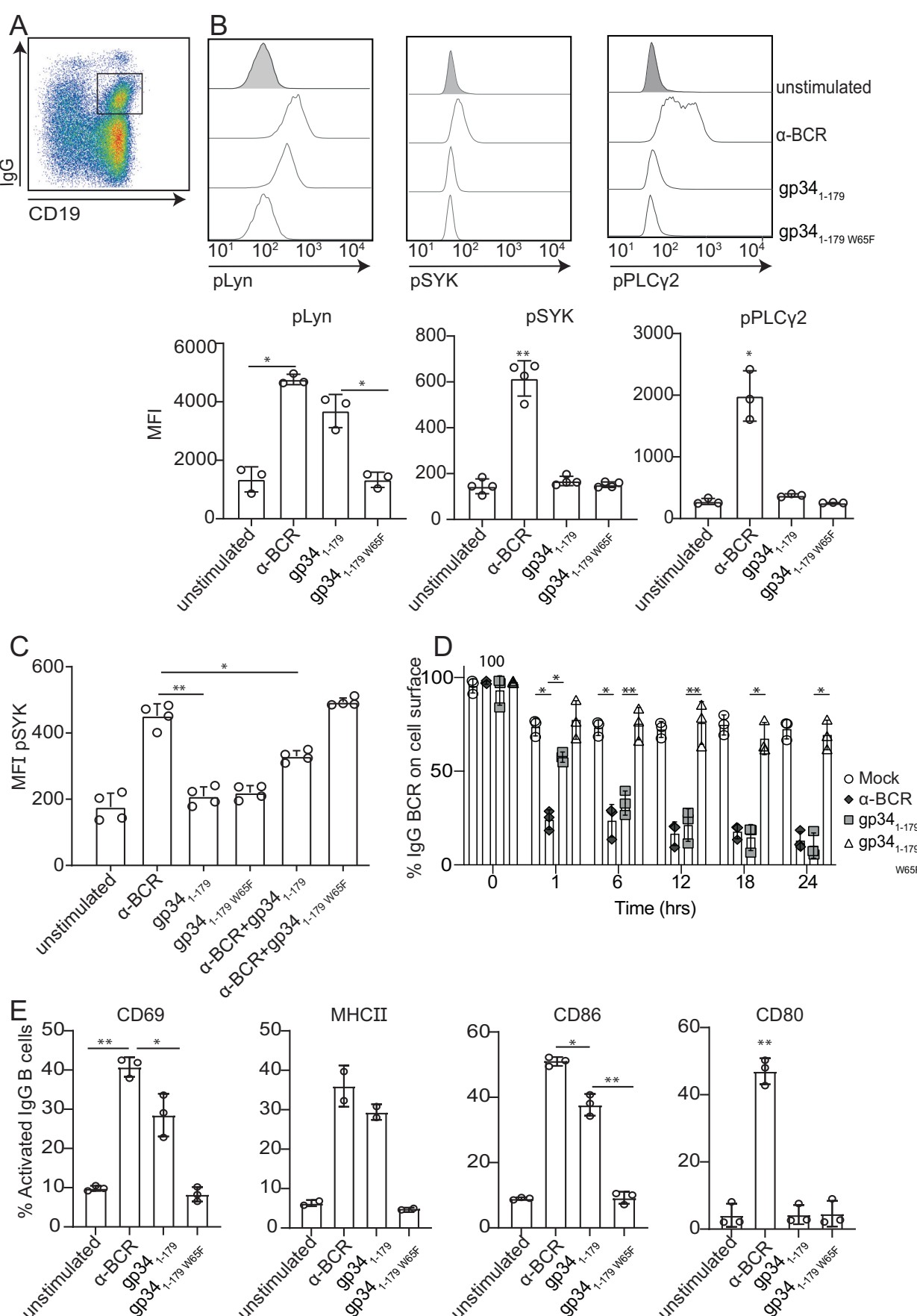

◄

**Figure 3. gp34$_{1-179}$ activates IgG$^+$ B cells in a SYK-independent manner.**

(A) Gating strategy identifying IgG$^+$ B cells in live CD19$^+$ B cells. (B) Differential expression of phosphorylated Lyn, SYK, and PLCγ2 proteins by flow cytometry in IgG$^+$ B cells upon stimulation with α-BCR, gp34$_{1-179}$, and gp34$_{1-179W65F}$ for 5 min. Bar graphs represent the mean fluorescence intensity (MFI) of the phosphorylated proteins in at least three independent experiments performed in triplicate from different donors. Error bars represent the standard deviation of the mean. pLyn and PLCγ2 were analyzed using a Kruskal–Wallis test, while pSYK was analyzed with a one-way ANOVA. *$P = 0.0384$ and *$P = 0.0395$ (pLyn), **$P = 0.0243$ (pSYK) and *$P = 0.0461$(pPLC γ2). (C) B cells were incubated with the indicated stimuli for 5 min. Phosphorylated SYK in IgG$^+$ B cells was measured by flow cytometry, and the MFI is shown as bar graphs of four independent experiments of technical duplicates. Error bars indicate the standard deviation of the mean. The statistical test performed was a one-way ANOVA with *$P = 0.0335$ and **$P = 0.022$. (D) Flow cytometry assessment of IgG BCR internalization after in vitro stimulation of IgG$^+$ B cells. Error bars indicate the standard error of the mean of three independent experiments. Two-way ANOVA (Tukey's test, with Greenhouse–Geisser correction) was performed to assess statistical significance *$P = 0.043$, **$P = 0.0062$. (E) Assessment of B cell activation by measuring CD69, MHCII, CD86, and CD80 in IgG$^+$ B cells upon stimulation with gp34$_{1-179}$, gp34$_{1-179W65F}$, or α-BCR. Analysis of CD69 and MHCII expression was performed after 24 h, and upregulation of CD86 and CD80 was determined after 48 h. Data are two (for MHCII) or three (for CD69, CD86, CD80) biological replicates. Only markers with $n = 3$ were analyzed using Kruskal–Wallis test [(CD69: *$P = 0.0361$, **$P = 0.014$), CD80: **$P = 0.048$ and CD86: *$P = 0.0409$, **$P = 0.020$)]. Error bars = mean ± SD.

## gp34$_{1-179}$-mediated regulation of B-cell responses is independent of apoptotic cell death

Because of the severe block in B-cell proliferation and differentiation by gp34$_{1-179}$, we hypothesized these effects could be mediated through the induction of cell death. We observed that apoptosis was induced only in staurosporine-treated B cells, and in α-BCR, gp34$_{1-179}$, or gp34$_{1-179W65F}$ stimulation (Fig. 6A,B). Also, in CD40L/IL-21 pre-activated B cells, active apoptosis, indicated by caspase-3 cleavage, was induced only in staurosporine and not in gp34$_{1-179}$-treated cells (Fig. 6C,D).

CD69 is an early and transient activation marker following B cell activation, and is important for retention of lymphocytes in the lymph node (González-Amaro et al, 2013). The addition of gp34$_{1-179}$ in B cells stimulated with CD40L/IL-21 completely blocked proliferation (Appendix Fig. S4C), but resulted in a marked and persistent CD69 expression for up to 9 days (Fig. 7A). These findings suggest that gp34$_{1-179}$ initiates very early signaling and activation events but impairs further progression to effector functions in a T-dependent B cell response. Hence, gp34$_{1-179}$ binding does not induce cell death, but rather induces a hyporesponsive state where not only IgG$^+$, but also IgM$^+$ and IgA$^+$ B cells no longer respond to T-dependent stimuli.

## The hyporesponsive state of B cells by gp34$_{1-179}$ is reversible

Finally, we examined whether the impaired B cell responses, induced by gp34$_{1-179}$, were reversible. To this end, B cells were cultured with CD40L/IL-21 in the presence of gp34$_{1-179}$, after which gp34$_{1-179}$-neutralizing antibodies (mtrp.06) were added for the remaining culture period. At day six of the culture, the effects of gp34$_{1-179}$ on B cell proliferation and plasmablast formation could partly be restored (Fig. 7B,C). Our findings therefore indicate that gp34$_{1-179}$-induced B cell hyporesponsiveness towards T-dependent stimulation is partially reversible.

## Discussion

We report that soluble gp34 (gp34$_{1-179}$), used here as a tool to mimic HCMV virion-resident gp34, specifically binds to mIgG. IgG$^+$ MBC stimulation with gp34$_{1-179}$ resulted in the activation of the PI3K/AKT/mTOR/S6 pathway in a SYK-independent manner

and a subsequent delay in BCR internalization. Prolonged stimulation with gp34$_{1-179}$ also resulted in the upregulation of early B-cell activation markers, such as CD69 and CD86. In the context of B-cell responses towards T helper-like stimuli, like CD40L and IL-21, gp34$_{1-179}$ blocked proliferation, plasmablast formation, and IgG secretion. We also demonstrated that it had an unexpected influence on neighboring non-IgG$^+$ B cells. Specifically, via conditioning IgG$^+$ B cells, it indirectly impaired the proliferation, plasma cell formation, and immunoglobulin secretion of IgM$^+$ and IgA$^+$ B cells. In addition, gp34$_{1-179}$ stimulated TNF-α production in these B-cell cultures. We found that this was not mediated by B-cell death induced by apoptosis, but rather by the transition into a hyporesponsive state. Hence, we describe HCMV-derived gp34 as a novel type of viral evasion from B-cell effector function by impairing secondary immune responses.

gp34 forms stable disulfide-linked homodimers and oligomers which recognize and bind selectively to the hinge region of the IgG heavy chain, a site known to mediate contact between IgG and FcγRs (Kolb et al, 2021; Sprague et al, 2008). Recent electron-nmicroscopic analysis of human IgG revealed that the conformation of the Fc domain is conserved in both sIgG and mIgG, which is confirmed by the free accessibility of gp34 to the BCR on IgG$^+$ MBCs (Ma et al, 2022). The interaction between gp34$_{1-179}$ and IgG was shown to block IgG-mediated activation of FcγRI, II, and III (Corrales-Aguilar et al, 2014). Due to its particularly high affinity to Fcγ, with Kd values in the low nanomolar range (Qerqez et al, 2025), gp34$_{1-179}$ readily precipitated and cross-linked mIgG and thereby displaced the inhibitory receptor CD32b from its interaction with mIgG when bound to an antigen (Appendix Fig. S6) (Kolb et al, 2021). According to classical concepts, CD32b plays a key role in the regulation of late humoral responses when complexes of antigen and IgG are present and co-ligate CD32b with the BCR (Ravetch and Bolland, 2001). While the stoichiometry of the gp34$_{1-179}$:mIgG complex is not yet known, the structural features of gp34$_{1-179}$ homodimers and oligomers are readily compatible with a dissociation activation model of BCR signaling (Maity et al, 2015; Yang and Reth, 2010, 2016), rather than a crosslink model of BCR monomer activation. This notion is based on the fact that SYK, but not Lyn, is required for opening and stabilizing the BCR by binding to dual-phosphorylated ITAMS on Igα/β. The finding that gp34$_{1-179}$ prevents SYK phosphorylation (Fig. 3B,C) highlights the critical role of SYK in BCR opening and amplification of the signaling strength and duration (Kläsener et al, 2014). A limitation of our results, however, is that we cannot

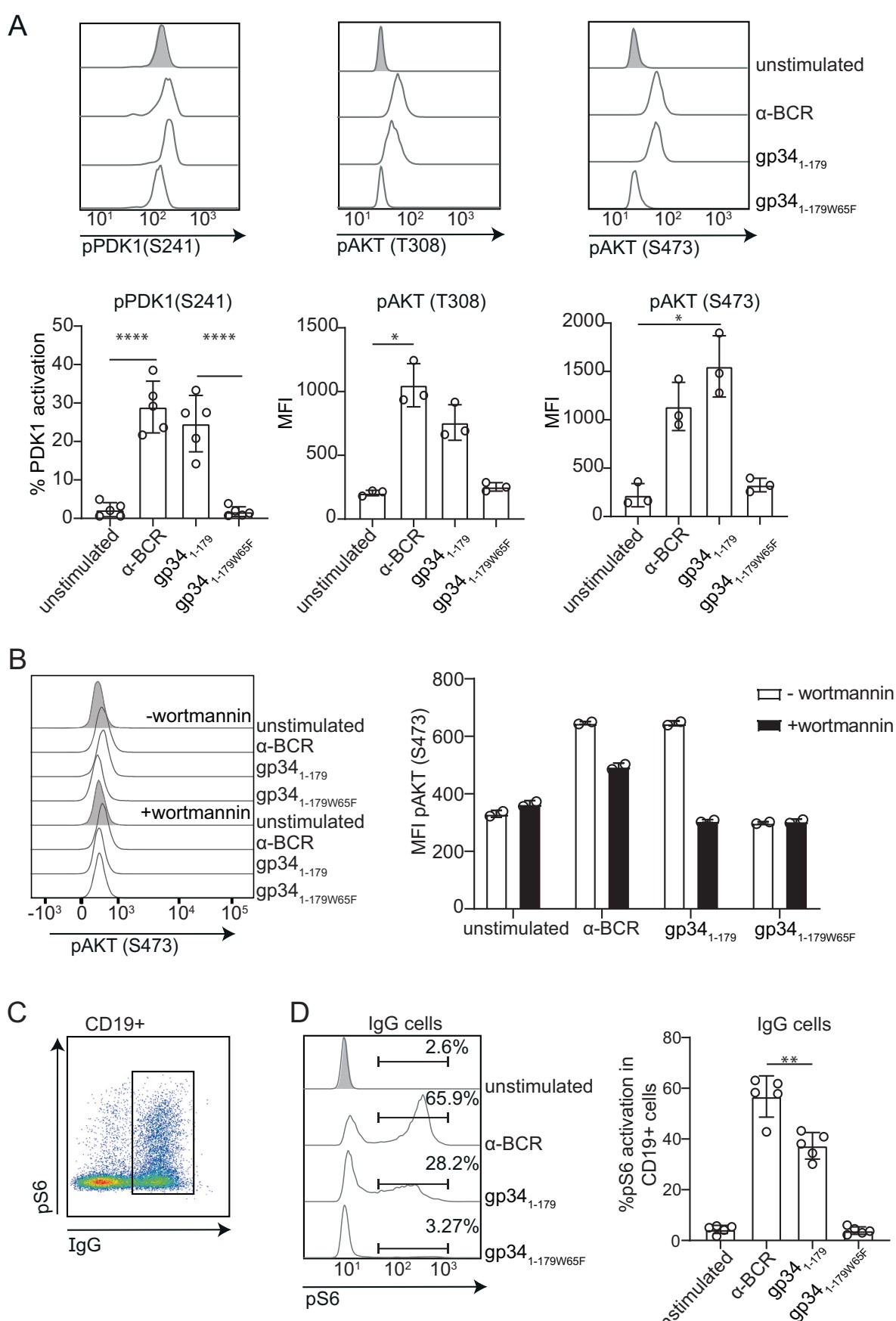

**Figure 4. Recombinant gp34$_{1-179}$ activates the mTORC1 signaling axis in IgG + B cells.**

(A) Phosphorylation of PDK1 and AKT at the indicated amino acid residues after stimulation of isolated B cells with 10 μg/mL of α-BCR, gp34$_{1-179}$, or gp34$_{1-179W65F}$ for 10 min. Bars represent percentages of PDK1-positive IgG$^+$ B cells or MFI of pAKT (T308) and pAKT (S473) in IgG$^+$ B cells. At least three replicates were analyzed, and differences were analyzed using one-way ANOVA for PDK1 (****$P < 0.0001$) or a Kruskal–Wallis test for pAKT (T308, *$P = 0.0343$) and pAKT (S473, *$P = 0.023$). Error bars + standard deviation of the mean. (B) B cells were incubated for 15 min with 2 μM Wortmannin or DMSO prior to stimulation with α-BCR, gp34$_{1-179}$, or gp34$_{1-179W65F}$ for 10 min. pAKT (S473) was analyzed by flow cytometry ($n = 2$). (C) Gating strategy to identify IgG$^+$ cells in the live B cells gate expressing phosphorylated S6. (D) Histograms show the MFI of p-S6 within the IgG$^+$ B cells gate. Bars represent the percentage of pS6-positive cells within the IgG$^+$ B cell population. Data represent five independent experiments; error bars indicate standard deviation, and differences between groups were analyzed using one-way ANOVA (**$P = 0.0047$).

exclude that gp34 induces a very early but transient activation of SYK.

Our finding that gp34$_{1-179}$ possesses the ability to disrupt the engagement of CD32b with mIgG upon antigen encounter provides an explanation for the previously puzzling observation that gp34 blocks not only the activating FcγRI, II, and III (Corrales-Aguilar et al, 2014), but also the inhibitory FcγRIIb with undiminished rigor. Due to its specificity for IgG but no other BCR isotype (Fig. 2B–D), gp34$_{1-179}$ could be used as a unique tool to study very early events after mIgG crosslinking and help resolve the temporal and mechanistic sequence of events in MBCs.

The Fcγ-binding proteins encoded by HCMV are part of a growing number of viral as well as microbial virulence factors that subvert humoral immunity by recognizing the Fcγ domain (Otero et al, 2025). Most of the Fcγ-targeting mechanisms interfere with the effector function mediated by host FcγRs and complement system (C1q) through competition with the recruitment of IgG, as documented for protein A (*Staphylococcus aureus*, SpA), protein G (*Streptococcus sp.*), the phage-encoded protein TspB from *Neisseria meningitidis*, and glycoprotein E of herpes simplex virus (Dubin et al, 1991; Forsgren and Nordstrom, 1962; Forsgren and Sjöquist, 1966; Müller et al, 2013). One of the best characterized is SpA, which forms a 42 kDa oligomeric surface and secreted protein. SpA contains a binding site which recognizes Fcγ between the $C_H2$ and $C_H3$ domains and is well known to confound B cell responses (Falugi et al, 2013; Silverman and Goodyear, 2006). While SpA blocks IgG hexamerization and complement activation through competitive binding to the Fc-Fc interaction interface on IgG monomers (Cruz et al, 2021), it has additional binding properties involving engaging certain VH framework regions of the BCR (Graille et al, 2000). This interaction results in downregulation of the BCR, induction of an activation phenotype, limited rounds of proliferation, and eventually apoptotic B-cell deletion (Goodyear and Silverman, 2003). These reports prompted us to compare gp34$_{1-179}$ and SpA with regard to its effect on plasmablast formation in the presence of CD40L and IL-21 (Fig. EV5B). In marked contrast to the inhibition by gp34$_{1-179}$, SpA strongly increased the number of plasmablasts formed, underscoring the totally opposed consequences of mIgG engagement.

The life cycle of cytomegaloviruses comprising alternating phases of lifelong latency and periodic reactivation is built on the well-elaborated manipulation of immune cells and their signaling pathways. While CMV infection of myeloid cells allows for their immediate manipulation (Baasch et al, 2021), non-permissive lymphocytes like T cells and NK cells are indirectly influenced leading to the accumulation of CMV-specific CD8$^+$ T memory cells and the induction and expansion of adaptive memory-like NK cells with distinct epigenetic and functional features (Gumá et al, 2004;

Klenerman and Oxenius, 2016). In line with the impact of CMV on immunological memory, our data show that HCMV controls B cell memory as well. Overall frequencies of late differentiated B cells including the CD27$^+$IgG$^+$ subpopulation remained quantitatively unaltered in HCMV-infected versus uninfected healthy individuals (Goldeck et al, 2016). Assessment of HCMV-specific MBC precursor frequencies by limiting dilution approaches revealed slightly increased numbers in older healthy subjects (Aberle and Puchhammer-Stöckl, 2012). Frequencies of glycoprotein B-binding MBC ranged from 0.33 to 1.4% of all MBCs (Pötzsch et al, 2011). Given the huge HCMV proteome of more than 750 translational products (Stern-Ginossar et al, 2012) and the high frequency of repetitive exposure to HCMV antigens produced during reactivation events, the absolute numbers of MBCs appear to be surprisingly low. Considering the relatively low overall levels of HCMV-specific antibodies and the conspicuously protracted time course of seroconversion after primary HCMV infection (Corrales-Aguilar et al, 2016; Fornara et al, 2015; Lilleri et al, 2016), it is tempting to speculate that gp34 may be involved in this attenuation of humoral immunity. It is therefore conceivable that gp34 released from infected cells and gp34-containing HCMV virions represents an auto-regulatory mechanism that is essential to avoid harmful overstimulation of the humoral immune response and to escape rapid reactivation of resting MBCs upon HCMV antigen encounter. Under pathological conditions with uncontrolled systemic HCMV replication, however, gp34 on the virions could contribute to the loss of MBCs and anamnestic IgG and IgM responses as reported for patients with chronic HIV-1 infection (Titanji et al, 2006).

Compatible with the reports of intact MBC frequencies in healthy HCMV-infected individuals (Aberle and Puchhammer-Stöckl, 2012; Goldeck et al, 2016), we could not detect the hallmarks of apoptosis in gp34$_{1-179}$ and CD40L/IL-21-stimulated MBCs (Fig. 6). Instead, gp34$_{1-179}$ binding to mIgG modulated the response to T-dependent reactivation of IgG$^+$ MBCs, inhibiting their proliferation and differentiation to plasmablasts. In fact, the activation resulting from gp34$_{1-179}$ binding to mIgG is unconventional, with selective activation of proximal tyrosine kinase Lyn (but not SYK or PLCγ2) and mTORC1 signaling, which failed to drive proliferation (Fig. 5A; Appendix Fig. S4C). Remarkably, in the context of T-dependent activation mimicking germinal center responses, gp34$_{1-179}$ induced a state of prolonged CD69 expression in the total B cell compartment, a surface molecule involved in lymphocyte retention and metabolism in lymphoid organs (Chen et al, 2023; Germain et al, 2021; González-Amaro et al, 2013). We showed that removal of gp34$_{1-179}$ restores in part, the functionality of B cells (Fig. 7B,C), however, HCMV establishes lifelong persistence in patients. As such, it is conceivable that B cells that come into contact with gp34 will remain in a hyporesponsive state

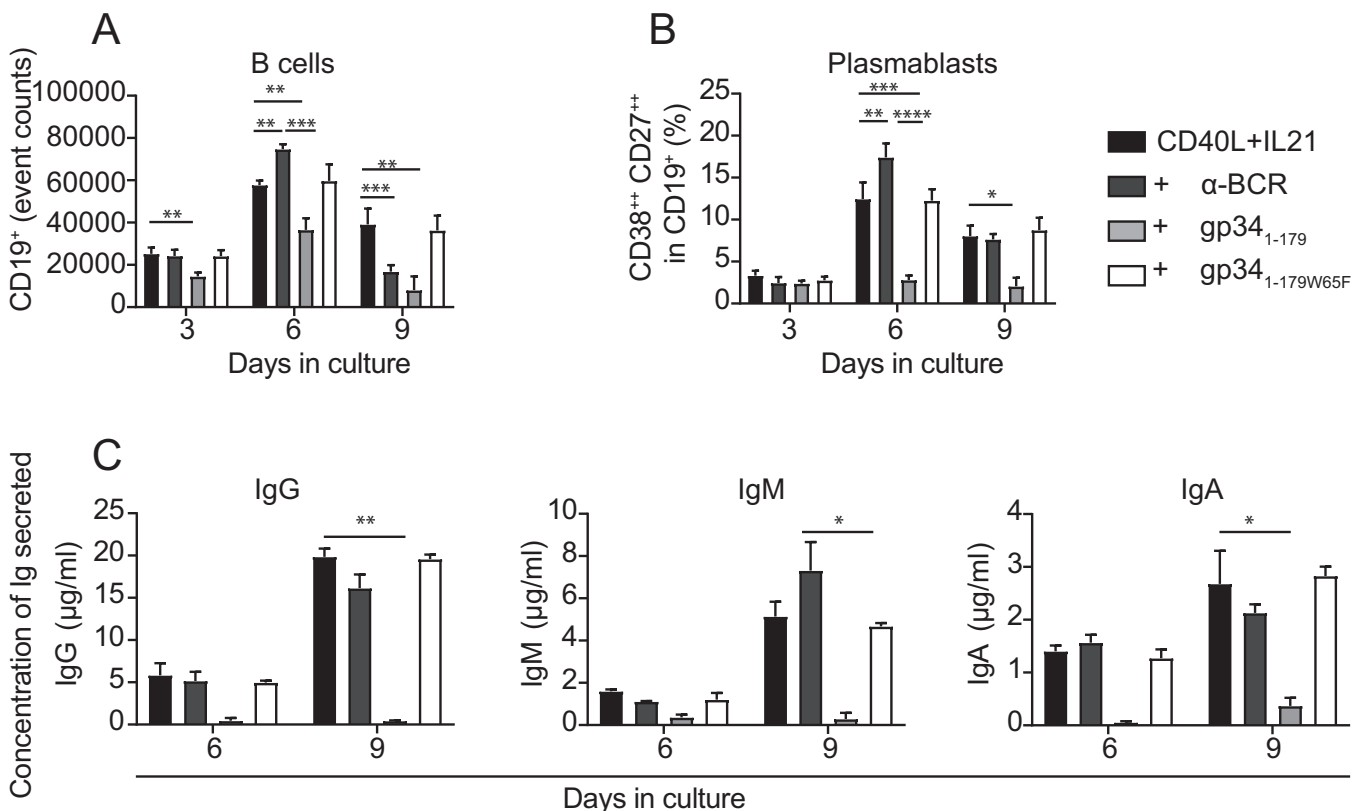

B cells **depleted** of IgG+ B cells

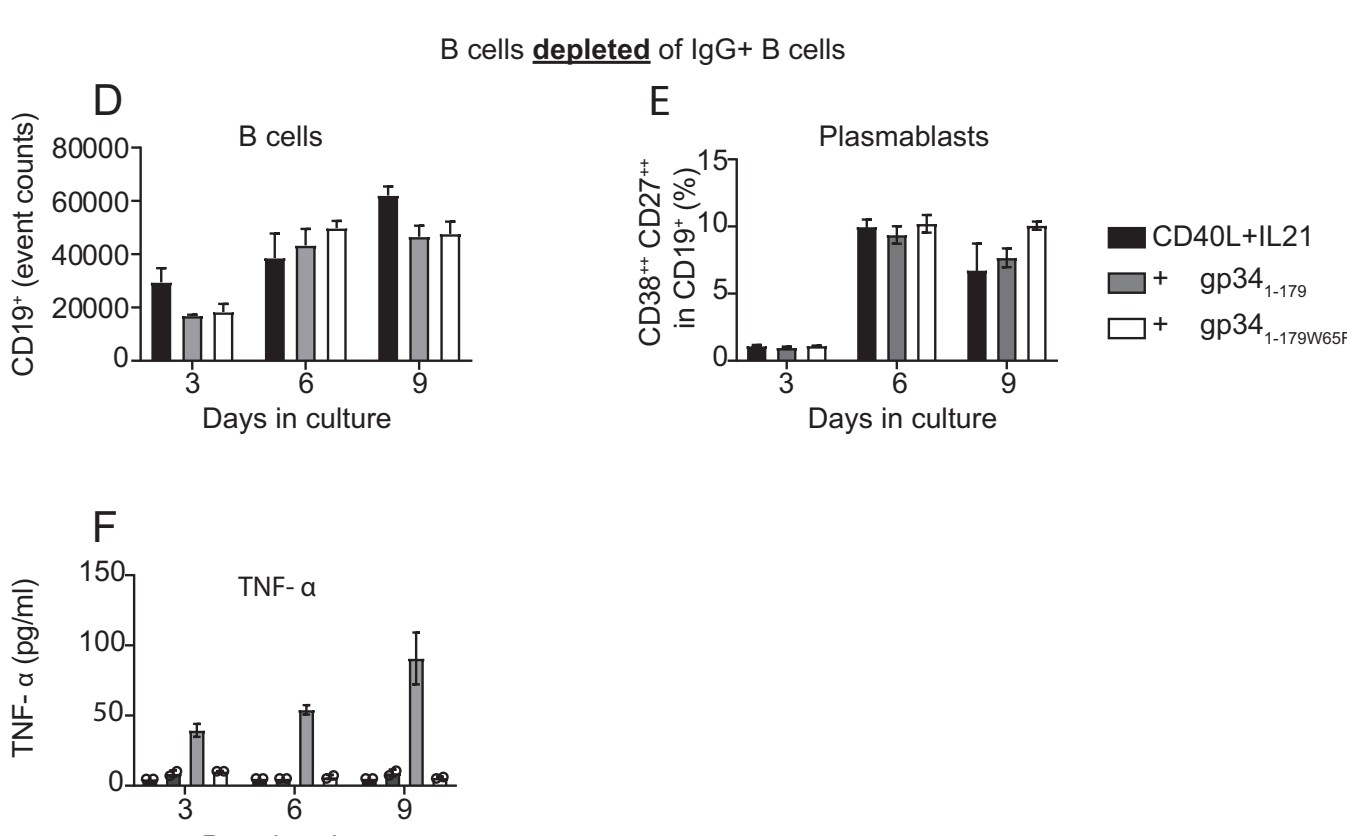

**Figure 5.   gp34₁₋₁₇₉ blocks proliferation, plasmablast differentiation, and subsequent immunoglobulin production by B cells.**

Isolated CD19⁺ cells were stimulated in vitro with CD40L and IL-21 in the presence or absence of gp34$_{1-179}$, gp34$_{1-179W65F}$, or α-BCR and analyzed by flow cytometry at the indicated time points for (**A**) counts of total B cells and (**B**) percentage of CD38$^{++}$ CD27$^{++}$ plasmablasts of total B cells. Error bars indicate the standard error of the mean of six independent experiments from different donors. Analysis was done with a multiple $t$ test for CD19 counts (**$P = 0.0044$, ***$P < 0.0005$) and a Wilcoxon test for percentage of CD38$^{++}$ CD27$^{++}$ plasmablasts (*$P = 0.0319$, **$P = 0.0023$, ***$P = 0.0008$, ****$P = < 0.0001$). (**C**) The concentration of IgG, IgM, and IgA in the supernatants of three ($n = 3$) in vitro cultures was measured by ELISA. Differences were analyzed by one-way ANOVA (*$P = 0.047$, **$P = 0.029$), and error bars represent the standard deviation of the mean. (**D**, **E**) The B-cell population was depleted of IgG⁺ B cells and stimulated with either CD40L and IL21 alone, or with the addition of gp34$_{1-179}$ or gp34$_{1-179W65F}$ for 3, 6, and 9 days. The bars show the quantification of total B-cell counts and percentage of plasmablasts formed in the IgG⁺-depleted B-cell pool. Data are shown as the mean ± standard deviation of the mean of two independent experiments, each in triplicate. (**F**) The TNF-α concentration was determined in the supernatant from the in vitro cultures using cytokine-based flow cytometry. Data are represented as the mean ± standard error of the mean of three independent experiments from different donors, each in duplicates.

and hence suppress humoral immunity. Weaker responses to certain vaccines in healthy HCMV-infected individuals might be associated with such a mechanism (Frasca and Blomberg, 2016; Mukhiya et al, 2024; van den Berg et al, 2019).

Of the limited number of screened cytokines, TNF-α production was increased in these gp34$_{1-179}$-stimulated B cell cultures. B cell-secreted cytokines, such as TNF-α, can be potent against infectious agents such as *Toxoplasma gondii, Heligomosomoides polygyrus*, or *Pneumocystis carinii* by enhancing expansion and differentiation of naive T cells and memory Th1 and Th2 cells (Lund et al, 2006; Menard et al, 2007; Wojciechowski et al, 2009), but exerts also direct antiviral effects on CMV replication (Lucin et al, 1994). Pre-incubation of B cells with TNF-α before LPS stimulation decreases the effector response in B cells, while adding anti-TNF-α antibodies in these cultures restores their function (Frasca et al, 2012) (Frasca et al, 2012). Remarkably, during primary HCMV infection, TNF-α-producing activated CD27⁺CD20⁺CD21$^{low}$ MBCs were observed among pregnant women (Dauby et al, 2014). Although causality between the increased TNF-α levels and the parallel dysregulation in non-IgG⁺ B cells was lacking (Appendix Fig. S5), prior reports of aberrant TNF-α secretion by MBCs in patient serum correlated with negative B cell responses (Frasca et al, 2014) and support our speculation that HCMV, via gp34, may modulate non-IgG B cells in a paracrine manner to attenuate an effective humoral immunity. Hence, the mechanism responsible for the inhibitory crosstalk between the IgG⁺ B cells and surrounding non-IgG⁺ B cells needs to be further deciphered.

HCMV replicates in multiple cell types and tissues, thereby disseminating virions and antigenic material on many paths and various aggregation forms. HCMV particles transport gp34 when excreted in body fluids such as blood (Halwachs-Baumann et al, 2002), saliva (Correia-Silva et al, 2007; Miller et al, 2006), semen (Bezold et al, 2007; Bresson et al, 2003), tears (Cox et al, 1975), urine (Cox et al, 1975; Cui et al, 2017; Daiminger et al, 1994) as well as the breastmilk (Meier et al, 2005; Schleiss, 2006; Yasuda et al, 2003) and amniotic fluid of pregnant mothers (Davis et al, 2017; Hemmings and Guilbert, 2002). In addition to virions, gp34 derived from infected tissue can find access as infected cells or extracellular vesicles (Streck et al, 2020; Zicari et al, 2018) to MBC inside and outside of germinal centers (Mesin et al, 2020; Victora and Nussenzweig, 2012), e.g., when transported to secondary lymphoid organs. Lastly, soluble antigens less than 70 kDa rapidly diffuse into lymph nodes through the afferent lymph vessel and thus encounter the B cells in soluble form (Batista and Harwood, 2009; Katakai et al, 2004; Szakal et al, 1983). Although gp34$_{1-179}$ was used as a model for membrane-resident gp34 in this study, it could also

occur in a soluble form, as several other *RL11* family members do (Engel et al, 2011; Pérez-Carmona et al, 2018; Windheim et al, 2013). In this way, soluble gp34 could readily reach and manipulate MBCs in the draining lymph node as well as MBCs present in HCMV-infected tissues.

Fully activated B cells express co-stimulatory molecules on their cell surface, which allow the B cell to directly interact with T cells. Mouse and human CMV are known to actively downregulate co-stimulating receptors like CD86, MHC molecules, CD80, and ICOS-L from infected cells to evade T-cell responses, thereby targeting humoral immunity (Angulo et al, 2021; Loewendorf et al, 2004; Mintern et al, 2006). In contrast to such cis-acting effects exerted on infected cells, gp34 present on virions or infected cells may represent the first example of an immune evasion that affects T-cell immunity via manipulating MBC in trans.

In summary, we report on a new strategy by which HCMV exploits MBCs to impair effective humoral responses using the Fcγ-binding gp34. The delineated novel function of gp34 provides new insights into viral immune evasion, demonstrating for the first time a viral protein modulating the B cell compartment by the exploitation of BCR signaling and initiating a multi-step process of B cell-B cell interactions. This elaborate strategy results in a secondary attenuation of humoral immunity and reactivation of recall responses to all antigens beyond HCMV. Our study warrants a more detailed molecular assessment of the state of gp34-modulated B cells, especially in immunodeficient patients experiencing virus reactivation. The co-expression of three additional HCMV glycoproteins that share the Fcγ-binding function with gp34 may indicate another level at which the virus could intensify its MBC-manipulating functions further.

## Methods

**Reagents and tools table**

| Reagent/resource | Reference or source | Identifier or catalog number |
| --- | --- | --- |
| **Viruses** | | |
| HCMV HB5 | Institute of Virology, University of Freiburg (Dr. Katja Hoffmann) | N/A |
| **Cell lines** | | |
| RPMI8866 | ECACC | 95041316 |
| Daudi | ATCC | CCL-213 |

| Reagent/resource | Reference or source | Identifier or catalog number |
|---|---|---|
| HEK293T | ATCC | CRL-11269 |
| MRC5 | ATCC | CCL-171 |
| BW:5147 | ATCC | TIB-47 |
| BW:5147-hCD16ζ[167] | Corrales-Aguilar et al, 2013 | N/A |
| SKOV3 | ATCC | HTB-77 |
| African green monkey CV-1 | ATCC | CCL-70 |
| EBV-transformed B cells | This study | N/A |
| **Oligonucleotides/sequence-based reagents** | | |
| Primer | Eurofins genomics | N/A |
| pGene/ V5-His B | Invitrogen | N/A |
| pIRES2-EGFP | Clontech | N/A |
| pSG5-Env_del1 | Institut für Virologie, Universität Düsseldorf (Dr. Corinna Asang) | N/A |
| pTRL11ΔFlag | Institut für Virologie, Universität Düsseldorf (Dr. Albert Zimmermann) | N/A |
| **Antibodies** | | |
| Rabbit anti-His6 | Bethyl Laboratories Inc | A190-114A |
| HRP anti-His6 | Rockland Immunochemicals | 200-303-382 |
| Human F(ab')$_2$ fragment | Rockland Immunochemicals | 009-010 |
| Anti-gp34 (mtrp 04) | In-house preparation | N/A |
| Cytotect | Biotest | N/A |
| Anti-DOK3 | Abcam | ab1357 |
| Rabbit IgG-Fc | Rockland Immunochemicals | 011-0103 |
| Goat anti-human IgG (H + L) F(ab')$_2$ | Jackson ImmunoResearch | 109-006-003 |
| Goat anti-human IgG (H) F(ab')$_2$ | Jackson ImmunoResearch | 109-006-008 |
| Goat anti-human Ig F(ab')$_2$ | Southern Biotech | 2012-01 |
| Anti-IgGAM | Jackson ImmunoResearch | AB_2337548 |
| Anti-IgG (Fab) | Jackson ImmunoResearch | AB_2337555 |
| Biotinylated anti-IgG F(ab')$_2$ | Bio-Rad AbD serotec | AB_10698339 |
| Anti-IgG (H + L) | Jackson ImmunoResearch | AB_2337532 |
| AP goat anti-human-IgG | Jackson ImmunoResearch | AB_2337601 |
| AP goat anti-human-IgA | Jackson ImmunoResearch | AB_2337602 |
| AP goat anti-human-IgM | Jackson ImmunoResearch | AB_2337603 |
| Anti-CD3 (UCHT1)-FITC | Biolegend | AB_2564148 |
| Anti-CD16 (3G8)-BV421 | Biolegend | AB_10898112 |
| Anti-IgA | Southern Biotech | AB_2561578 |
| Anti-IgG-APC | Biolegend | AB_2565790 |

| Reagent/resource | Reference or source | Identifier or catalog number |
|---|---|---|
| Anti-IgD- FITC | Southern Biotech | AB_2795624 |
| Anti -IgD- PE | Southern Biotech | AB_2795630 |
| IgM (MHM-88)-FITC | Biolegend | AB_2493009 |
| Anti-IgG (IS11-3B2.2.3) | Miltenyi Biotec | 130-093-197 |
| Anti-IgA (IS11-8E10) | Miltenyi Biotec | 130-093-073 |
| Anti-IgE (MB10-5C4) | Miltenyi Biotec | 130-093-127 |
| Anti-IgM (REA740) | Miltenyi Biotec | 130-124-308 |
| Anti-CD19 (HIB19)-BV510 | Biolegend | AB_2561688 |
| CD19-PerCP | Biolegend | AB_893272 |
| Anti -CD80 (2D10)- PE_Cy7 | Biolegend | AB_2076148 |
| Anti -CD86 (IT2.2)- BV510 | Biolegend | AB_2562064 |
| Anti-CD86 PerCP.Cy5.5 | Biolegend | AB_1575068 |
| Anti -CD69 (FN50)-FITC | Biolegend | AB_314839 |
| Anti -CD38 (HB-7)-PE-Cy7 | Biolegend | AB_2561904 |
| Anti -CD27 (O323)-BV421 | BD Biosciences | 665416 |
| Anti -HLA-DR(G46-6)- APC | BD Biosciences | 560896 |
| HRP anti-His | R&D Systems | MAB050H |
| Anti-pAKT (S473) | Cell Signaling Technology | 9271 |
| Anti-pAKT (T308) | Cell Signaling Technology | 9275 |
| Anti-pS6 (S240/244) (D68F8)- PE | Cell Signaling Technology | 14236 |
| Anti-pSYK (Y348) (I120-722)- PE | BD Biosciences | AB_647247 |
| Anti-pPLC-γ2 (pY759)- APC | Miltenyi Biotec | 130-104-971 |
| pLyn (Tyr 397) | Cell Signaling Technology | 94361 |
| pPDK1(S241) (C49H2) | Cell Signaling Technology | 3438 |
| Pan AKT- APC | Miltenyi Biotec | 130-137-400 |
| Lyn (C13F9)- PerCP | Cell Signaling Technology | 2796 T |
| SYK (B435666)- PE | BD Biosciences | 558529 |
| PLC-γ2 (K86-1161)-AF488 | BD Biosciences | AB_1645307 |
| S6 (54D2)- AF700 | Cell Signaling Technology | 56196 |
| Anti-Rabbit IgG- PE | Santa Cruz | sc-3753 |
| Streptavidin (APC) | BD Pharmingen | AB_10050396 |
| Anti-cleaved caspase 3 | Cell Signaling Technology | 9602 |
| Infliximab | University Hospital Pharmacy Freiburg | N/A |

| Reagent/resource | Reference or source | Identifier or catalog number |
|---|---|---|
| Rituximab | University Hospital Pharmacy Freiburg. | N/A |
| Human F(ab')₂ isotype control | Rockland | 009-0104 |
| **Viability dyes** | | |
| Zombie (NIR) | Biolegend | 423106 |
| AnnexinV-FITC | ThermoFischer scientific | BMS500FI |
| DAPI | ThermoFischer scientific | D3571 |
| **Chemicals/proteins** | | |
| Wortmannin | S2758 | Selleckchem |
| Staurosporine | Sigma | 62996-74-1 |
| CD40L | In-house preparation (https://doi.org/10.1073/pnas.0903543106) | N/A |
| IL21 | In-house preparation | N/A |
| **Software** | | |
| Adobe illustrator | Adobe Inc | N/A |
| Flowjo V10 | Tree Star Inc | N/A |
| Prism 10 | GraphPad | N/A |
| Geneious | Biomatters | N/A |

## Ethics

Written informed consent was obtained from healthy blood donors, and the study was conducted according to federal guidelines, local ethics committee regulations (Albert-Ludwigs-University, Freiburg, approval number 474/18), Germany) and the Declaration of Helsinki (1975).

Animal experiments were approved by the Ethics Committee of the University of Rijeka and the National Ethics Committee for the Protection of Animals Used for Scientific Purposes (Ministry of Agriculture, approval number HR-POK-004). Twelve female wild-type strain BALB/cJ mice (The Jackson Laboratory, #000651) were bred at the Center for Breeding and Engineering of Laboratory Mice, Faculty of Medicine, University of Rijeka (LAMRI). All procedures were carried out in accordance with the International Guiding Principles for Biomedical Research Involving Animals. In addition to the WMA Declaration of Helsinki, all experiments also conformed to the principles set out in the Department of Health and Human Services Belmont Report.

## Cell isolation and cell culture

MRC5 cells (human lung fibroblasts, ATCC:CCL-171), HEK293T (human embryonic kidney, ATCC: CRL-11269), SKOV-3 (human ovarian adenocarcinoma, ATCC: HTB-77), and CV-1 (Green monkey kidney fibroblasts, ATCC: CCL-70) cells were cultured in Dulbecco's modified Eagle's medium (DMEM) supplemented with 100 U/mL of penicillin, 100 μg/mL streptomycin, 50 μM 2-Mercaptoethanol and 10% heat-inactivated fetal calf serum (FCS) (Corrales-Aguilar et al, 2014; Kolb et al, 2021).

Peripheral blood mononuclear cells (PBMCs) were isolated from buffy coats or whole blood by density gradient centrifugation using Ficoll. B cells were isolated with the EasySep™ Human B-cell isolation Kit (Stemcell Technologies) following the manufacturer's instructions. IgG+ B cells were depleted with the EasySep™ Human IgG+ memory B-cell isolation kit (Stemcell Technologies), and untouched naive B cells were used for the respective experiments. Cells were cultivated in modified Dulbecco's Medium (IMDM) completed with 10% FCS.

To induce plasmablast formation, $3.0 \times 10^5$ B cells isolated by negative selection were plated in U-shaped 96-well plates. The cells were then stimulated with CD40L and IL21 in IMDM completed with 10% FCS, 1 μg/mL insulin, 2.5 μg/mL apo-transferrin, 1% non-essential amino acids, 2 mmol/L glutamine, and 1 μg/mL reduced glutathione. The cultures were maintained for 6–9 days in the presence or absence of 10 μg/mL each of α-IgG, recombinant soluble gp34$_{1-179}$, and gp34$_{1-179W65F}$.

Epstein–Barr virus-transformed cell lines were grown in IMDM supplemented with 10% FCS. B cell lines (Daudi and RPMI 8866, ATCC: CCL-213 and ECACC: 95041316, respectively) and BW5147-hCD16-ζ reporter cells (mouse thymoma transfected with human CD16-ζ) were cultured in RPMI-1640 medium supplemented with 1% non-essential amino acids and 10% heat-inactivated FCS.

## Virus stocks and infections

The bacterial artificial chromosome (BAC) derived HCMV strain HB5 (Borst et al, 1999) was used as the parental virus from which HB5 TKO (virus lacking *RL11*, *RL12*, and *UL119-118*) was generated.

Virus stocks were grown by infecting MRC5 cells with a multiplicity of infection (moi) of 0.2. Supernatants were collected after 7 days when cells showed maximum cytopathic effect. To purify the viruses, the cell-free supernatants were ultracentrifuged at $20,000 \times g$ for 3 h at 10 °C. Pellets from this first ultracentrifugation were dounced and resuspended in 1× PBS before layered onto a cold 15% sorbitol cushion. A second ultracentrifugation was performed at $100,000 \times g$ for 1.5 h at 10 °C. The virus pellet was dissolved in 1× PBS, aliquoted, and frozen at −80 °C for later use. Virus titers were determined by titrating purified viruses onto MRC5 cells and detecting the immediate early antigen 1(IEA1) with a monoclonal antibody in fluorescence microscopy.

## Generation of gp34 monoclonal antibody

BALB/c mice were injected with 50 μg of the immunogen (gp34$_{1-179W65F}$) in complete Freund's adjuvant (CFA), followed by 50 μg of the same immunogen in incomplete Freund´s adjuvant (IFA) 14 days post first immunization. Next, the sera of the immunized mice were tested for anti-gp34$_{1-179W65F}$ antibody titers using ELISA. The mice with the highest titers were further boosted with 50 μg of gp34$_{1-179W65F}$. After three days, the spleen of the immunized mice was harvested, lysed off red blood cells, and the splenocytes were fused with SP2/0 cell lines. The potential hybridoma cells were seeded in 20% RPMI 1640 medium containing hypoxanthine, aminopterin, and thymidine (HAT) for selection of stable hybridoma cell lines. The top six hybridoma cell lines specific for gp34$_{1-179W65F}$ protein were sub-cloned, and the supernatants were collected. To eliminate cross-reactive antibodies, the supernatants were tested against a non-Fcγ protein, ICOS-

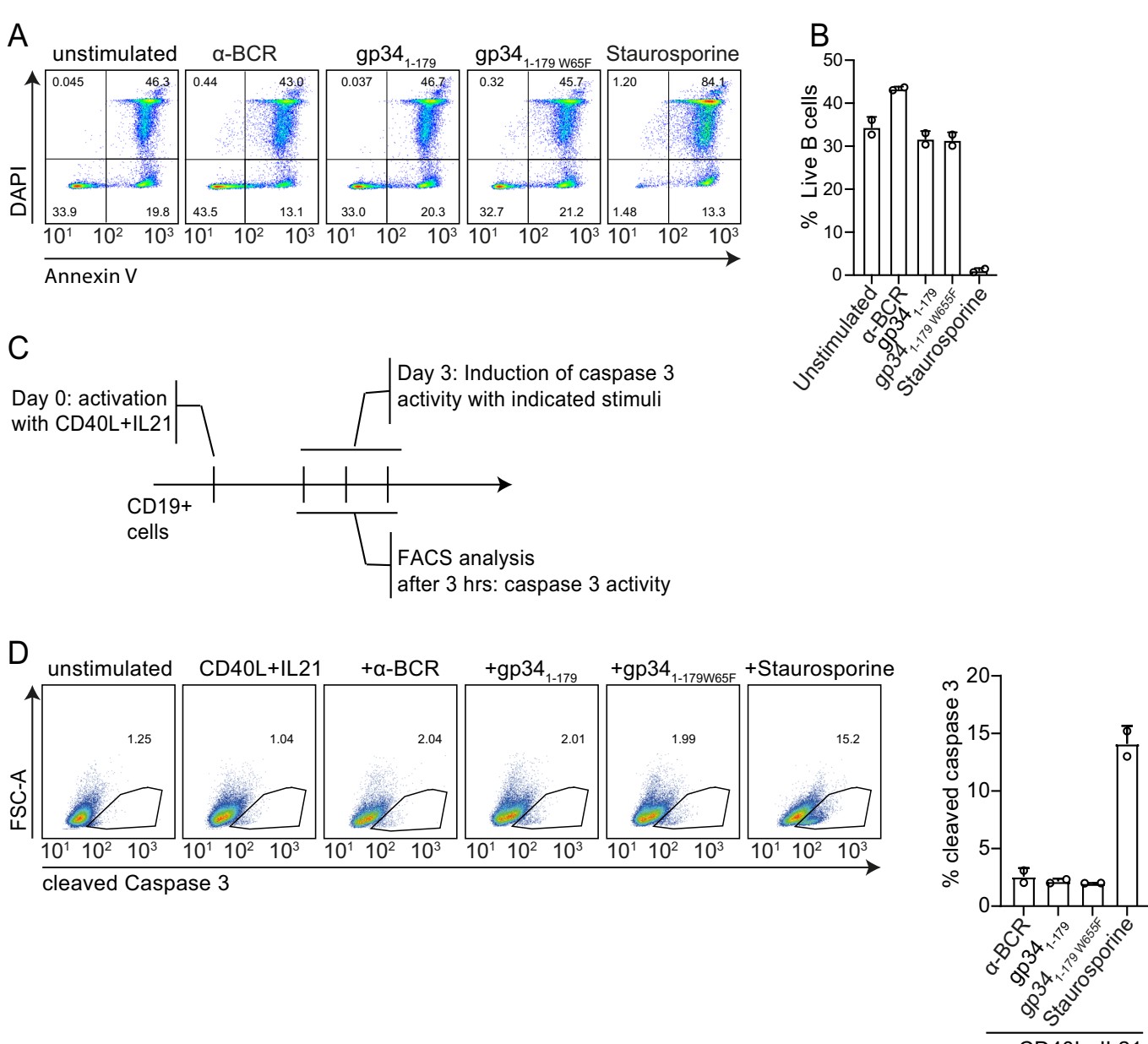

**Figure 6. gp34$_{1-179}$ does not induce cell death in B cells.**

(A) Apoptosis in response to α-BCR, gp34$_{1-179}$, gp34$_{1-179W65F}$, or staurosporine (1 μM) for 24 h was examined in resting B cells by AnnexinV and DAPI staining by flow cytometry. (B) The percentage of living cells after the induction of apoptosis with α-BCR, gp34$_{1-179}$, gp34$_{1-179W65F}$, or staurosporine in resting B cells after 24 h stimulation was quantified by flow cytometry. Living cells are defined as DAPI⁻/AnnexinV⁻ cells. Data are two biological replicates. (C) Experimental setup to analyze cleaved caspase-3 in CD40L/IL-21-activated B cells by flow cytometry. (D) Representative flow cytometry analysis showing cleaved caspase-3 in 3-day in vitro cultured CD40L/IL-21 activated B cells after 3 h of α-BCR, gp34$_{1-179}$, gp34$_{1-179W65F}$, or staurosporine treatment. Bars represent the percentages of cleaved caspase-3-positive cells ($n = 2$). All error bars are the mean ± standard deviation.

L$_{1-268}$. The monoclonal antibodies used in this manuscript are gp34 mtrp.04 and gp34 mtrp.06

## Generation of infliximab F(ab')$_2$ fragments

For the neutralization of soluble TNF-α in the cell cultures, infliximab F(ab')$_2$ fragments were generated using the Pierce F(ab')$_2$ preparation kit (44988, ThermoFisher). Human F(ab')$_2$

fragments (009-0104, Rockland) were used as a control. A concentration of 1 μg/mL was used for both conditions.

## Visualization of gp34 on HCMV virions using transmission electron microscopy (TEM)

Purified HB5 and HB5 TKO virions were adhered onto glow-discharged Formvar-coated TEM grids for 5 min, washed, and

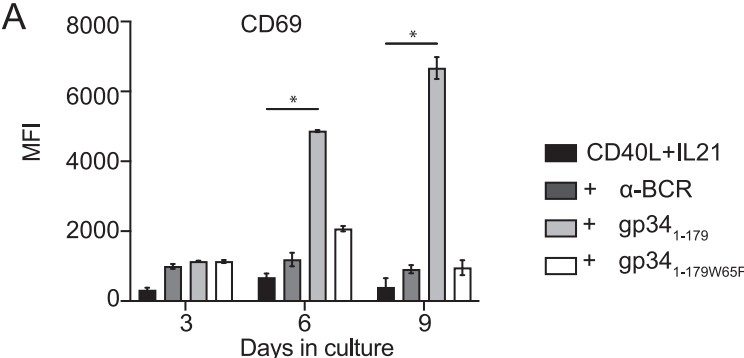

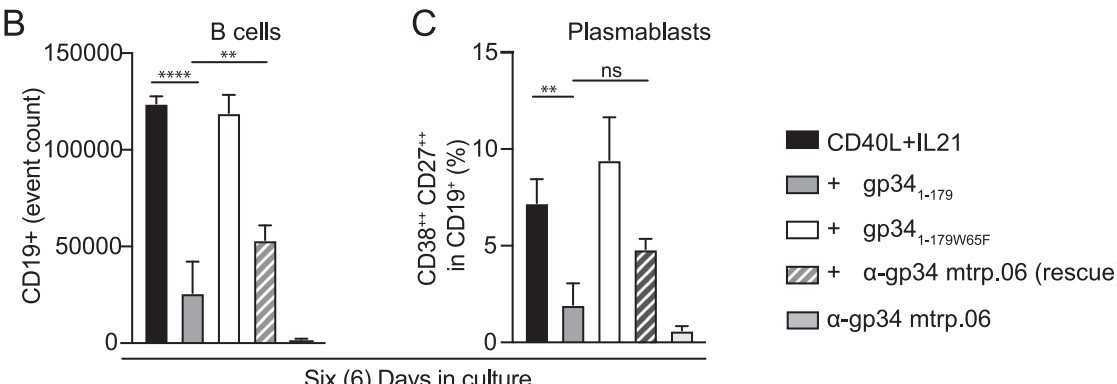

**Figure 7.  B cells interacting with gp34$_{1-179}$ are transiently hyporesponsive.**

(A) Isolated B cells were stimulated in vitro with CD40L and IL-21 in the presence or absence of gp34$_{1-179}$, gp34$_{1-179W65F}$, or α-BCR and analyzed by flow cytometry at the indicated time points for the expression of surface CD69. Error bars indicate the median ± standard deviation of duplicates. Statistical significance was calculated using a Kruskal–Wallis test (*P = 0.0429). Total B cells were stimulated in vitro with CD40L and IL-21 in the presence or absence of gp34$_{1-179}$ or gp34$_{1-179W65F}$ for 3 days. gp34$_{1-179}$-treated cells were then supplemented with α-gp34 mtrp.06 monoclonal antibody and CD40L and IL-21 for the remaining 3 days. Counts of total B cells (B) and percentage of CD38$^{++}$CD27$^{++}$ plasmablasts within the live B cell gate (C) were analyzed by flow cytometry. Error bars indicate the standard error of the mean of three independent experiments from different donors. Analysis was done with the Kruskal–Wallis test. For total B cells, **P = 0.0061, ****P < 0.0001. For the percentage of CD38$^{++}$CD27$^{++}$ plasmablasts, **P = 0.099.

blocked for 10 min with 0.1% BSA in 1× PBS. The grids were then incubated with gp34 mtrp.04 monoclonal antibody at RT for 15 min followed by detection with α-mouse antibody coupled to 10-nm gold particles (810.022, Aurion, Netherlands) for 15 min at RT. Post washing, the grids were stained with 0.5% uranyl acetate in water, subjected to EM, and imaged with a Jeol1400 at an acceleration voltage of 120 kV.

## Cloning and purification of recombinant soluble proteins

gp34 truncation and gp34 flag-tagged mutants (Appendix Fig. S1A,B) were cloned as described (Reinhard, 2010). V5/His6-tagged soluble recombinant proteins (Appendix Fig. S1C) were constructed as follows:

gp34$_{1-179}$: The ectodomain of gp34 was PCR amplified from cDNA of HCMV AD169 BAC-infected MRC5 cells using the *RL11-*(XhoI)-for and *RL11-*(BamHI)-rev primer pair.

gp34$_{1-179W65F}$: The primer pair *IRL11-*(XhoI)-for and *IRL11-*(BamHI)-rev were used to amplify the ectodomain of gp34. With a gBlock (IDT) containing an amino acid substitution from

tryptophan to phenylalanine, the mutation was introduced into gp34$_{1-179}$ at position 65. Thus, creating gp34$_{1-179W65F}$, which has all Fcγ-mediated functions abrogated (Kolb et al, 2021).

ICOS-L$_{1-268}$: The N terminus of ICOS-L was amplified from cDNA isolated from T cells using primer pair Sol ICOS-L (XhoI)-for and Sol ICOS-L (BamHI)-rev.

The V5/His6 tags were inserted in-frame at the C terminus before the stop codon of each coding sequence using the reverse primers (RL11-(BamHI)-rev and ICOS-L$_{1-268}$ (BamHI)-rev). Each amplicon was gel-purified, digested with the restriction enzymes *XhoI* and *BamHI* and cloned into a pIRES-EGFP expression vector (Clontech, USA) between the CMV-IE promoter and the coding sequences.

All plasmids were sequenced with Eurofins Genomics.

For stable cell lines, the coding plasmids of gp34$_{1-179}$, gp34$_{1-179W65F}$, and ICOS-L$_{1-268}$ were sub-cloned into a pUCIP vector and transduced into HEK293T cells via lentiviruses under puromycin selection.

For transient transfection, plasmids were transfected into HEK293T cells using polyethylenimine (PEI, branched, Sigma-

Aldrich, Germany). 24 h post transfection, the medium was replaced with DMEM supplemented with 2% FCS. After 6–7 days, supernatants were collected, cleared of cell debris, and adjusted to 20 mM Imidazole concentration and passed over a His-Trap FF crude column (GE Healthcare, USA). Proteins were eluted in Imidazole/Phosphate buffer (500 mM Imidazole, 20 mM sodium phosphate, 500 mM NaCl) and dialyzed with 1× PBS. Protein amounts were adjusted with the Pierce bicinchoninic acid (BCA) protein kit (ThermoFischer Scientific), and western blot analysis was performed using α-Histidine antibody (Bethyl Laboratories Inc.).

Primers

RL11-(XhoI)-for: 5'-CGCGCTCGAGATGCAGACCTACAGCACCCC-3'

RL11-(BamHI)-rev: 5'-CGCGGGATCCTCAATGGTGATGGTGATGATGACCGGTACGCGTAGAATCGAGACCGAGGAGAGGGTTAGGGATAGGCTTACCGGACCACTGGCGTTT-3'

ICOS-L$_{1-268}$ (XhoI)-for: 5'-GCGCCTCGAGATGCGGCTGGGCAGTCCTGGACTG-3'

ICOS-L$_{1-268}$ (BamHI)-rev: 5'-GCGCGGATCCTCAATGGTGATGGTGATGATGACCGGTACGCGTAGAATCGAGACCGAGGAGAGGGTTAGGGATAGGCTTACCCGTGGCCGCGTT-3'.

## Radioactive labeling and immunoprecipitation

gp34 truncation variants and gp34 point mutants were expressed in CV-1 cells by recombinant vaccinia virus (rVACV) for 14 h as described (Atalay et al, 2002; Sprague et al, 2008). In brief, CV-1 cells were labeled for 1 h with $^{35}$S-Pro-Mix (GE Healthcare) before lysates were prepared and proteins precipitated with human Fcγ fragment (Rockland Immunochemicals, Inc., Limerick, PA 19468). The precipitated complex was split in half and deglycosylated with Endo-H overnight at 37 °C. The proteins were separated on a 10–13% gradient SDS–PAGE. Cell lysates were precipitated with anti-Flag-coupled agarose and used as the expression control. The lysates from the Flag-tagged proteins were separated on a 12% SDS–PAGE gel.

## Binding assay

An ELISA plate was coated with 4 µg/mL human Fcγ fragment (Rockland Immunochemicals, Inc.) diluted in sodium carbonate buffer (pH 9.6), incubated overnight at 4 °C, and then blocked with PBS/1% BSA solution. The plate was then incubated with increasing concentrations of purified recombinant histidine-tagged soluble gp34. Bound gp34 was detected using an anti-histidine antibody (Bethyl Laboratories Inc.), and the optimal density was measured at 460 nm with a spectrophotometer. The optimal concentration of gp34 suitable for binding to IgG$^+$ BCR cells was determined by coating an ELISA plate with an α-IgG antibody before serial dilutions of the histidine-tagged purified gp34 were added to the plate. Peroxidase-conjugated anti-His antibody was used to detect bound gp34 to the B cells.

## Fcγ receptor activation assay

Skov-3 cells expressing the Her2 antigen were cultured till 90% confluence. The Skov-3 target cells were then incubated with dilutions of recombinant proteins and Herceptin® in complete

DMEM for 30 min at 37 °C. The cells were washed and co-cultured with BW5147-hCD16 effector cells (E:T 10:1) for 16 h at 4 °C. IL-2 secreted as a result of activation of the effector cells was measured in ELISA. The IL2 ELISA was performed as described previously (Corrales-Aguilar et al, 2013).

## Pulldown assay and western blots

EBV-transformed B cell clones were harvested and lysed in 1× lysis buffer (20 mM Tris-HCl pH 7.5, 1 mM EDTA, 150 mM NaCl, and 1% Triton X-100) supplemented with protease inhibitors (1 mM Na$_3$VO4, 1 mM PMSF, and 2 µg/mL Leupeptin). In total, 10 µg/mL of the recombinant proteins were coupled to anti-Histidine resin and incubated with EBV-transformed B-cell lysates overnight at 4 °C. The resin was centrifuged, washed, and the precipitate deglycosylated with PNGase F at 37 °C. Immune-precipitated proteins were separated by 12% SDS–PAGE and transferred to nitrocellulose membranes (Protran, GE Healthcare Life Sciences). The membranes were blocked with 5% milk/TBST solution and incubated with the indicated antibodies. The membranes were developed with the SignalFire ECL reagent (Cell Signaling Technology) according to the manufacturer's protocol. β-actin (clone AC-74, Sigma Life Science) was used as a loading control.

## Determination of immunoglobulins

ELISA plates were coated with 1.2 µg/mL anti-human IgGAM (Jackson ImmunoResearch) in sodium bicarbonate buffer and incubated overnight at 4 °C. Supernatants from stimulated cells or protein standard (N protein standard SL, Siemens) were incubated on the plates at room temperature for 2 h. Bound Ig was detected with alkaline phosphatase-conjugated anti-human IgG (0.3 µg/mL), IgM (0.15 µg/mL), and IgA (0.15 µg/mL) (all from Jackson ImmunoResearch). Development was done with p-nitrophenyl phosphate (Sigma-Aldrich) in DEA buffer.

## Flow cytometry

### Surface staining

Phenotypic identification of PBMCs and B cells was done by flow cytometry using the following antibodies: CD19 BV510 (clone HIB19), IgD FITC (clone 2032-09), IgD PE and IgA (polyclonal, Southern Biotech), IgM FITC (MHM-88), CD38 PE-Cy7 (clone HB-7), CD27 BV421 (clone O323), IgG (APC), CD86 PerCP.Cy5.5, CD86 BV510 (IT2.2), CD80 PE-Cy7 (2D10), CD69 FITC, and CD19 PerCP, CD3 FITC (UCHT1), CD16 BV421 (3G8) (all from Biolegend), and HLA-DR APC (BD Biosciences). Dead cells were excluded from data analysis with 4, 6-diamidino-2-phenylindole (DAPI) staining or by Zombie NIR fixable Viability Kit (Biolegend).

### B-cell receptor stimulation

For the detection of phosphorylated proteins, PBMCs or isolated B cells were stimulated with goat anti-human IgG (H + L) F(ab')$_2$, goat anti-human IgG (H) F(ab')$_2$ (Jackson ImmunoResearch), goat anti-human Ig F(ab')$_2$ (Southern Biotech; all 10 µg/mL) or purified recombinant proteins including gp34$_{1-179}$ at an ideal concentration of 10 µg/mL for the indicated time points.

### Phosphoflow

B cells were fixed and permeabilized using the Cytofix/Perm buffer III (BD Biosciences) or the Foxp3/Transcription factor staining buffer (Invitrogen) in accordance with the manufacturer's protocol. Phosphorylated proteins were detected using the following antibodies: pLyn (Tyr 397), pPDK1 (S241) (clone C49H2), pAKT (S473), pAKT (T308), pS6 (S240/244) (all Cell Signaling Technology), pPLCγ2 (pY759) (Miltenyi Biotec), and pSyk (Y348) (BD Biosciences).

### Total protein expression levels

To determine total protein expression levels after $gp34_{1-179}$ stimulation, PBMCS were stimulated for 10 min with the indicated stimuli. Thereafter, cells were directly put on ice and washed with ice-cold PBS. Following staining for surface markers, cells were fixed, and intracellular staining was performed using the Cytofix/Cytoperm kit (BD Biosciences). Proteins were detected using the following antibodies: Lyn PerCP (clone C13F9, Cell Signaling Technology), SYK PE (clone B435666, Biolegend), PLCγ2 AF488 (clone K86-1161, BD Biosciences), pan AKT APC (Miltenyi Biotec), and S6 AF700 (clone 54D2, Cell Signaling Technologies).

### BCR internalization

IgG BCR internalization was analyzed by first labeling all IgG+ B cells with goat anti-human biotinylated IgG $(Fab)_2$ (Bio-Rad AbD serotec) at 4 °C for 1 h. Cells were then washed and bound $(Fab)_2$ cross-linked with either anti-IgG (H + L) (Jackson ImmunoResearch), recombinant soluble $gp34_{1-179}$, $gp34_{1-179W65F}$, or left untreated (Mock) for 30 min at 4 °C. The samples were moved to a 37 °C incubator for 1, 6, 12, 18, and 24 h. Any further BCR internalization was stopped by moving the samples to 4 °C after the indicated time points. The cells were washed with ice-cold PBS and stained for remaining IgG+ BCR using APC-conjugated Streptavidin antibody (BD Pharmingen) in flow cytometry.

### Cytokine production

Culture supernatants were analyzed for 13 different cytokines using the Biolegend LEGENDplex™ human B cell panel kit.

### Data analysis

Mean fluorescence intensity (MFI) of the indicated markers was determined among B cells and subpopulations. All cells and cytokine conjugated beads were acquired with a FACS Canto II (BD Bioscience) or a 5-laser Cytek Aurora flow cytometer (Cytek Biosciences), and analyzed with FlowJo (version 10.10.0, TreeStar Inc.).

## Cell viability assay

In total, $3.0 \times 10^5$ cells/mL resting B cells were seeded and induced for 24 h with 1 µM Staurosporine or 10 µg/mL each of α-IgG, recombinant soluble $gp34_{1-179}$, and $gp34_{1-179W65F}$ to initiate apoptosis. To identify apoptotic B cells, cells were stained with FITC-conjugated AnnexinV and DAPI according to the manufacturer's instructions. The percentages of live cells remaining after 24 h of treatment were plotted as AnnexinV⁻ and DAPI⁻ cells.

To determine caspase-3 activation, $3.0 \times 10^5$ cells/mL B cells were seeded and activated with CD40L and IL-21 for 70 h. The activated B cells were then stimulated with 10 µg/mL each of α-IgG, $gp34_{1-179}$, $gp34_{1-179W65F}$, and 1 µM Staurosporine for 3 h to induce cleavage of caspase-3. The cells were fixed, and the percentage of cleaved caspase-3+ cells was determined by flow cytometry.

## Statistical analyses

One-way ANOVA, two-way ANOVA, Student's $t$ test, Kruskal–Wallis test, and multiple $t$ tests were used to assess the differences between group means/median. Tukey test was used to correct for multiple $t$ tests comparisons. Exact $P$ values are considered statistically significant, and statistical details of experiments are indicated in the figure legends. All statistical analyses were performed using GraphPad Prism software version 8.0 or 10.0.

## Graphics

Schematic diagrams (Fig EV2) and the graphical abstract were created using BioRender.com.

## Data availability

The source data produced in this study are available in BioStudies: Accession Number S-BSST2276.

The source data of this paper are collected in the following database record: biostudies:S-SCDT-10_1038-S44321-026-00372-1.

---

**The paper explained**

**Problem**

Human Cytomegalovirus (HCMV) is a conditional pathogen widespread in human populations. It persists lifelong with periodic phases of latency and virus reactivation. B lymphocytes establish immune memory by producing IgG antibodies, which are either secreted or inserted into the plasma membrane, where they serve as a part of B-cell receptors (BCR). To counteract antiviral antibodies, HCMV expresses IgG-Fc-binding decoy receptors such as gp34, blocking IgG-mediated effector responses. Whether gp34 also affects IgG-bearing memory B cells was never investigated.

**Results**

In this study, we demonstrate that gp34 binds to IgG BCRs, which initiates BCR internalization and unconventional activation of BCR signaling pathways. Formation of plasmablasts from memory B cells and antibody secretion becomes blocked, while gp34 triggers the secretion of the cytokine TNF-α. This results in the induction of hyporesponsiveness in all B cells and impaired antibody production.

**Impact**

Our findings suggest that HCMV can control antibody production and secondary B cell responses by manipulating BCRs of IgG memory B cells via gp34. This identifies gp34 as a promising target for intervention in HCMV disease. Known IgG-Fc binding proteins from other viruses and microorganisms could also influence humoral immunity in a similar way as gp34, i.e., beyond their blockade of soluble IgG via an interaction with IgG-bearing memory B cells.

## Peer review information

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

## Acknowledgements

We thank Hermann Eibel for the critical reading of the manuscript. We show appreciation to Suzana Malic and Karmela Miklic for technical assistance in biochemical methods and monoclonal antibody production. We thank Stefan Tenzer for MS/MS analysis. This study was supported in part by the Excellence Initiative of the German Research Foundation (GSC-4, Spemann Graduate School), the Ministry for Science, Research and Arts of the State of Baden-Wuerttemberg, the Deutsche Forschungsgemeinschaft through HE2526/9-2, JO1634/1-1, and the SFB1160 (B02 to MR). MF was supported by the Düsseldorf Entrepreneur Foundation at the Heinrich-Heine-University. This study was also supported in part by the Croatian Science Foundation projects (grant IP-2020-02-6617 to TLR); the University of Rijeka under the project number uniri-biomed-18-233 to TLR; SJ grant "Strengthening the capacity of CerVirVac for research in virus immunology and vaccinology", KK.01.1.1.01.0006, awarded to the Scientific Centre of Excellence for Virus Immunology and Vaccines and co-financed by the European Regional Development Fund. MR was funded by the Deutsche Forschungsgemeinschaft (DFG, German Research Foundation), project number 468499998.

## Author contributions

**Precious Cramer**: Data curation; Formal analysis; Funding acquisition; Validation; Investigation; Visualization; Methodology; Writing—original draft; Project administration; Writing—review and editing. **Stefan FH Neys**: Data curation; Formal analysis; Validation; Investigation; Visualization; Methodology; Project administration; Writing—review and editing. **Manuela Fiedler**: Data curation; Formal analysis; Funding acquisition; Methodology; Writing—review and editing. **Raquel Lorenzetti**: Data curation; Formal analysis; Investigation; Methodology. **Henrike Reinhard**: Data curation; Formal analysis. **Iga Janowska**: Resources; Data curation; Formal analysis; Investigation; Methodology. **Julian Staniek**: Formal analysis; Writing—review and editing. **Ann-Katrin Kohl**: Resources. **Petra Hadlova**: Resources. **Magdalena Huber**: Data curation. **Bodo Plachter**: Resources; Formal analysis. **Clarissa Read**: Data curation; Methodology. **Valeria Falcone**: Resources. **Jens von Einem**: Formal analysis; Writing—review and editing. **Katja Hoffmann**: Resources; Data curation; Supervision; Methodology; Project administration. **Tihana Lenac Rovis**: Resources; Funding acquisition; Writing—review and editing. **Stipan Jonjic**: Resources; Funding acquisition; Writing—review and editing. **Philipp Kolb**: Data curation; Supervision; Visualization; Methodology; Project administration; Writing—review and editing. **Marta Rizzi**: Conceptualization; Resources; Supervision; Funding acquisition; Investigation; Writing—original draft; Project administration; Writing—review and editing. **Hartmut Hengel**: Conceptualization; Supervision; Funding acquisition; Investigation; Writing—original draft; Project administration; Writing—review and editing.

Source data underlying figure panels in this paper may have individual authorship assigned. Where available, figure panel/source data authorship is listed in the following database record: biostudies:S-SCDT-10_1038-S44321-026-00372-1.

## Funding

## Disclosure and competing interests statement

The authors declare no competing interests.

# Expanded View Figures

**Figure EV1.  Functional characterization of truncated gp34 variants and ICOS ligand.**

(**A**) The predicted structure of gp34 (Atalay et al, 2002) includes an N terminal signal sequence (S) of 23 amino acids (aa 1–23), a connecting Ig-like domain (aa 24–122) with 3 N-linked glycosylation sites (Y) and the putative intramolecular disulfide bridge (dashed bracket), a 22 aa transmembrane domain (TM; aa 183-204), and a C terminal cytoplasmic tail of 30 aa in length (aa 205–234). Truncation variants of gp34 were cloned and tested for Fcγ binding properties. The truncation variants of gp34 were expressed by rVACV for 14 h in CV-1 cells. $^{35}$S metabolic labeling of cells for 1 h was followed by precipitation of proteins from lysates with human IgG-Fc fragment. Half of the precipitate from each sample was deglycosylated with Endoglycosidase H overnight at 37 °C. Separation of proteins was performed by 10–13% gradient SDS–PAGE. (**B**) The tryptophan-to-phenylalanine substitution at position 65 of the gp34$_{1-179}$ sequence designated as gp34$_{1-179W65F}$ variant was used as a non-Fcγ binding control. Both variants were expressed in CV-1 cells for 14 h using rVACV. Metabolic labeling was followed by precipitation of the proteins from the lysates with human IgG-Fc fragment. Proteins were separated by 12% SDS–PAGE. Immunoprecipitation with α-FLAG-coupled agarose served as an expression control. (**C**) gp34$_{1-179}$, gp34$_{1-179W65F}$ and N terminal variant of ICOS ligand (ICOS-L$_{1-268}$) with V5 and His6 epitope tags were expressed in HEK293T cells. The supernatants were collected after 6 days and purified by affinity chromatography on the Äkta purification system. Maturation and glycosylation pattern of purified recombinant proteins was analyzed by digesting the proteins with EndoH and PNGase F overnight. Proteins were separated by 10% SDS–PAGE and detected with a peroxidase conjugated anti-His antibody. (**D**) Human Fcγ fragment was coated onto an ELISA plate and incubated with titrated amounts of purified gp34$_{1-179}$, gp34$_{1-179W65F}$ or ICOS-L$_{1-268}$. Peroxidase-conjugated anti-His antibody was used to detect bound proteins to the Fcγ fragment by spectrophotometry. (**E**) Skov3 cells expressing the Her2 antigen were incubated with Herceptin together with gp34$_{1-179}$, gp34$_{1-179W65F}$ or ICOS-L$_{1-268}$. Inhibition of Fcγ receptor activation was measured as mIL-2 production of BW5147-FcγRIII reporter cells. Mock condition represents the absence of reporter cells. The Bar graphs show two independent experiments performed in triplicates. Error bars =SD.

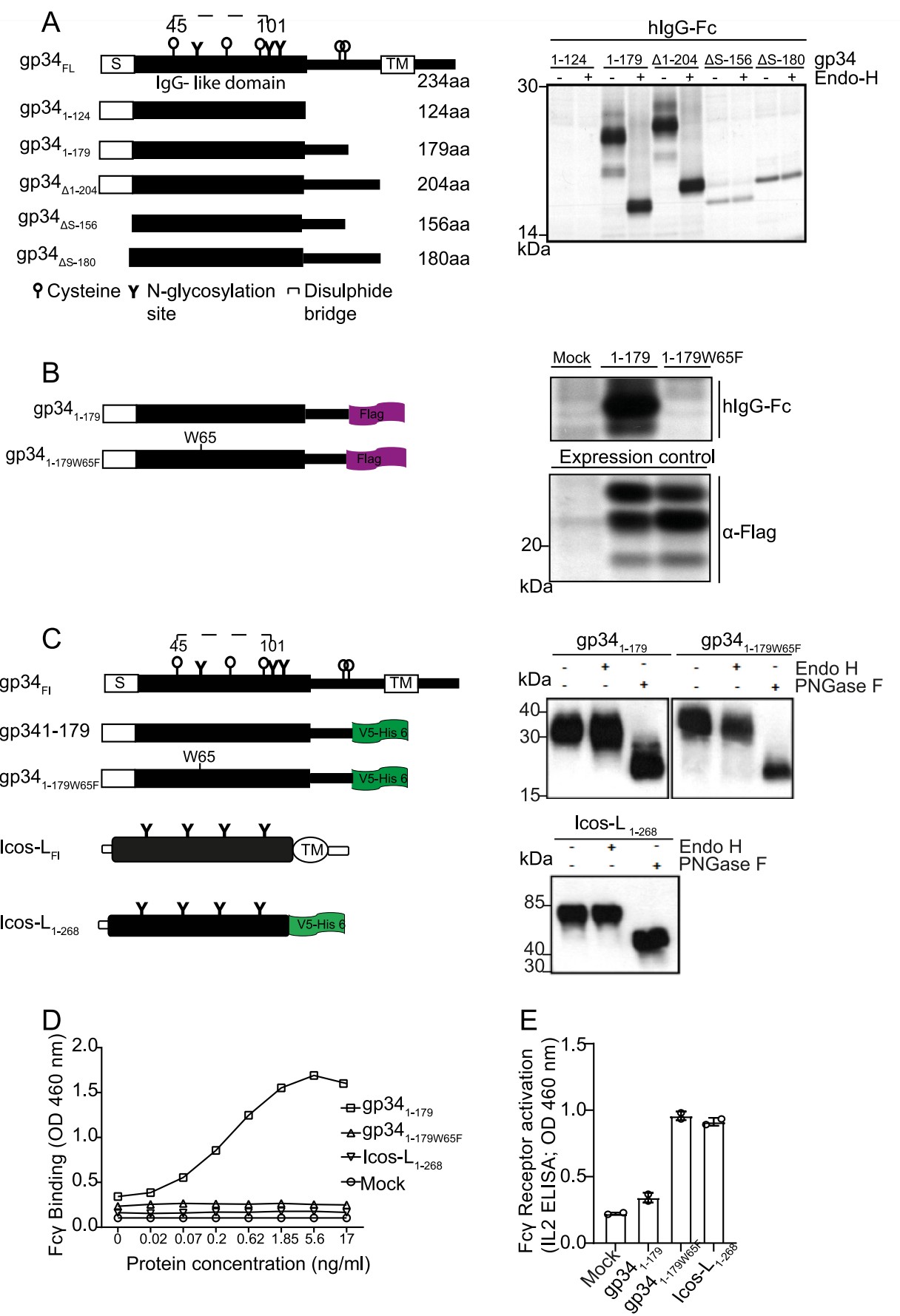

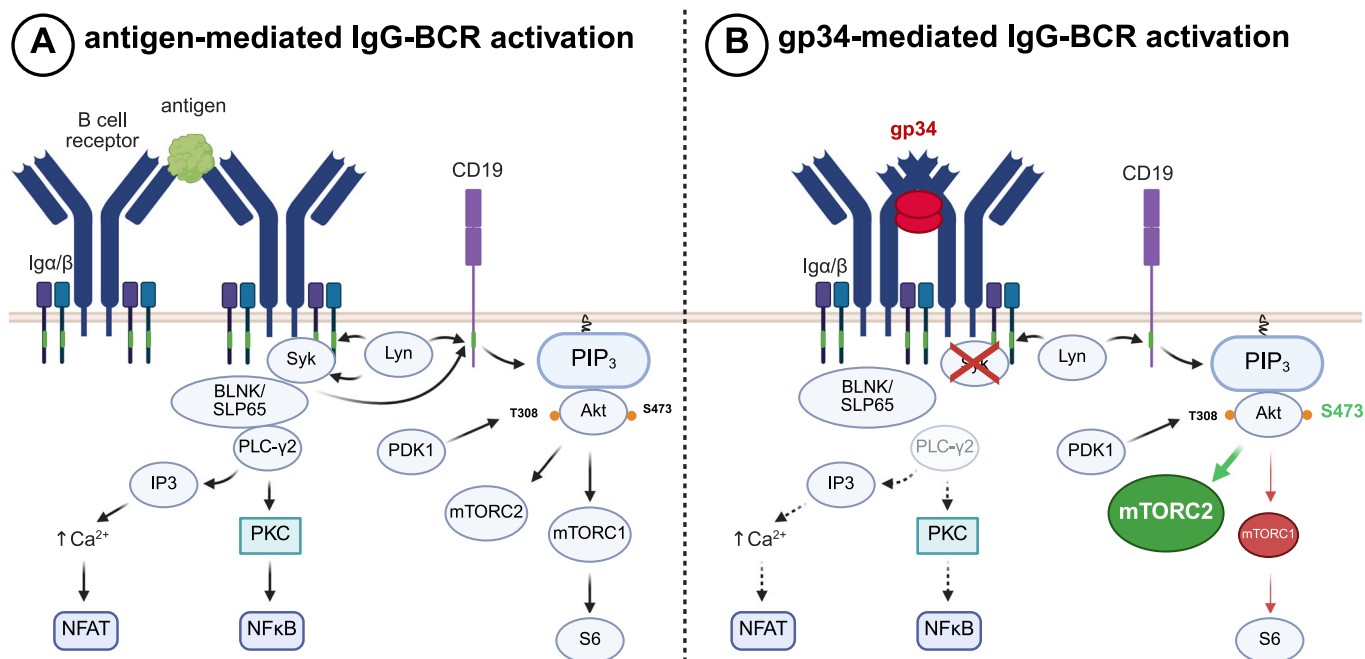

**Figure EV2.　IgG-BCR mediated signaling in memory B cells and its modification by gp34.**

Model for (**A**) conventional antigen induced and (**B**) gp34 modified signaling pathways regulating memory B cell activation. Boxes indicate signaling effectors, ovals indicate signaling cascades. Transparency indicates lack of phosphorylation. Dashed arrows indicate downstream effects which were not experimentally addressed in this study. Adapted from Laidlaw and Cyster, Nat Rev Immunol. 21, 209–220 (2021); and Puri et al, Int Rev Immunol. 32:397–427 (2013).

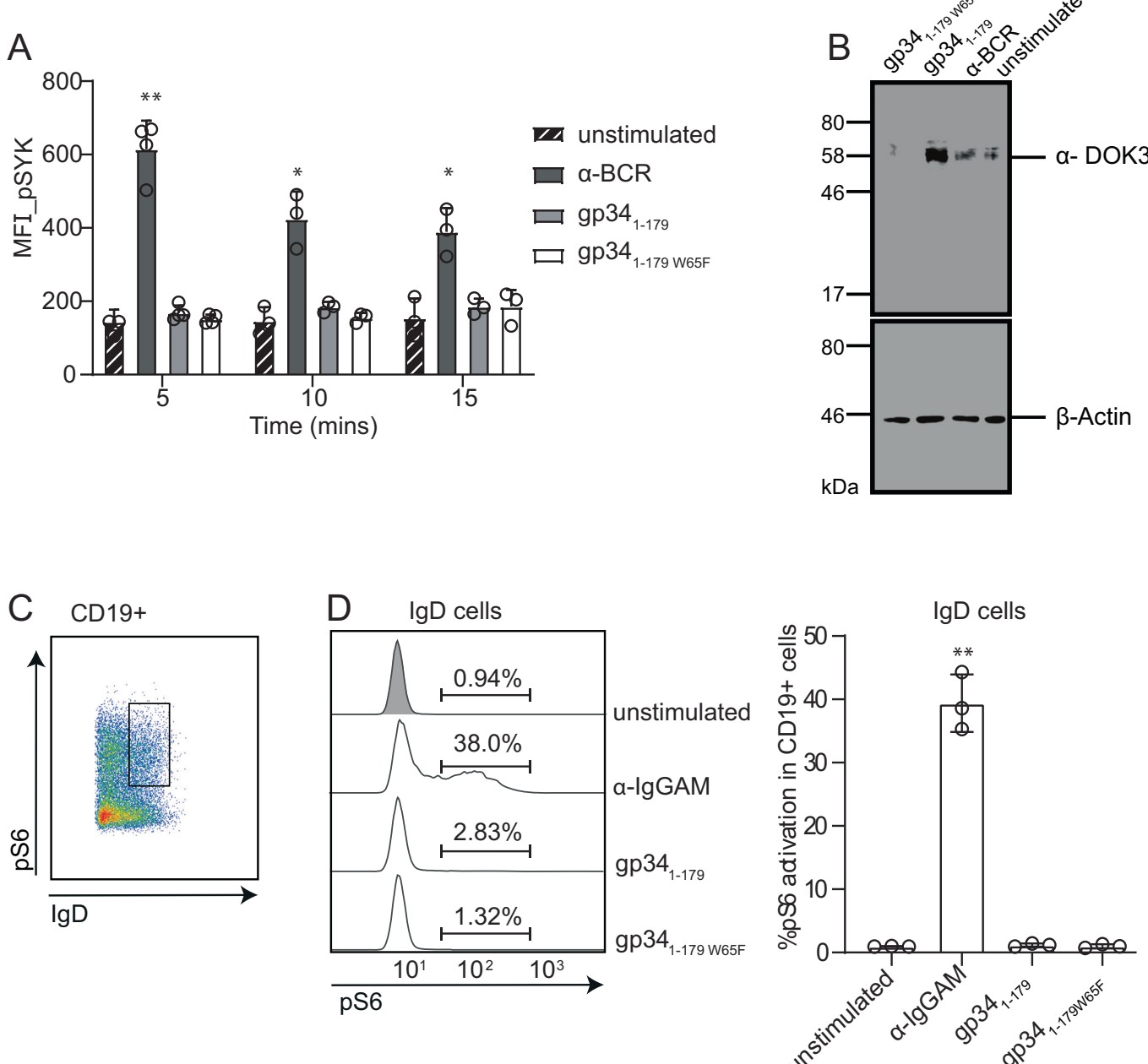

**Figure EV3.   gp34₁₋₁₇ blocks SYK activation, fails to induce S6 activation in IgD + B cells, yet upregulates DOK3 protein levels.**

(A) Levels of phosphorylated SYK induced upon BCR stimulation with α-IgG, gp34$_{1-179}$ and gp34$_{1-179W65F}$ over the indicated time points, was examined by flow cytometry and depicted as MFI of three biological replicates. A two-way ANOVA (with mixed-effects analysis) was performed for statistical analysis, *$P = 0.0416$, **$P = 0.0388$. Data are shown as the mean ± standard of the mean. (B) B cells were stimulated with α-IgG, gp34$_{1-179}$, and gp34$_{1-179W65F}$ for 60 min at 37 °C. The cells were lysed and separated on a 12% SDS–PAGE. An α-DOK3 antibody was used to detect DOK3 expressed by B cells in each condition. Results are representative of two independent experiments. β-actin served as loading control. (C) Gating strategy showing IgD⁺ B cells expressing phosphorylated S6. (D) The histograms show the percentages of pS6⁺ cells in IgD⁺ B cells upon stimulation with gp34$_{1-179}$, gp34$_{1-179W65F}$ or α-BCR. Bar graphs depict frequency of pS6⁺ cells in IgD⁺ B cells in three biological replicates. A non-parametric Kruskal–Wallis test was performed to analyze the differences in each group (**$P = 0.0012$). Data are shown as the mean ± standard of the mean.

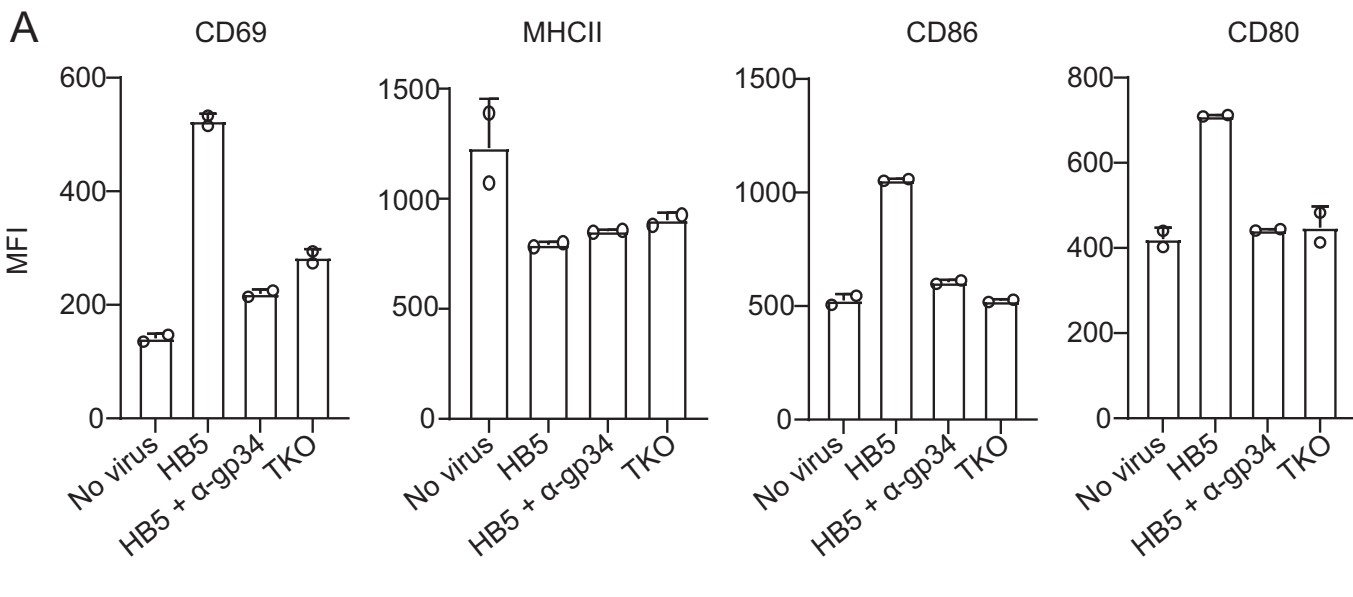

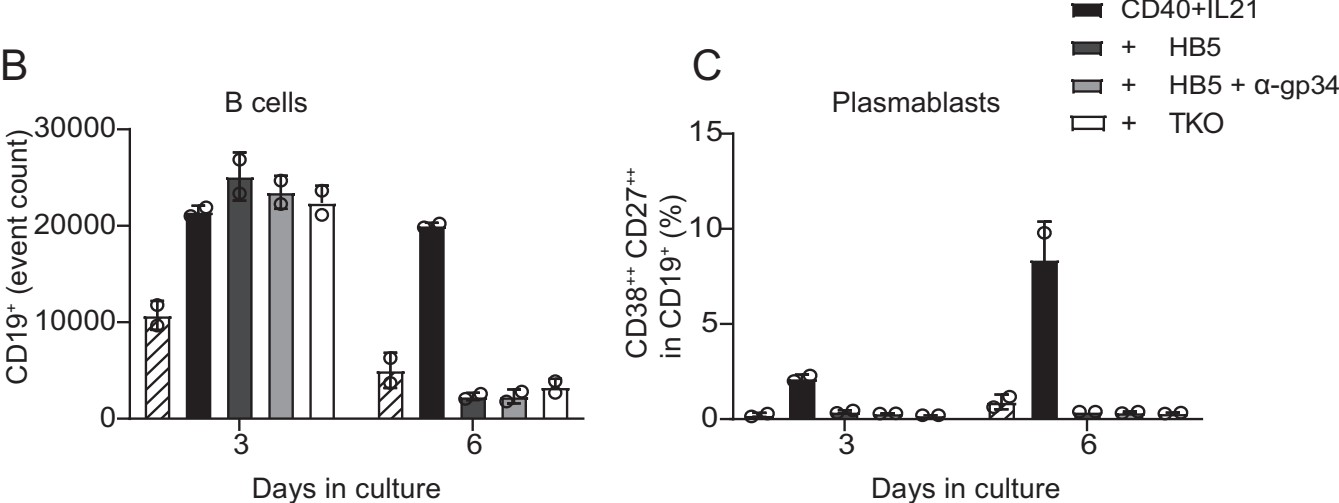

**Figure EV4. HCMV virions activate B cells in a gp34 dependent manner.**

(A) Primary B cells were incubated with HB5 or TKO virions in the presence or absence of α-gp34 mtrp.04 antibody. Upregulated CD69 and MHCII were examined after 24 h while CD86 and CD80 were assessed after 48 h by flow cytometry. (B, C) CD40L/IL21 stimulated primary B cells were incubated with HCMV HB5 virions and analyzed for CD19 B cell count and plasmablasts. All data are shown as the mean ± standard of the mean of two biological replicates.

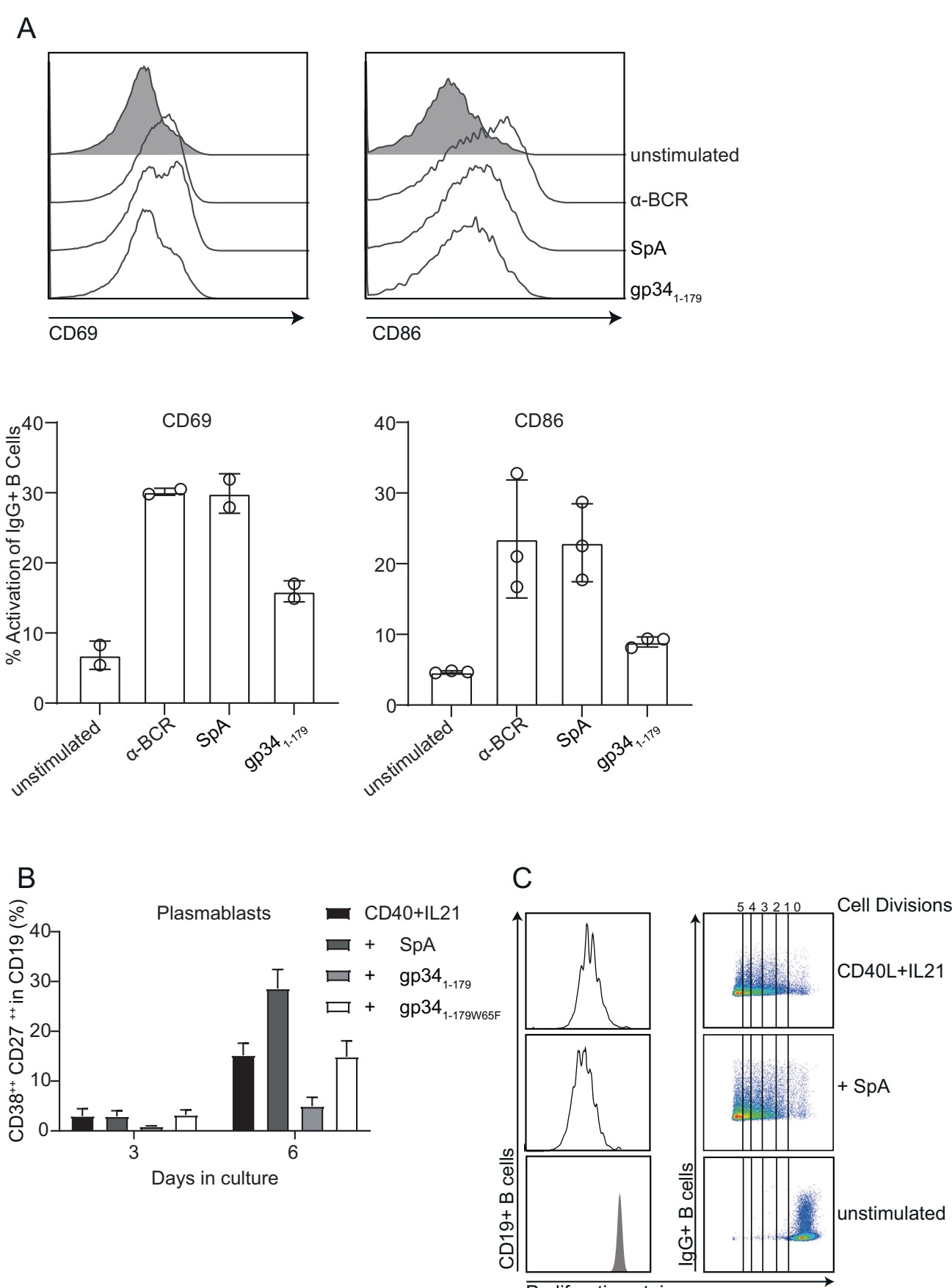

◄ **Figure EV5.** *Staphylococcus aureus* **protein A (SpA) activates IgG + B cells and supports proliferation and plasmablast formation.**

(A) Primary B cells were treated with 10 μg/mL each of α-BCR, SpA or gp34$_{1-179}$. Upregulation of activation markers CD69 and CD86 in IgG+ B cells were analyzed after 24 and 48 h respectively by flow cytometry. At least two biological replicates ($n = \geq$) were used in this experiment and error bars represent the standard deviation of the mean. (B) The CD19 + B cells were stimulated with 10 μg/mL of gp34$_{1-179}$, gp34$_{1-179W65F}$ or SpA in the presence of CD40L/IL21 for 3 and 6 days. The bars show the percentage of plasmablasts formed over the culture period. Data are represented as mean ± SEM of two independent experiments, each in duplicates. (C) B cells pre-stained with cell trace violet were treated with 10 ug/mL of SpA together with CD40L/IL-21, CD40L/IL-21 alone, or left unstimulated. Proliferation of the cells was examined after 6 days by flow cytometry.

