## [Peer Review File · EMBO Molecular Medicine]

A viral glycoprotein targets IgG memory B cells to mediate humoral immune evasion

Precious Cramer, Stefan Neys, Manuela Fiedler, Raquel Lorenzetti, Henrike Reinhard, Iga Janowska, Julian Staniek, Ann-Kathrin Kohl, Petra Hadlova, Magdalena Huber, Bodo Plachter, Clarissa Read, Valeria Falcone, Jens von Einem, Katja Hoffmann, Tihana Lenac Rovis, Stipan Jonjic, Philipp Kolb, Marta Rizzi, and Hartmut Hengel

Corresponding author(s): Hartmut Hengel (Hartmut.hengel@uniklinik-freiburg.de) , Marta Rizzi (marta.rizzi@uniklinik-freiburg.de)

Review Timeline:

Submission Date:	26th Aug 24
Editorial Decision:	19th Sep 24
Revision Received:	15th Oct 25
Editorial Decision:	24th Nov 25
Revision Received:	23rd Dec 25
Accepted:	23rd Dec 25

Editor: Zeljko Durdevic

Transaction Report:

19th Sep 2024

Dear Dr. Hengel,

Thank you for the submission of your manuscript to EMBO Molecular Medicine. We have now received feedback from the three reviewers who agreed to evaluate your manuscript. As you will see from the reports pasted below, all three referees recognize potential interest of the study but also raise important concerns that should be addressed in a major revision. If you would like to discuss further the points raised by the referees, I am available to do so via email or video. Let me know if you are interested in this option.

We would welcome the submission of a revised version within three months for further consideration. Please let us know if you require longer to complete the revision.

I look forward to receiving your revised manuscript.

Yours sincerely,

Zeljko Durdevic

We require:

- 1) A .docx formatted version of the manuscript text (including legends for main figures, EV figures and tables). Please make sure that the changes are highlighted to be clearly visible.
- 2) Individual production quality figure files as .eps, .tif, .jpg (one file per figure). For guidance, download the 'Figure Guide PDF': (<https://www.embopress.org/page/journal/17574684/authorguide#figureformat>).
- 3) A .docx formatted letter INCLUDING the reviewers' reports and your detailed point-by-point responses to their comments. As part of the EMBO Press transparent editorial process, the point-by-point response is part of the Review Process File (RPF), which will be published alongside your paper.
- 4) A complete author checklist, which you can download from our author guidelines (<https://www.embopress.org/page/journal/17574684/authorguide#submissionofrevisions>). Please insert information in the checklist that is also reflected in the manuscript. The completed author checklist will also be part of the RPF.
- 5) Please note that all corresponding authors are required to supply an ORCID ID for their name upon submission of a revised manuscript.
- 6) It is mandatory to include a 'Data Availability' section after the Materials and Methods. Before submitting your revision, primary

datasets produced in this study need to be deposited in an appropriate public database, and the accession numbers and database listed under 'Data Availability'. Please remember to provide a reviewer password if the datasets are not yet public (see <https://www.embopress.org/page/journal/17574684/authorguide#dataavailability>).

12) Author contributions: You will be asked to provide CRediT (Contributor Role Taxonomy) terms in the submission system. These replace a narrative author contribution section in the manuscript.

13) A Conflict of Interest statement should be provided in the main text.

14) Every published paper now includes a 'Synopsis' to further enhance discoverability. Synopses are displayed on the journal webpage and are freely accessible to all readers. They include a short stand first (maximum of 300 characters, including space) as well as 2-5 one-sentences bullet points that summarizes the paper. Please write the bullet points to summarize the key NEW findings. They should be designed to be complementary to the abstract - i.e. not repeat the same text. We encourage inclusion

of key acronyms and quantitative information (maximum of 30 words / bullet point). Please use the passive voice. Please attach these in a separate file or send them by email, we will incorporate them accordingly.

15) Include a Reagents and Tools Table as part of the Methods section, which can be downloaded from our author guidelines (<https://www.embopress.org/page/journal/17574684/authorguide#structuredmethods>)

**** Reviewer's comments ****

Referee #1 (Comments on Novelty/Model System for Author):

See also remarks to the author below.

High degree of novelty, as to the best of my knowledge binding of viral proteins to a specific Ig subclass and how this affects signaling has not been described to date. This has implications for our understanding of host defence against viruses in general.

Major: technical quality: comparison of gp34 and anti-IgG stimulation need improvement. Also anti-TNF α antibodies should show the relevance of TNF α for the molecular mechanism

Minor: possibly incorrect use of parametric statistical test.

Referee #1 (Remarks for Author):

This is an interesting manuscript in which the authors provide evidence that the HCMV viral protein gp34 has the capacity to bind IgG $^+$ - but not IgM $^+$ or IgA $^+$ - memory B cells. As a result, it induces BCR internalization and an aberrant, transient B cell proliferation that, surprisingly, appears to be independent of SYK which is a central tyrosine kinase in BCR signaling. In vitro plasmablast formation and immunoglobulin production is limited. Because gp34 triggers robust TNF α secretion, the authors conclude that this pro-inflammatory cytokine induces anergy in B cells, including IgA $^+$ and IgM $^+$ B cells.

This work is significant in that it uncovers a novel process by which viruses may impair B cell differentiation and function. Their findings definitely advance the field, whereby the authors almost reached the point of identifying a possible mechanism. They show that the gp34-IgG interaction induces TNF α .

Major:

- Page 7, line 216. It is intrinsically difficult to compare responsiveness (by quantifying the level of phosphorylation of signaling proteins) between stimuli, i.e. between gp34 and anti-IgG. It seems reasonable to assume that gp34 binds only to the C region of IgG, because the isotype-specific binding of gp34 excludes its binding to the V region) of IgG. It cannot be excluded that a different conclusion was reached if a different anti-IgG antibody was used. In fact, the antibody used is polyclonal anti-Ig(H+L). The additional binding to the Ig L chain might be the reason that stronger phosphorylation levels are reached for Lyn. Did the authors also test different anti-IgG clones (recognition the C-region only), or anti-total Ig? Likewise, what about the concentrations of anti-IgG and gp34 used; where they optimized? This would also be of relevance for other parts of the manuscript, e.g. Figure 3.

- Although it is conceivable that TNF α is a critical factor in the resulting aberrant B cell activation, in fact the authors only show a correlation. Blocking experiments with anti-TNF α antibodies (Figure 5, panels 1-C) is required to demonstrate the crucial role of TNF α production in this process.

- Their findings indicate that the ineffective B cell activation and differentiation is not due to apoptosis induction, but it remains unknown how B cells respond to gp34 activation: is proliferation hampered, or is there a specific arrest in differentiation.

Minor:

- Page 7, line 223. Regarding "... was confirmed by the absence of phosphorylated PLC γ 2" This should be tuned down. PLC γ has several phosphorylation sites, and only pY759 was analyzed.

- Page 7, line 233. Regarding "...resulted in lower reduction of CD69, MHCII and CD86". This should be rephrased, because MHCII was not significantly different between anti-IgG and gp34.

- Page 9, line 272-278 (Figure 4A). I'm not so convinced that gp34 stimulation results in lower pAKT(T308) and higher pAKT(473), compared with anti-IgG stimulation. In the histograms shown, the differences seem very small and the numbers of replicates/samples are small (n=3). Significance was evaluated by one-way ANOVA, which is a parametric test that assumes that the data is normally distributed. How can this be assumed on the basis of only 3 data points?

- Fig 7. According to the legend the figure shows that gp34 induces transient anergy in B cells. But, the figure shows B cell numbers and plasma cell differentiation. Classically, Ca-flux is studied to evaluate anergy.

Referee #2 (Comments on Novelty/Model System for Author):

This is a very impressive manuscript with high novel and interesting findings.

Referee #2 (Remarks for Author):

In this manuscript Cramer and colleagues show that human cytomegalovirus-encoded Fc γ -binding glycoprotein gp34 binds to IgG⁺ memory B cells and leading to induces internalization of the B cell receptor. This interaction results in impaired antibody production and also induction of transient anergy in B cells. This induction B cell anergy is massive secretion of pro-inflammatory cytokine TNF- α by IgG⁺ memory B cells. Mechanistically, gp34 interaction with IgG⁺ memory B cells initiates mTORC1 signalling responsible for protein synthesis and B cell proliferation. Overall this is a fascinating study and provides a novel insight on how HCMV evades B cell-mediated immune regulation. Authors have meticulously conducted detailed experiments to justify their conclusions. I have few minor comments which author may like to consider while revising their manuscript.

(a) Previous studies by Hidde Pleogh and colleagues have shown that gp34 binds HLA class I molecule and promotes egress of these molecules to the cell surface. This process protects virus-infected cells from NK cell mediated killing (The EMBO Journal (1997) 16: 685 - 694). Did authors observe any such interaction in IgG⁺ B cells?

(b) I was also wondering if the binding of gp34 to B cells can also lead to ADCC which may mediated by the antibodies directed to gp34. Do you detect gp34-specific antibodies in seropositive individuals.

(c) While authors have used both soluble gp34 and virions to demonstrate binding to B cells, can authors clarify if gp34 is released as a soluble protein from virus-infected cells. I understand author noticed that binding of virion leads to B cell death.

(d) Previous studies have shown that CMV-infected cells release dense bodies. I was wondering if gp34 is present in these dense bodies which may be one of the possible mechanism HCMV modulates the B cell function.

(e) It would be interesting if authors can isolate gB-specific memory B cells and demonstrate directly that binding of gp34 to these B cells impairs their ability to impaired gB-specific antibody production. Isolation of gB-specific B cells may be technically challenging. I would suggest authors use multimeric form of gB protein (eg. Tetrameric gB using streptavidin) to enrich GB-specific B cells. It would be nice if these gB-specific B cells can be matured as antibody-secreting plasma cells in the presence or absence of gp34.

Referee #3 (Comments on Novelty/Model System for Author):

For technical quality see comments below.

I think the medical impact could be elevated if the authors could demonstrate that gp34 is present in acute HCMV infection in serum.

Referee #3 (Remarks for Author):

Cramer and colleagues perform a series of convincing experiments to demonstrate that gp34, which is an HCMV encoded Fc γ -binding protein, binds to B cells and impairs their functionality. Finally leading to B-cell anergy and likely impaired immune response in vivo. They further characterize the mechanism and conclude that the process is Syk-independent, triggers mTORC1 signaling and TNF α . I have few criticisms to the data and experiments that were performed to a high level of rigor and stringency (see comments below). Weakest point is the Syk-conclusion that should crucially be backed up by using inhibitors of this signaling cascade/pathway. On top, total protein level controls need to be included. Apart from that, the authors should be careful not to overinterpret the whole virion incorporated gp34 story. At some points in the discussion, they also mention free circulating gp34. However, in the (graphical) abstract and throughout they mainly focus on gp34 as a virion associated factor even though nearly all of the experiments were done with soluble gp34. A concept they are missing at all is the potential incorporation of gp34 in extracellular vesicles. Do these forms have differential effects? Which one is the predominant in

patients? What is the fraction of virion vs free vs exosome associated gp34 in their preparation? Figure 1 totally fails to assess this and I think it should be minimum to clarify this aspect and tune down the conclusions and discussions accordingly.

Fig 1A. As the authors mention, it is already known that gp34 is a transmembrane protein, that was detected in the surface of HCMV infected cells (as shown in a previous publication of the lab, Corrales-Aguilar et al., 2014), therefore no novel information is provided by this graph. I would encourage the authors to point out the novelty of this graph, compared to previous published data. On top, it would be very relevant to discriminate between non-particle associated free released gp34 and particle bound. By the strategy used by the authors, they are not able to discriminate both forms; free as well as virion associated gp34 will be degraded by ProteinaseK; also both will stay upon SDS treatment. They should perform gradient centrifugation with SDS-treated vs non-treated samples to see if gp34 will float differentially upon SDS/detergent treatment; this fraction is the virion/vesicle associated form. Furthermore, could it be that gp34 is released non-virion associated in extracellular vesicles? This is highly relevant and could be assessed by co-staining of viral proteins and EV markers in the fractions.

Line120-122. The phrasing is a bit confusing since it indicates that membrane gp34 was present in the TKO variant and did not bind Fcy, while in fact, gp34 was absent in this condition.

Fig3B,C. In this graph the information about the total levels of proteins (Lyn, SYK, PLCy2) is missing, which cannot exclude the possibility that the different phosphorylation levels is due to differences in total expression. Perform also a Wblot here to allow to conclude on AB specificity. Flow cytometry signals might be compromised by non-specific ab binding.

Fig3D. It is not clear for which data sets the statistical analysis was performed and compared to what. Additionally, the lack of 3 replicates for gp34(1-179), 1hour sample might have an impact on the statistical analysis, and lead to misleading result either if it is included or omitted.

Generally, to strengthen the bold conclusion, that the process is Syk-independent I would suggest to repeat the experiment and include a Syk inhibitor.

Fig4B. This graph is confusing. Histograms and graph do not always agree. Unstimulated +wortmannin treated cells show higher pAKT in histograms compared to a-IgG +wortmannin, which is not the case in the graph. a-IgG -wortmannin cells show higher pAKT levels in the histograms compared to gp34(1-179) -wortmannin, while in the graphs the levels looks similar. I would advise the authors to double check for outliers in the raw plots that might affect the MFI values.

Fig4C,D. Similar to Fig3B,C, no total levels of S6 are shown.

Fig 6A. The calculation that the authors use for the % of apoptotic cells has the disadvantage of misleading results in the case when more cells are double negative, therefore healthy, in the Condition of Interest compared to the control. Specifically, in the a-IgG condition, a bigger % of the cells is double negative compared to the unstimulated, but the graph shows differently. Additionally, the y axis in Fig 6B is misleading since this is not the % of Apoptotic cells, but the per cent decrease of double negative cells in the Condition of Interest, compared to unstimulated control. I personally find this way of calculation rather confusing and unnecessary complicated.

Point by point response (EMM-2024-20477)

We are grateful to all reviewers for their very careful and thoughtful evaluation of our article and their helpful suggestions and comments.

The revision of the manuscript was accompanied by special circumstances and therefore took a long time. The first author Precious Cramer took up a postdoctoral position at the Center for Virology and Vaccine Research at the Beth Israel Deaconess Medical Center of Harvard Medical School in 2022, after which Precious was no longer available for further experiments. Fortunately, Prof. Rizzi was able to recruit Stefan Neys as a postdoc. Thanks to his pertinent experience with B cells, he was able to familiarise himself with the project and take over crucial parts of it. Stefan Neys is therefore now a joint first author of the work. We analysed the reviewers' criticisms and suggestions very thoroughly and put a great deal of effort and time into the experiments. We would like to thank the reviewers for their valuable feedback, which has significantly improved our work.

We respond to the issues raised by the reviewers in a point-by-point fashion. For orientation, we have responded to the reviewers' comments *in italic*.

Referee #1:**Referee #1 (Comments on Novelty/Model System for Author):**

High degree of novelty, as to the best of my knowledge binding of viral proteins to a specific Ig subclass and how this affects signaling has not been described to date. This has implications for our understanding of host defence against viruses in general.

Major: technical quality: comparison of gp34 and anti-IgG stimulation need improvement. Also anti-TNF α antibodies should show the relevance of TNF α for the molecular mechanism

Minor: possibly incorrect use of parametric statistical test.

Referee #1 (Remarks for Author):

This is an interesting manuscript in which the authors provide evidence that the HCMV viral protein gp34 has the capacity to bind IgG⁺ - but not IgM⁺ or IgA⁺ - memory B cells. As a result, it induces BCR internalization and an aberrant, transient B cell proliferation that, surprisingly, appears to be independent of SYK which is a central tyrosine kinase in BCR signaling. In vitro plasmablast formation and immunoglobulin production is limited. Because gp34 triggers robust TNF α secretion, the authors conclude that this pro-inflammatory cytokine induces anergy in B cells, including IgA⁺ and IgM⁺ B cells.

This work is significant in that it uncovers a novel process by which viruses may impair B cell differentiation and function. Their findings definitely advance the field, whereby the authors almost reached the point of identifying a possible mechanism. They show that the gp34-IgG interaction induces TNF α .

Author Reply: *We would like to thank the reviewer for their appreciation of the novelty of our study.*

Major:

Q1: Page 7, line 216. It is intrinsically difficult to compare responsiveness (by quantifying the level of phosphorylation of signaling proteins) between stimuli, i.e. between gp34 and anti-IgG. It seems reasonable to assume that gp34 binds only to the C region of IgG, because the isotype-specific binding of gp34 excludes its binding to the V region) of IgG. It cannot be excluded that a different conclusion was reached if a different anti-IgG antibody was used. In fact, the antibody used is polyclonal anti-Ig(H+L). The additional binding to the Ig L chain might be the reason that stronger phosphorylation levels are reached for Lyn. Did the authors also test different anti-IgG clones (recognition the C-region only), or anti-total Ig? Likewise, what about the concentrations of anti-IgG and gp34 used; where they optimized? This would also be of relevance for other parts of the manuscript, e.g. Figure 3.

Authors' response

AI: *We agree that binding of gp34₁₋₁₇₉ to the heavy chain of IgG is indeed different from that of our control stimulus, α -IgG (H+L). This has now been updated to α -BCR for clarification. While generally accepted for BCR stimulation, we agree that for the purposes of this study we needed to include further controls to compare BCR stimulation to gp34 stimulation on IgG⁺ B cells. To this end, we compared the stimulation of IgG⁺ B cells with anti-IgG (H+L) to anti-human Fc γ (anti-IgG (H)) and anti-human total Ig (anti-Ig). Similar levels of p-Lyn, p-SYK, p-PLC γ 2, p-AKT and p-S6 were observed with all three stimuli (see figure below). This indicates that the inability of B cells to induce p-SYK and p-PLC γ 2 following gp34₁₋₁₇₉ stimulation, compared with anti-IgG (H+L) stimulation, is irrespective of differences in binding site to mIgG. We have now incorporated these findings in our results section (see manuscript, lines 212–221).*

Figure: Stimulation of IgG⁺ B cells through anti-BCR (anti-IgG (H+L)), total Ig (anti-Ig), or heavy chain only (anti-IgG (H)). Phosphorylation was measured after 10 minutes (p-Lyn, p-SYK, and p-PLCγ2) or 1 hr (p-AKT and p-S6) post incubation with the indicated stimuli. Data were analyzed by a Kruskal-Wallis test.

With regards to the concentration of stimuli used, we selected 10 ug/mL as the ideal concentration of gp34₁₋₁₇₉ at which binding the BCR was not over-saturated (see figure below). A concentration of 10 ug/mL of α-BCR was used to allow for comparison with gp34₁₋₁₇₉ (see figure below). This figure is not included in the main text however the findings can be found in the materials and methods section (lines 627-628)

Figure: The optimal concentration of gp34 suitable for binding to IgG+ BCR without was determined by coating an ELISA plate with an α -IgG antibody. B cells were isolated from PBMCs and incubated with the coated plate overnight. Serial dilutions of the histidine-tagged purified recombinant proteins were then added to the plates. Peroxidase conjugated anti-His antibody was used to detect bound recombinant proteins to the B cells.

Q2: Although it is conceivable that TNF α is a critical factor in the resulting aberrant B cell activation, in fact the authors only show a correlation. Blocking experiments with anti-TNF α antibodies (Figure 5, panels 1-C) is required to demonstrate the crucial role of TNF α production in this process.

Authors' response

A2: Indeed, based on our data we hypothesized a correlation between increased TNF- α levels in the culture and the observed effects of gp34₁₋₁₇₉ on B cell proliferation and differentiation. Experiments neutralizing TNF- α in these cultures were a logical next step. We first used blocking antibodies targeting TNF- α (e.g. Adalimumab, Infliximab, Etanercept, and Golimumab) that have been proven effective and are commonly used in in vivo and in vitro studies. Unfortunately, these are all humanized antibodies containing an Fc γ region. As a result, gp34₁₋₁₇₉ binds to these monoclonal antibodies in our culture and provided inconclusive results.

Afelimomab is a more recently designed humanized F(ab')₂ anti-TNF- α , therefore lacking the Fc γ region. However, to the best of our knowledge, there is no existing literature on the effectiveness of this monoclonal antibody in vitro. Blocking experiments with Afelimomab were performed but remained inconclusive.

To avoid scavenging of gp34₁₋₁₇₉ from the anti-TNF monoclonal antibodies, we produced Infliximab F(ab')₂ fragments using the Pierce F(ab')₂ preparation kit (44988, ThermoFisher). Human F(ab')₂ fragments (009-0104, Rockland) were used as control. In this setting, TNF- α neutralization did not change the gp34₁₋₁₇₉-induced reduction in B cell proliferation and differentiation (see figure).

Figure: Total B cells were cultured with the indicated stimuli in the presence of 1 μ g/ml Infliximab or control F(ab')₂ fragments. Total B cell counts, and plasmablast counts and frequencies were assessed by flow cytometry after 3 days.

In addition, we tested the outcome of artificially increasing TNF- α levels in these cultures. The addition of recombinant TNF- α to the B cell cultures did not show a reduction in B cell proliferation nor plasma blast differentiation after 3, 6, or 9 days (see figure below). Thus, the current data do not allow us to provide a causal link between gp341-179-induced TNF- α secretion and the block in proliferation and plasma blast differentiation in our culture system.

Figure: Total B cells were cultured with 0, 20, or 100 ng/mL recombinant TNF- α in the presence of CD40L and IL-21. Total B cell counts, plasmablast counts, and plasmablast frequencies were assessed by flow cytometry at the indicated time points.

We have therefore adjusted these implications in the abstract (indicated in yellow) and in the graphical abstract. In the results and discussion, the description of these findings have been adjusted accordingly (see lines 282-288 in the results section, and onwards in the discussion (lines 417-422)).

Q3: Their findings indicate that the ineffective B cell activation and differentiation is not due to apoptosis induction, but it remains unknown how B cells respond to gp34 activation: is proliferation hampered, or is there a specific arrest in differentiation.

Authors' response

A3: The data shown in Fig. 5A, 5D, 7B, and S4C provide evidence that in the context of T-dependent B cell stimulation, gp341-179 hampers B cell proliferation. At the same time, the data presented in Fig. 5B, 5C, 7C, S4B, show that gp341-179 impairs B cell differentiation with no significant cell death. In these cultures, we observed persistent upregulation of CD69 on activated B cells in the presence of gp341-179 for up to 9 days (Fig 7A, and also below). CD69 is an early activation marker that normally wanes within hours of BCR engagement. CD69 promotes lymphocyte retention in tissues and inhibits egress into circulation, and is involved in B cell metabolism [Germain et al., 2021; Sic et al., 2014]. Hence, by prolonging CD69 expression, gp341-179 may contribute to B cell retention in secondary lymphoid organs. All together, our data show that gp341-179 modulates B cell function on multiple levels, by i) impairing proliferation of all B cells, ii) arresting plasmablast differentiation and iii) prolonging expression of CD69, which could indicate retaining B cells in secondary lymphoid organs, thereby contributing to

lymphadenomegaly typical of CMV infection. We introduced these additional data in Fig. 7A, and commented on them in lines 298-301.

We are therefore of the opinion that we clearly describe these blocking effects of gp34₁₋₁₇₉ on both B cell proliferation and differentiation.

Figure: The expression of CD69 on total B cells cultured with the indicated stimuli for 3, 6, or 9 days was assessed by flow cytometry.

Minor:

Q: Page 7, line 223. Regarding "... was confirmed by the absence of phosphorylated PLCγ2" This should be tuned down. PLCγ has several phosphorylation sites, and only pY759 was analyzed.

Authors' response

A: PLCγ2 indeed carries multiple phosphorylation sites. In this work, we looked at phosphorylation site Y759. This tyrosine residue is phosphorylated by SYK and serves as an important site for the enzymatic activity of PLCγ2. We now explain these signaling events downstream of the BCR to a greater extent (lines 189-193).

Q: Page 7, line 233. Regarding "...resulted in lower reduction of CD69, MHCII and CD86". This should be rephrased, because MHCII was not significantly different between anti-IgG and gp34.

Authors' response

A: We agree with the reviewer that we could not prove statistically significant differences between these different stimuli. We have now rephrased our observations regarding MHC-II expression in lines 206-207.

Q: Page 9, line 272-278 (Figure 4A). I'm not so convinced that gp34 stimulation results in lower pAKT(T308) and higher pAKT(473), compared with anti-IgG stimulation. In the histograms

shown, the differences seem very small and the numbers of replicates/samples are small (n=3). Significance was evaluated by one-way ANOVA, which is a parametric test that assumes that the data is normally distributed. How can this be assumed on the basis of only 3 data points?

A: We agree with the reviewer about the wrongful use of the parametric one-way ANOVA test for assessing differences between gp34₁₋₁₇₉ and α -BCR with only 3 data points. These data have been reanalyzed using the Kruskal-Wallis test and indicate no significant differences between gp34₁₋₁₇₉ and α -BCR stimulation for p-AKT (T308) and p-AKT(473). We have addressed this issue by alterations in the figure (Fig. 4A) and the description in the results (lines 230-234) accordingly.

Q: Fig 7. According to the legend the figure shows that gp34 induces transient anergy in B cells. But, the figure shows B cell numbers and plasma cell differentiation. Classically, Ca-flux is studied to evaluate anergy.

Authors' response

A: Our data show that gp34₁₋₁₇₉ induces an impaired induction of B cell proliferation, plasma blast differentiation, and the secretion of immunoglobulins in response to T-dependent stimuli. These effects on the B cell response indeed do not describe a classically defined state of B cell anergy, which is indicated by a lack in BCR signaling responses.

To this end, we cultured B cells for 3 days in the presence of CD40/IL-21 only, or in CD40/IL-21 with the addition of gp34₁₋₁₇₉ or gp34_{1-179W65F} to explore the potential induction of anergy. After 3 days, B cells were stimulated by α -BCR for 10 minutes and PLC γ 2 phosphorylation was measured by flow cytometry. We found that p-PLC γ 2 was significantly decreased in IgG⁺ B cells from gp34₁₋₁₇₉-treated cultures after α -BCR stimulation compared to controls (see figure below), indicating reduced BCR signaling responses. This suggests that gp34₁₋₁₇₉ affects the responsiveness of the BCR but does not induce a classical anergic response in IgG⁺ B cells. It is important to note that in this setting, however, already bound gp34₁₋₁₇₉ could also interfere with the α -BCR stimulation (see Fig. 3C) instead of an intrinsically reduced response. These observations therefore provide inconclusive results and were therefore not included in the manuscript.

We agree with the reviewer that the term “anergy” in this context is misleading. Would like to refer to the abstract and lines 305, 310, 324, 405 where we have adjusted our wording. We now reformulated that these B cells become hyporesponsive towards T-dependent stimuli in the presence of gp34₁₋₁₇₉.

Figure. Total B cells were incubated with the indicated stimuli. After 3 days, B cells were stimulated with anti-BCR for 10 minutes. Following fixation and permeabilization, the phosphorylation of PLCγ2 was determined by flow cytometry. Differences were analyzed by a Kruskal-Wallis test.

Referee #2:

Referee #2 (Comments on Novelty/Model System for Author):

This is a very impressive manuscript with high novel and interesting findings.

Referee #2 (Remarks for Author):

In this manuscript Cramer and colleagues show that human cytomegalovirus-encoded Fc γ -binding glycoprotein gp34 binds to IgG⁺ memory B cells and leading to induces internalization of the B cell receptor. This interaction results in impaired antibody production and also induction of transient anergy in B cells. This induction B cell anergy is massive secretion of pro-inflammatory cytokine TNF- α by IgG⁺ memory B cells. Mechanistically, gp34 interaction with IgG⁺ memory B cells initiates mTORC1 signalling responsible for protein synthesis and B cell proliferation. Overall this is a fascinating study and provides a novel insight on how HCMV evades B cell-mediated immune regulation. Authors have meticulously conducted detailed experiments to justify their conclusions. I have few minor comments which author may like to consider while revising their manuscript.

Author response:

We thank the reviewer for his/her careful reading of our manuscript and providing us with excellent feedback to improve our manuscript. Please find our point-by-point reply below.

Q: (a) Previous studies by Hidde Pleogh and colleagues have shown that gp34 binds HLA class I molecule and promotes egress of these molecules to the cell surface. This process protects virus-infected cells from NK cell mediated killing (The EMBO Journal (1997) 16: 685 - 694). Did authors observe any such interaction in IgG⁺ B cells?

(b) I was also wondering if the binding of gp34 to B cells can also lead to ADCC which may be mediated by the antibodies directed to gp34. Do you detect gp34-specific antibodies in seropositive individuals?

Authors' response:

A: The mouse cytomegalovirus-encoded m04/gp34 glycoprotein binds to MHC class I/ β 2m complexes in the ER and escorts them in conjunction with another MCMV protein, MATp1, to the cell surface (Kleinen et al. EMBO J 1997) to encounter inhibitory Ly49 NK cell receptors that compensate for the MHC I downregulation strategy of the virus (missing-self) resulting in NK cell resistance (Železnjak et al., Journal of Experimental Medicine, 2019). The m04/gp34 protein which bears no sequence relatedness to RL11/gp34 protein of HCMV does not bind MHC class I.

Nevertheless, it shares NK attenuating effects by inhibiting antibody dependent cell-mediated cytotoxicity (ADCC) via CD16/FcγRIII (Corrales-Aguilar et al, PLoS Pathogens, 2014).

We have previously analysed human IgG preparations Octagam and Gamunex, which are pooled from thousands of healthy donors, and also included the CMV hyperimmunoglobulin preparation Cytotect® which is used to treat HCMV diseased patients (see figure below). We found that gp34-specific antibody levels in HCMV-immune individuals are extremely low compared to other HCMV surface proteins like gB (see figure below). Since gp34 competes with CD16 for binding (Kolb et al., Elife, 2021), it is very unlikely that gp34 bound to a mIgG would modulate the NK cell response to the B cell. For this reason, this was not studied in more detail and these data were not used in the manuscript.

Figure. 96-well ELISA MaxiSorp plates (Thermo Fisher Scientific) were coated with the indicated concentrations of (A) either purified recombinant HCMV Fcγ-binding protein sgp34 or sgp68 protein or recombinant HCMV glycoprotein B and (B) purified recombinant HCMV Fcγ-binding protein sgp34 or recombinant HCMV matrix phosphoprotein pp65 (Abcam, cat# ab43041) or recombinant glycoprotein B (HCMV Towne, Sino Biological, produced in HEK293, cat# 10202-VCCH1). Fab₂ fragments were produced from Cytotect®, Octagam and Gamunex. Bound Fab₂ was detected by Fab-specific goat-anti human IgG Peroxidase antibodies (Sigma-Aldrich, cat# A029).

Q: (c) While authors have used both soluble gp34 and virions to demonstrate binding to B cells, can authors clarify if gp34 is released as a soluble protein from virus-infected cells. I understand author noticed that binding of virion leads to B cell death.

Authors' response

A: We agree this is a valid and vital point of the implications of our study. As pointed out in the manuscript (see line 208 to 211, Fig. EV5A) we exposed B cells to HCMV virions to study gp34-induced effects. HCMV virions could interact and activate B cells in a gp34-specific manner by upregulating CD69, CD86, and CD80 after at least 24 hours (see extended view 5A). Unfortunately, for long term B cell differentiation assays, HCMV virions (as HCMV-infected fibroblasts) were not suitable since they resulted in cell death before B cells could differentiate

(see extended view 5B, C)). We therefore used the reductionist experimental model based on the soluble gp34₁₋₁₇₉ variant. We agree with the reviewer that the shedding of gp34 from infected cells and gp34 detection in clinical situations is of prime interest. However, both aspects require extensive experimental effort and time for the investigations, and are therefore beyond the scope of this study. We would like to present an interim overview of the data we have collected so far, emphasising that these are preliminary findings which require further experimentation and are therefore not part of this paper.

Since several other glycoproteins from the RL11 gene family were demonstrated to be proteolytically cleaved (Engel et al., 2011; Pérez-Carmona et al., 2018; Windheim et al., 2013; Rubina et al., 2023), we tested gp34 shedding from transfected HeLa cells (see figure below). Indeed, gp34 was detected in the supernatant depending on the expression of the sheddase ADAM17 proving gp34 as an ADAM17 substrate, providing proof of concept that gp34 could be released from cells. However, shedding of gp34 was not readily detected in supernatants of HCMV-infected MRC-5 fibroblasts, possibly due to a limited expression and function of sheddases like ADAM17. Notably, we and others have recently shown that HCMV impairs ADAM17 surface expression through the action of two HCMV-encoded proteins in its UL/b' region, UL148 and UL148D (Rubina et al., 2023; Le-Trilling et al., 2023). Further experiments are therefore needed to determine the extent to which gp34 is released from infected cells.

Figure: HeLa cells were co-transfected using Superfect (Qiagen) with a pcDNA3-ADAM17-HA construct (Addgene #65105) and a non-functional ADAM-17-HA mutant (Addgene #65221) and variants of gp34. The supernatants were collected after 24 hrs and Fcγ binding ELISA was performed. ELISA plates were coated with 2μg/ml human Fc fragments before adding 100μl of supernatant. Detection of HA-epitope gp34 proteins was carried out with goat anti mouse HA-Bio (Mouse anti-HA-biotin from Sigma # B9183). The following constructs were used: RL11-HA (C terminus HA epitope tagged gp34), HA-RL11 (N terminus HA epitope tagged gp34), HA-solRL11 (N terminus HA tagged soluble gp34 comprising the gp34 ectodomain aa 1-179 as a control), and HA-RL11mtrp (N terminus HA epitope tagged Fc-binding deficient gp34₁₋₁₇₉W65F mutant).

Q: (d) Previous studies have shown that CMV-infected cells release dense bodies. I was wondering if gp34 is present in these dense bodies which may be one of the possible mechanism HCMV modulates the B cell function.

Authors' response

A: We would like to thank the reviewer for his/her reference to the presence of gp34 in dense bodies (DBs). DBs are subviral particles which are released in abundance from HCMV-infected fibroblast cultures and currently under development as a HCMV vaccine due to their immunogenicity in small animal models (e.g. Krauter et al., Vaccines 2022). This is why the incorporation of gp34 into DBs could be important for the vaccines' immunogenicity in humans. In collaboration with the laboratory of Prof. Bodo Plachter, Institute of Virology, University of Mainz, 3 preparations of DBs derived from AD169-derived recombinant viruses produced by Prof. Manfred Marschall, Institute of Clinical and Experimental Virology, Friedrich-Alexander University of Erlangen (König et al., Journal of General Virology 2017;98:2850–2863) were analysed for the presence of gp34 using tandem mass spectrometry (MS/MS) (see figure below). gp34-derived peptides were readily detected in purified DB preparations at similar levels as in virions. As no functional data was collected with DBs, we are presenting the results to the reviewers here, rather than including them formally in the publication. However, if the reviewers wish to include this MS/MS data, it can be incorporated. Prof. Bodo Plachter who generated the data is listed as a co-author of the paper.

Figure: Excerpt of HCMV components detected in Dense Bodies (n = 3 exp.). gp34 is indicated as IR11. ppm = parts per million. MS analysis was performed as described for HCMV virions in König et al. Journal of General Virology 2017;98:2850–2863.

Q: (e) It would be interesting if authors can isolate gB-specific memory B cells and demonstrate directly that binding of gp34 to these B cells impairs their ability to impaired gB-specific antibody production. Isolation of gB-specific B cells may be technically challenging. I would suggest authors use multimeric form of gB protein (eg. Tetrameric gB using streptavidin) to enrich GB-specific B cells. It would be nice if these gB-specific B cells can be matured as antibody-secreting plasma cells in the presence or absence of gp34.

Authors' response

A: We agree that the effect of gp34 should be further investigated in experiments involving antigen-specific restimulation of memory B cells (MBC) in the context of HCMV infection. In this regard, both T-dependent and T-independent approaches (Hebeis et al., J Exp Med., 2004) must be considered. Implementation of suitable settings for such in vitro experiments is particularly time-consuming and complex, so these experiments have to be reserved for a possible follow-up publication.

Referee #3

We highly appreciate the reviewer's time to critically assess our manuscript and provide us with feedback to improve our manuscript. Please find our point-by-point reply below.

Q: I think the medical impact could be elevated if the authors could demonstrate that gp34 is present in acute HCMV infection in serum.

Authors' response

A: We agree with the reviewer that detecting gp34 in serum is interesting and could provide a potential biomarker of active HCMV infection, although a direct interaction with IgG⁺MBC cannot be assumed under these conditions due to the abundance of IgG molecules. It is reasonable to assume that gp34 could be present in serum i) as part of complexes with IgG, ii) as a component of the HCMV virion, and iii) in exosomes produced by HCMV-infected cells.

To approach these different possibilities, we set up a gp34 detecting sandwich ELISA based on our monoclonal antibodies and tested human sera i) as supernatant after ultracentrifugation over a sucrose cushion to ensure that extracellular vesicle-associated or virion-bound gp34 remained in the pellet, and ii) after precipitation using polyethylenglycol (PEG, given the multimeric nature of gp34) as described by our group for the detection of soluble immune complexes (Chen et al., EMBO Mol Med. 2022 Jan 11;14(1):e14182; Ankerhold et al., Nature Communications, 2022).

The capture antibody, an anti-gp34 monoclonal antibody (mtrp04), was incubated with patient serum and test samples of HCMV-seronegative control individuals. Bound gp34 was detected using a biotinylated anti-gp34 antibody (see cartoon below). The test produced consistent measurement results with recombinant gp34-spiked serum samples and detected gp34 in serum at concentrations of ng/ml. Detection of gp34 in serum was obtained for some patients with HCMV reactivation, i.e. positive HCMV-DNA PCR test results in blood (see figure below). While most of the primary infected and reactivated HCMV patients did not seem to have soluble gp34, few individuals showed varying levels of gp34, with patient #1110149 showing the most soluble gp34 in the serum. There was no indication of gp34 in the serum of control patients in this test series. However, as some HCMV-seronegative controls did show positive test results following PEG precipitation, these results are still considered preliminary and require further experimentation and modification of the assay. In line with the precautionary principle of good scientific praxis, we refrain from making a definitive statement on the presence of soluble gp34 in serum in our study. We would like to remind the reviewers that it took decades to develop robust and reliable test systems for viral antigens such as HBsAg, HBeAg and HIVp24.

Hence, these gp34 tests are not sufficiently mature for diagnostic purposes and the data are too preliminary. We therefore did not go into more details in this manuscript.

Figure: Serum from HCMV+DNA+ (patients with reactivated HCMV infection), HCMV+DNA- (HCMV seropositive individuals), and HCMV seronegative controls was overlaid onto a sucrose gradient and ultracentrifuged at 100,000g for 1 hour at 4 degrees. The supernatants were collected and stored for use. An ELISA plate was coated with anti-gp34 antibody (mtrp04) for 2 hours and incubated with the ultracentrifuged serum overnight at 4 degrees. In each sera, gp34 was detected with anti-gp34 biotin antibody followed by streptavidin conjugated-HRP. The absorbance was measured and calculated as the measure of gp34 in each serum.

Q: Cramer and colleagues perform a series of convincing experiments to demonstrate that gp34, which is an HCMV encoded Fc-gamma binding protein, binds to B cells and impairs their functionality. Finally leading to B-cell anergy and likely impaired immune response in vivo. They further characterize the mechanism and conclude that the process is Syk-independent, triggers

mTORC1 signaling and TNF α . I have few criticisms to the data and experiments that were performed to a high level of rigor and stringency (see comments below). Weakest point is the Syk-conclusion that should crucially be backed up by using inhibitors of this signaling cascade/pathway. On top, total protein level controls need to be included. Apart from that, the authors should be careful not to overinterpret the whole virion incorporated gp34 story. At some points in the discussion, they also mention free circulating gp34. However, in the (graphical) abstract and throughout they mainly focus on gp34 as a virion associated factor even though nearly all of the experiments were done with soluble gp34. A concept they are missing at all is the potential incorporation of gp34 in extracellular vesicles. Do these forms have differential effects? Which one is the predominant in patients? What is the fraction of virion vs free vs exosome associated gp34 in their preparation? Figure 1 totally fails to assess this and I think it should be minimum to clarify this aspect and tune down the conclusions and discussions accordingly.

Authors' response

A: Previous studies have shown that after BCR stimulation, SYK is - partially - responsible for the activation of AKT through PI3K and BTK activation [Craxton et al. J. Biol. Chem. 1999; Li et al. Proc. Natl. Acad. Sci. 1999; Jiang et al. Blood, 2003]. In addition, the blocking effect of SYK inhibition (SYKi) on S6 phosphorylation has previously been reported (Rip et al. J Immun, 2020). SYKi would therefore lead to non-selective effects on the AKT signaling pathway.

Nonetheless, two different SYKi were tested. First generation fostamatinib (R406; IC₅₀: 41 nM), inhibited SYK phosphorylation, but also significantly reduced p-AKT and p-S6 following BCR stimulation. These effects may be a result of the off-target inhibitory effects on Lyn (IC₅₀: 63 nM) and Lck (IC₅₀: 37 nM). Therefore, second generation SYKi entospletinib (IC₅₀: 7.7 nM), described to lack any off-target affinity towards Lyn or Lck, was used in a subsequent experiment but provided similar results (see figure below).

The question as to whether a similar signaling phenotype could be established using SYKi could not be answered, and these findings were therefore not added to our manuscript.

Figure. Total B cells were pre-incubated with fostamatinib (R406), entospletinib, or DMSO control for 30 minutes. IgG+ B cells were stimulated via the BCR for 10 minutes (p-SYK) or 1 hour (p-AKT and p-S6) and phosphorylation of the indicated proteins was measured using flow cytometry. Data were analyzed by a Kruskal-Wallis test. * $p < 0.05$; ** $p < 0.01$

There is indeed increasing evidence on several divergent HCMV egress pathways which are thought to generate virion diversity including subviral particles (e.g. Flomm et al., PLoS Pathog. 2022 Aug 4;18(8):e1010575). We appreciate the special note of the reviewer and agree that gp34 is thus likely to be a component of CD63⁺CD81⁺ extracellular vesicles (EVs, exosomes) that were observed to be readily released from HCMV-infected cells in vitro (Walker et al., J Immunol 182:1548-1559, 2009; Zicari et al., Virology 524:97-105, 2018; Streck et al, J Virol 94:e00609-20, 2020) and demonstrated to contain HCMV proteins with late domain sequences such as gB and UL82/pp71. However, EVs have not yet been detected in HCMV-infected individuals, nor have they been separated from other HCMV particles (such as virions and potentially DBs) in patient material. We mention the possibility of gp34 incorporation in EVs now in the discussion in line 429 - 430 and refer to the relevant literature.

Q: Fig 1A. As the authors mention, it is already known that gp34 is a transmembrane protein, that was detected in the surface of HCMV infected cells (as shown in a previous publication of the lab, Corrales-Aguilar et al., 2014), therefore no novel information is provided by this graph. I would encourage the authors to point out the novelty of this graph, compared to previous published data. On top, it would be very relevant to discriminate between non-particle associated free released

gp34 and particle bound. By the strategy used by the authors, they are not able to discriminate both forms; free as well as virion associated gp34 will be degraded by ProteinaseK; also both will stay upon SDS treatment. They should perform gradient centrifugation with SDS-treated vs non-treated samples to see if gp34 will float differentially upon SDS/detergent treatment; this fraction is the virion/vesicle associated form., could it be that gp34 is released non-virion associated in extracellular vesicles? This is highly relevant and could be assessed by co-staining of viral proteins and EV markers in the fractions.

Authors' response

A: We thank the reviewer for his/her critical comment. As correctly noted, Fig. 1A reproduced data earlier published by us (Corrales-Aguilar et al., 2014) and has now been omitted from the manuscript. Fig 1B and Fig 1C aimed to independently demonstrate the presence of gp34 in the HCMV particle. During the time of revising the manuscript, Benteley et al. published a comprehensive MS/MS study of HCMV virion composition and found gp34 to be present in the virion, reproducing an earlier report of Reyda et al. J Virol. 2014 Sep 1;88(17):9633-46. Given this evidence from published literature, we refer now to this work in line 137 - 141. We have consequently removed this data from the figure. However, we would be in favor of keeping Figure 1B (now Fig. 1) because it is highly original using transmission electron microscopy and highly relevant as it demonstrates gp34 to be localized in the envelope of the virion. The complex TEM data was created in collaboration with colleagues from the University of Ulm who are co-authors of this paper.

Q: Line 120-122. The phrasing is a bit confusing since it indicates that membrane gp34 was present in the TKO variant and did not bind Fc γ , while in fact, gp34 was absent in this condition.

Authors' response

A: Fig 1A has now been removed from the original manuscript along with the corresponding lines.

Q: Fig3B,C. In this graph the information about the total levels of proteins (Lyn, SYK, PLC γ 2) is missing, which cannot exclude the possibility that the different phosphorylation levels is due to differences in total expression. Perform also a Wblot here to allow to conclude on AB specificity. Flow cytometry signals might be compromised by non-specific ab binding.

Authors' response

A: We optimized a protocol for and analyzed total protein levels of Lyn, SYK, PLC γ 2, AKT, and S6 in IgG⁺ B cells after stimulation with gp34₁₋₁₇₉ or gp34_{1-179W65F}. After incubation with either of the stimuli, we observed no significant changes in the expression levels of these proteins compared with unstimulated cells (see figure below). Previous studies also confirm these findings [Berry et al. Cell Reports. 2020; Awoniyi et al. J. Cell Sci. 2020]. We acknowledge the relevance of total protein levels when assessing protein phosphorylation, and have therefore included these findings

in supplementary figure 3B. We have also incorporated these findings in our results section (lines 218-219, and 250-252).

The antibodies used in this study to detect either total protein levels or phosphorylation of selected proteins have been extensively tested and validated for their specificity by the supplier (i.e. BD/Cell Signalling Technologies/Miltenyi). We therefore did not perform additional WB analyses for each of the different (phospho-)proteins analyzed in this study.

Figure: Total B cells were incubated with the indicated stimuli, or were left unstimulated. Following fixation and permeabilization, the expression of the indicated proteins was determined by flow cytometry. Data were analyzed by a Kruskal-Wallis test.

Q: Fig3D. It is not clear for which data sets the statistical analysis was performed and compared to what. Additionally, the lack of 3 replicates for gp34(1-179), 1 hour sample might have an impact on the statistical analysis, and lead to misleading result either if it is included or omitted.

Generally, to strengthen the bold conclusion, that the process is Syk-independent I would suggest to repeat the experiment and include a Syk inhibitor.

Authors' response

A: For better visualization, the presentation of Fig 3D has now been changed from a line graph to a column plot. Each bar represents the mean of three independent experiments from 3 technical replicates. The effects of each stimuli on BCR internalization with time was tested with a two-way ANOVA, followed by Tukey's post-hoc tests for pairwise comparisons within each timepoint.

Previous studies have shown that after BCR stimulation, SYK is - partially - responsible for the activation of AKT through PI3K and BTK activation [Craxton et al. J. Biol. Chem. 1999; Li et al. Proc. Natl. Acad. Sci. 1999; Jiang et al. Blood, 2003]. In addition, the blocking effect of SYK inhibition (SYKi) on S6 phosphorylation has previously been reported (Rip et al. J Immun, 2020). SYKi would therefore lead to non-selective effects on the AKT signaling pathway.

Nonetheless, two different SYKi were tested. First generation fostamatinib (R406; IC₅₀: 41 nM), inhibited SYK phosphorylation, but also significantly reduced p-AKT and p-S6 following BCR stimulation. These effects may be a result of the off-target inhibitory effects on Lyn (IC₅₀: 63 nM) and Lck (IC₅₀: 37 nM). Therefore, second generation SYKi entospletinib (IC₅₀: 7.7 nM), described to lack any off-target affinity towards Lyn or Lck, was used in a subsequent experiment but provided similar results (see figure below).

The question as to whether a similar signaling phenotype could be established using SYKi could not be answered, and these findings were therefore not added to our manuscript.

Figure. Total B cells were pre-incubated with fostamatinib (R406), entospletinib, or DMSO control for 30 minutes. IgG⁺ B cells were stimulated via the BCR for 10 minutes (p-SYK) or 1 hour (p-AKT and p-S6) and phosphorylation of the indicated proteins was measured using flow cytometry. Data were analyzed by a Kruskal-Wallis test. * $p < 0.05$; ** $p < 0.01$

Q: Fig4B. This graph is confusing. Histograms and graph do not always agree. Unstimulated +wortmannin treated cells show higher pAKT in histograms compared to a-IgG +wortmannin, which is not the case in the graph. a-IgG -wortmannin cells show higher pAKT levels in the histograms compared to gp34(1-179) -wortmannin, while in the graphs the levels looks similar. I would advise the authors to double check for outliers in the raw plots that might affect the MFI values.

Authors' response

A: *In this experiment, we compared the differences observed within each group rather than across groups. The focus here was to check the effect of Wortmannin in each group and this is captured in Fig 4B. In the presence of the inhibitor, p-AKT (S473) is reduced to a larger degree within gp34(1-179)-treated cells than in α -BCR treated cells. Another reason for such comparison is that in the presence of stimuli, Wortmannin affects the viability of the cells variably. It will therefore be unfair to make comparisons across groups. Differences in p-AKT (S473) levels with or without Wortmannin in unstimulated cells seem negligible in histogram and column graph, and no outliers were found in the raw data upon further inspection. We have therefore not made any alterations to the current figure.*

Q: Fig4C,D. Similar to Fig3B,C, no total levels of S6 are shown.

Authors' response

A: *We optimized a protocol for and analyzed total protein levels of Lyn, SYK, PLC γ 2, AKT, and S6 in IgG⁺ B cells after stimulation with gp34₁₋₁₇₉ or gp34₁₋₁₇₉W65F. After incubation with either of the stimuli, we observed no significant changes in the expression levels of these proteins compared with unstimulated cells (see figure below). Previous studies also confirm these findings [Berry et al. Cell Reports. 2020; Awoniyi et al. J. Cell Sci. 2020]. We acknowledge the relevance of total protein levels when assessing protein phosphorylation, and have therefore included these findings in supplementary figure 3B. We have also incorporated these findings in our results section (lines 218-219, and 250-252).*

The antibodies used in this study to detect either total protein levels or phosphorylation of selected proteins have been extensively tested and validated for their specificity by the supplier (i.e. BD/Cell Signalling Technologies/Miltenyi). We therefore did not perform additional WB analyses for each of the different (phospho-)proteins analyzed in this study.

Figure: Total B cells were incubated with the indicated stimuli, or were left unstimulated. Following fixation and permeabilization, the expression of the indicated proteins was determined by flow cytometry. Data were analyzed by a Kruskal-Wallis test.

Q: Fig 6A. The calculation that the authors use for the % of apoptotic cells has the disadvantage of misleading results in the case when more cells are double negative, therefore healthy, in the Condition of Interest compared to the control. Specifically, in the a-IgG condition, a bigger % of the cells is double negative compared to the unstimulated, but the graph shows differently. Additionally, the y axis in Fig 6B is misleading since this is not the % of Apoptotic cells, but the per cent decrease of double negative cells in the Condition of Interest, compared to unstimulated control. I personally find this way of calculation rather confusing and unnecessary complicated.

Authors' response

A: We have changed the depiction of the data in Fig. 6B to reflect the percentage of live B cells remaining after each treatment condition, instead of the ratio of total apoptotic cells. The figure has been altered accordingly.

24th Nov 2025

Dear Prof. Hengel,

Thank you for the submission of your revised manuscript to EMBO Molecular Medicine and please accept my apologies for the delay in getting back to you, which is due to the fact that one referee needed more time to complete his/her review. We have now heard back from the two referees who agreed to re-evaluate your manuscript. As you will see from the reports below, while the referee #3 is supporting publication of the manuscript, the referee #1 acknowledges the improvements of the revised manuscript but also raises important concerns that should be addressed in additional and final round of revision. No additional experiments are required. Please address the referee #1 concerns by toning down some conclusions, revising the statistical evaluations, providing additional clarifications and clearly stating the limitations of the study.

Acceptance or rejection of the manuscript will depend on the completeness of your responses included in the next, final version of the manuscript. For this reason, and to save you from any frustrations in the end, I would strongly advise against returning an incomplete revision.

In addition, please amend the following:

1) Figures: During a standard image analysis, we noted that:

- FACS graph in Appendix Fig. 4B, 3rd row Day 6 and 9 are identical. Please review and clarify.
- In Figure EV4B Western blots images appear over-contrasted. Please replace them with less processed images and provide source data for this Western blots.

2) Authors: E-mail correspondence to Henrike Reinhard and Iga Janowska could not be delivered. Please update their e-mail addresses and make sure to enter correct e-mail addresses for all authors in our submission system.

3) In the main manuscript file, please do the following:

- Please address all comments suggested by our data editors listed below:

o Figure legends:

1. Please note that the figure EV2-EV5 is mislabeled as figure EV3-EV6 in the manuscript. This needs to be rectified.
2. Please define the annotated p values ****/***/**/* as well as provide the exact p-values for the same in the legend of figure 3B, C; EV3 D, EV4A, B as appropriate.
3. Please note that the exact p values are not provided in the legends of figures 3D, E; 4A, D; 5A, B, C; 7B, C; EV3 A.
4. Please indicate the statistical test used for data analysis in the legends of figures EV4 A, B.
5. Please note that information related to n is missing in the legends of figures 2C, 4B, 5C, EV4 A-C; S3 A, B; S5, S6.
6. Please note that n=2 in figures 3E, 6B, D; 7A, EV1 E, EV5A, B; S1 A.
7. Please note that the error bars are not defined in the legends of figures 2A, C; 3B, E; 4A, 5C, EV3 A, D; EV4 A-C; S3 A, B; S5, S6.
8. Please note that the measure of center for the error bars needs to be defined in the legends of figures 3C, 4D.

- Limit keywords to max. 5.

- Remove information about BioRender from the figure legends and add a dedicated "Graphics" section in the Methods, following this format:

Graphics:

(some of the... OR Figure #... OR synopsis) Graphics were created with BioRender.com.

- In Methods, provide the antibody dilutions that were used for each antibody.
- In Methods, provide the statement that in addition to the WMA Declaration of Helsinki the experiments also conformed to the principles set out in the Department of Health and Human Services Belmont Report.
- Rename "Conflict of interest" to "Disclosure Statement & Competing Interests". We updated our journal's competing interests policy in January 2022 and request authors to consider both actual and perceived competing interests. Please review the policy <https://www.embopress.org/competing-interests> and update your competing interests if necessary.
- Author contributions: Please remove it from the manuscript and specify author contributions in our submission system. CRediT has replaced the traditional author contributions section because it offers a systematic machine-readable author contributions format that allows for more effective research assessment. You are encouraged to use the free text boxes beneath each contributing author's name to add specific details on the author's contribution. More information is available in our guide to authors:

<https://www.embopress.org/page/journal/17574684/authorguide#authorshipguidelines>

- Indicate in legends exact n and exact p values, not a range, along with the statistical test used. To keep the figures "clear" some authors found providing an Appendix table Sx with all exact p-values preferable. You are welcome to do this if you want to.
- Please provide Reagents and Tools Table and uploaded it as a separate file. Structured Methods section includes Reagents and Tools Table followed by a Methods and Protocols section. More information on how to adhere to this format as well as downloadable templates (.docx) for the Reagents and Tools Table can be found in our author guidelines:

<https://www.embopress.org/page/journal/17574684/authorguide#structuredmethods>

An example of a paper with Structured Methods can be found here:

<https://www.embopress.org/doi/full/10.1038/s44320-024-00037-6#sec-4>

- Rename "Data and materials availability" to "Data availability". Please use the following format to report the accession number

of your data:

[data type]: [full name of the resource] [accession number/identifier] ([doi or URL or identifiers.org/DATABASE:ACCESSION])

Please check "Author Guidelines" for more information.

<https://www.embopress.org/page/journal/17574684/authorguide#availabilityofpublishedmaterial>

- Please correct the reference citation in the reference list. Where there are more than 10 authors on a paper, 10 will be listed, followed by "et al.". Also, please remove DOIs. Please check "Author Guidelines" for more information.

<https://www.embopress.org/page/journal/17574684/authorguide#referencesformat>

- Update the order of the figure legends so that the main figure legends come first, followed by the EV figure legends under the heading "Expanded View Figure Legends"

4) Appendix: Please rename the file with the suppl. figures to "Appendix", remove the yellow font and upload it in PDF format. Please add a table of contents with page numbers and correct the nomenclature to "Appendix Figures S1" etc. throughout Appendix and in the main manuscript file.

5) Funding: Please make sure that information about all sources of funding are complete in both our submission system and in the manuscript. Currently, University of Rijeka under the project number uniri-biomed-18-23 is missing in the submission system.

6) Synopsis:

- Synopsis image: Please remove it from the manuscript and upload it as a separate, high-resolution jpeg file 550 px-wide x (300-600)-px high.

- Synopsis text: Please remove it from the man manuscript and provide it as a sperate .doc file in the following format: A short standfirst (maximum of 300 characters, including space) as well as 2-5 one sentence bullet points that summarise the paper. Please write the bullet points to summarise the key NEW findings. They should be designed to be complementary to the abstract - i.e. not repeat the same text. We encourage inclusion of key acronyms and quantitative information (maximum of 30 words / bullet point). Please use the passive voice.

7) As part of the EMBO Publications transparent editorial process (see our Editorial at

<http://embomolmed.embopress.org/content/2/9/329>), EMBO Molecular Medicine will publish online a Review Process File (RPF) to accompany accepted manuscripts. This file will be published in conjunction with your paper and will include the anonymous referee reports, your point-by-point response and all pertinent correspondence relating to the manuscript. Let us know whether you agree with the publication of the RPF and as here, if you want to remove or not any figures from it prior to publication. Please note that the Authors checklist will be published at the end of the RPF.

8) Please provide a point-by-point letter INCLUDING my comments as well as the reviewer's reports and your detailed responses (as Word file).

I look forward to reading a new revised version of your manuscript as soon as possible.

Yours sincerely,

Zeljko Durdevic

Zeljko Durdevic
Senior Editor
EMBO Molecular Medicine

*** Instructions to submit your revised manuscript ***

To submit your manuscript, please follow this link:

When preparing your revised manuscript, please refer to our guidelines: <https://link.springer.com/journal/44321/submission-guidelines#cms-Revised-submissions>. We perform an initial quality control of all revised manuscripts before re-review; failure to include requested items will delay the evaluation of your revision.

We require:

2) Individual production quality figure files as .eps, .tif, .jpg (one file per figure). For guidance, download the 'Figure Guide PDF': <https://media.springernature.com/original/springer-cms/rest/v1/content/27825798/data/v1>.

3) A .docx formatted letter INCLUDING the reviewers' reports and your detailed point-by-point responses to their comments. As part of the EMBO Press transparent editorial process, the point-by-point response is part of the Review Process File (RPF), which will be published alongside your paper.

4) A complete author checklist, which you can download from our author guidelines. Please insert information in the checklist that is also reflected in the manuscript. The completed author checklist will also be part of the RPF.

6) It is mandatory to include a 'Data Availability' section after the Materials and Methods. Before submitting your revision, primary datasets produced in this study need to be deposited in an appropriate public database, and the accession numbers and database listed under 'Data Availability'. Please remember to provide a reviewer password if the datasets are not yet public.

7) For data quantification: please specify the name of the statistical test used to generate error bars and P values, the number (n) of independent experiments (specify technical or biological replicates) underlying each data point and the test used to calculate p-values in each figure legend. The figure legends should contain a basic description of n, P and the test applied. Graphs must include a description of the bars and the error bars (s.d., s.e.m.).

9) Our journal encourages inclusion of *data citations in the reference list* to directly cite datasets that were re-used and obtained from public databases. Data citations in the article text are distinct from normal bibliographical citations and should directly link to the database records from which the data can be accessed. In the main text, data citations are formatted as follows: "Data ref: Smith et al, 2001" or "Data ref: NCBI Sequence Read Archive PRJNA342805, 2017". In the Reference list, data citations must be labeled with "[DATASET]". A data reference must provide the database name, accession number/identifiers and a resolvable link to the landing page from which the data can be accessed at the end of the reference.

12) Author contributions: You will be asked to provide CRediT (Contributor Role Taxonomy) terms in the submission system. These replace a narrative author contribution section in the manuscript.

13) A Disclosure Statement & Competing Interests statement should be provided in the main text.

14) Every published paper includes a 'Synopsis' to further enhance discoverability. Synopses are displayed on the journal webpage and are freely accessible to all readers. They include a short stand first (maximum of 300 characters, including space) as well as 2-5 one-sentences bullet points that summarizes the paper. Please write the bullet points to summarize the key NEW findings. They should be designed to be complementary to the abstract - i.e. not repeat the same text. We encourage inclusion of key acronyms and quantitative information (maximum of 30 words / bullet point). Please use the passive voice. Please attach these in a separate file or send them by email, we will incorporate them accordingly.

15) Include a Reagents and Tools Table as part of the Methods section, which can be downloaded from our author guidelines.

Photos 400-800 DPI

*Additional important information regarding figures and illustrations can be found at

<https://media.springernature.com/original/springer-cms/rest/v1/content/27825798/data/v1>

***** Reviewer's comments *****

Referee #1 (Comments on Novelty/Model System for Author):

The manuscript describes an interesting, important and novel mechanism of viral evasion from B cell responses. Technical quality of the experiments is high, but statistical evaluations need to be revised. Also some of the data are overinterpreted and the wording should be revised, although some additional experiments also need to be done.

Referee #1 (Remarks for Author):

In this manuscript Cramer et al. present a novel mechanism of viral evasion from B cell activation. They provide evidence that human cytomegalovirus gp34 binds to the IgG Fc domain and thereby induces SYK-independent signaling that subsequently induces a short-lived IgG+ B cell activation associated with TNF-alpha production, which results in generally impaired immunoglobulin production.

The topic of this manuscript is very interesting, particularly because it identifies a novel concept. It is generally clearly written, although some of the findings seem overinterpreted and have to be tuned down. For example, the findings on the apparent absence of SYK phosphorylation (Page 7, lines 217-127). Although the author wrote: 'Did not induce SYK phosphorylation'. It

cannot formally be excluded that SYK phosphorylation is induced, but rapidly lost., e.g. by robust phosphatase activity. Therefore, it would be essential to investigate shorter time points than 5 minutes, before this conclusion can be reached. The same goes for the effects of gp34 on anti-IgG stimulated B cells (Fig. 3D). The lack of SYK phosphorylation in gp24-stimulated B cells cannot be confirmed by the absence of phosphorylated PLC γ 2, as there may be many other explanations for this. If anything, it can be stated that it is in line with lack of SYK phosphorylation. The finding of increased DOK3 synthesis in B cells after 60 minutes of stimulation with gp34 (Fig. S5B) cannot explain why SYK phosphorylation is reduced at 5 minutes of stimulation. The authors conclude that delayed internalization was caused by reduced Lyn activation in gp34-treated cells. There is no evidence that would support this, only an association was found. It also cannot be excluded that the internalization is independent of SYK, as it is 'normalized' between 6 and 12 hrs, and it remains possible that SYK undergoes very late activation, e.g. by a feedback loop.

Figure 4. Phosphorylation status of pAKT (S473) upon gp34 stimulation does not seem reproducible: Compared to anti-IgG it is significantly upregulated (Fig. 4A), reduced (Fig. 4B, histograms on the left) or equal (Fig. 4B, right).

Page 11, lines 329-333. Their findings suggest that the impact of gp34-stimulated IgG+ B cells on non-IgG B cells is due to TNF-alpha production, which may suppress the activation / differentiation of neighboring B cells. The finding is only an association, experiments in which TNF-alpha is neutralized, e.g. with antibodies, are required to substantiate the mechanism.

Page 11, lines 336-338. The authors state that SpA induced plasmablast formation and cell proliferation of both IgG+ and non-IgG+ B cells, unlike gp34. However, figure S9B-C does not show non-IgG+ B cells.

Page 13. Lines 373-377. The authors aim to investigate "how long gp34 can exert such anergic features on B cells". To this end, they treated B cell cultures with gp34 in the presence of CD40L/IL-21 for 3 days and then neutralized gp34 with specific antibodies. This experiment does not answer the research question. Rather, it addresses reversibility of the gp34 effects. To investigate how long gp34 can exert its effect, the setup should be different, involving stimulation with CD40L/IL-21 at several time points after gp34 exposure and washout.

How do the authors define anergic B cells? The gp34-activated B cells do not show increased apoptosis susceptibility and induction of cell surface markers is only partially affected. Proliferative capacity should be checked.

MINOR COMMENTS:

- Figure 1: Description of the bottom part of panel C is missing.
- Page 11, line 312. The authors refer to B cell development, but this term is generally used for developing B cells in the bone marrow. B cell differentiation would be more correct.
- The authors use parametric one-way or two-way ANOVA and t-tests that assume that variables are approximately normally distributed in each group analyzed. How do they know this is the case. It is more appropriate to use non-parametric statistical tests.

Referee #3 (Remarks for Author):

the authors comprehensively addressed the concerns of the reviewers

A viral Fcγ-binding glycoprotein targets IgG memory B cells to mediate humoral immune evasion

Corresponding Authors: Professor Hartmut Hengel and Professor Marta Rizzi

Dear Prof. Hengel,

Thank you for the submission of your revised manuscript to EMBO Molecular Medicine and please accept my apologies for the delay in getting back to you, which is due to the fact that one referee needed more time to complete his/her review. We have now heard back from the two referees who agreed to re-evaluate your manuscript. As you will see from the reports below, while the referee #3 is supporting publication of the manuscript, the referee #1 acknowledges the improvements of the revised manuscript but also raises important concerns that should be addressed in additional and final round of revision. No additional experiments are required. Please address the referee #1 concerns by toning down some conclusions, revising the statistical evaluations, providing additional clarifications and clearly stating the limitations of the study.

Acceptance or rejection of the manuscript will depend on the completeness of your responses included in the next, final version of the manuscript. For this reason, and to save you from any frustrations in the end, I would strongly advise against returning an incomplete revision.

Point by point response (EMM-2024-20477-V2)

Der Dr Durdevic,

We would like to express our gratitude to both reviewers for re-evaluating the revised manuscript, and to you and the entire editorial team at EMBO Molecular Medicine for your valuable and detailed instructions on how to improve the work and meet the publication requirements.

You will find a point-by-point response to your comments below, followed by our responses to the two reviewers. Please note that all changes made in the V2 version of the ms are highlighted throughout in **green**, the previous changes are kept in **yellow**. We are confident that the improvements we have made will ensure our manuscript meets the publication requirements.

Kind regards on behalf of all the co-authors.

Hartmut Hengel and Marta Rizzi

1) Figures: During a standard image analysis, we noted that:
- FACS graph in Appendix Fig. 4B, 3rd row Day 6 and 9 are identical. Please review and clarify.
Thank you for pointing out this error. We have inserted the correct FACS blot.

- In Figure EV4B Western blots images appear over-contrasted. Please replace them with less processed images and provide source data for this Western blots.

Although the Western blots appear highly contrasted, they are authentic and have not been edited using Photoshop. We wish to keep the images and are making the source data available.

2) Authors: E-mail correspondence to Henrike Reinhard and Iga Janowska could not be delivered. Please update their e-mail addresses and make sure to enter correct e-mail addresses for all authors in our submission system.

Done

3) In the main manuscript file, please do the following:

- Please address all comments suggested by our data editors listed below:

o Figure legends:

1. Please note that the figure EV2-EV5 is mislabeled as figure EV3-EV6 in the manuscript. This needs to be rectified.

Done

2. Please define the annotated p values ****/***/**/* as well as provide the exact p-values for the same in the legend of figure 3B, C; EV3 D, EV4A, B as appropriate.

Done

3. Please note that the exact p values are not provided in the legends of figures 3D, E; 4A, D; 5A, B, C; 7B, C; EV3 A.

Done

4. Please indicate the statistical test used for data analysis in the legends of figures EV4 A, B.

Done

5. Please note that information related to n is missing in the legends of figures 2C, 4B, 5C, EV4 A-C; S3 A, B; S5, S6.

Done

6. Please note that n=2 in figures 3E, 6B, D; 7A, EV1 E, EV5A, B; S1 A.

See the relevant information in the captions accompanying the figures.

7. Please note that the error bars are not defined in the legends of figures 2A, C; 3B, E; 4A, 5C, EV3 A, D; EV4 A-C; S3 A, B; S5, S6.

This information has been provided.

8. Please note that the measure of center for the error bars needs to be defined in the legends of figures 3C, 4D.

Done

- Limit keywords to max. 5.

Done

- Remove information about BioRender from the figure legends and add a dedicated "Graphics" section in the Methods, following this format:

Graphics:

(some of the... OR Figure #... OR synopsis) Graphics were created with BioRender.com.

Done

- In Methods, provide the antibody dilutions that were used for each antibody.

We regularly re-evaluate the optimal antibody concentrations for each delivery from the manufacturer and for each detection method. Hence the specific antibody dilution is valid only for a given 'lot' of the antibody. Using our provided dilution, an untested antibody may hinder reproducibility of data. We therefore decided to not provide this information, to prompt proper antibody testing prior experiments.

- In Methods, provide the statement that in addition to the WMA Declaration of Helsinki the experiments also conformed to the principles set out in the Department of Health and Human Services Belmont Report.

Done

- Rename "Conflict of interest" to "Disclosure Statement & Competing Interests". We updated our journal's competing interests policy in January 2022 and request authors to consider both actual and perceived competing interests. Please review the policy <https://www.embopress.org/competing-interests> and update your competing interests if necessary.

Done

- Author contributions: Please remove it from the manuscript and specify author contributions in our submission system. CRediT has replaced the traditional author contributions section because it offers a systematic machine-readable author contributions format that allows for more effective research assessment. You are encouraged to use the free text boxes beneath each contributing author's name to add specific details on the author's contribution. More information is available in our guide to authors:

<https://www.embopress.org/page/journal/17574684/authorguide#authorshipguidelines>

Done

- Indicate in legends exact n and exact p values, not a range, along with the statistical test used. To keep the figures "clear" some authors found providing an Appendix table Sx with all exact p-values preferable. You are welcome to do this if you want to.

Done, see above

- Please provide Reagents and Tools Table and uploaded it as a separate file. Structured Methods section includes Reagents and Tools Table followed by a Methods and Protocols section. More information on how to adhere to this format as well as downloadable templates (.docx) for the Reagents and Tools Table can be found in our author guidelines:

<https://www.embopress.org/page/journal/17574684/authorguide#structuredmethods>

An example of a paper with Structured Methods can be found here:

<https://www.embopress.org/doi/full/10.1038/s44320-024-00037-6#sec-4>

Done

- Rename "Data and materials availability" to "Data availability". Please use the following format to report the accession number of your data:

[data type]: [full name of the resource] [accession number/identifier] ([doi or URL or identifiers.org/DATABASE:ACCESSION])

Done

Please check "Author Guidelines" for more information.

<https://www.embopress.org/page/journal/17574684/authorguide#availabilityofpublishedmaterial>

- Please correct the reference citation in the reference list. Where there are more than 10 authors on a paper, 10 will be listed, followed by "et al.". Also, please remove DOIs. Please check "Author Guidelines" for more information.

<https://www.embopress.org/page/journal/17574684/authorguide#referencesformat>

Done

- Update the order of the figure legends so that the main figure legends come first, followed by the EV figure legends under the heading "Expanded View Figure Legends"

Done

4) Appendix: Please rename the file with the suppl. figures to "Appendix", remove the yellow font and upload it in PDF format. Please add a table of contents with page numbers and

correct the nomenclature to "Appendix Figures S1" etc. throughout Appendix and in the main manuscript file.

Done

5) Funding: Please make sure that information about all sources of funding are complete in both our submission system and in the manuscript. Currently, University of Rijeka under the project number uniri-biomed-18-23 is missing in the submission system.

Done

6) Synopsis:

- Synopsis image: Please remove it from the manuscript and upload it as a separate, high-resolution jpeg file 550 px-wide x (300-600)-px high.

Done

- Synopsis text: Please remove it from the man manuscript and provide it as a sperate .doc file in the following format: A short standfirst (maximum of 300 characters, including space) as well as 2-5 one sentence bullet points that summarise the paper. Please write the bullet points to summarise the key NEW findings. They should be designed to be complementary to the abstract - i.e. not repeat the same text. We encourage inclusion of key acronyms and quantitative information (maximum of 30 words / bullet point). Please use the passive voice.

Done

- Please check your synopsis text and image before submission with your revised manuscript.

Done

Please be aware that in the proof stage minor corrections only are allowed (e.g., typos).

7) As part of the EMBO Publications transparent editorial process (see our Editorial at <http://embomolmed.embopress.org/content/2/9/329>), EMBO Molecular Medicine will publish online a Review Process File (RPF) to accompany accepted manuscripts. This file will be published in conjunction with your paper and will include the anonymous referee reports, your point-by-point response and all pertinent correspondence relating to the manuscript. Let us know whether you *agree* with the publication of the RPF and as here, if you want to remove or not any figures from it prior to publication. Please note that the Authors checklist will be published at the end of the RPF.

We agree with the publication of the RPF

8) Please provide a point-by-point letter INCLUDING my comments as well as the reviewer's reports and your detailed responses (as Word file).

Done

I look forward to reading a new revised version of your manuscript as soon as possible.

Yours sincerely,

Zeljko Durdevic

Zeljko Durdevic
Senior Editor
EMBO Molecular Medicine

***** Reviewer's comments *****

Referee #1 (Comments on Novelty/Model System for Author):

The manuscript describes an interesting, important and novel mechanism of viral evasion from B cell responses. Technical quality of the experiments is high, but statistical evaluations need to be revised. Also some of the data are overinterpreted and the wording should be revised, although some additional experiments also need to be done.

Author Reply: *We would like to thank the reviewer for his/her appreciation of the novelty of our study. The statistical validation of the statements in the experiments was completed in full (see above for details).*

Referee #1 (Remarks for Author):

In this manuscript Cramer et al. present a novel mechanism of viral evasion from B cell activation. They provide evidence that human cytomegalovirus gp34 binds to the IgG Fc domain and thereby induces SYK-independent signaling that subsequently induces a short-lived IgG+ B cell activation associated with TNF-alpha production, which results in generally impaired immunoglobulin production.

Author Reply: *We agree with the reviewer that our data demonstrate that gp34 triggers unconventional BCR signalling. As we could not demonstrate dependence on SYK, we interpret our data as indicating SYK-independent signalling. However, this does not rule out the possibility of SYK being involved in time dynamics and interactions that we did not assess.*

The topic of this manuscript is very interesting, particularly because it identifies a novel concept. It is generally clearly written, although some of the findings seem overinterpreted and have to be tuned down. For example, the findings on the apparent absence of SYK phosphorylation (Page 7, lines 217-127). Although the author wrote: 'Did not induce SYK phosphorylation'. It cannot formally be excluded that SYK phosphorylation is induced, but rapidly lost., e.g. by robust phosphatase activity. Therefore, it would be essential to investigate shorter time points than 5 minutes, before this conclusion can be reached. The same goes for the effects of gp34 on anti-IgG stimulated B cells (Fig. 3D).

Author Reply: *We agree with the reviewer that the absence of evidence for SYK phosphorylation does not formally prove absolute evidence for its complete absence. Indeed, we cannot excluded a short Syk phosphorylation at very short time points, below 5 minutes. In our discussion of the results, we emphasise the limitations of our study in light of this irrefutable principle of experimental research. Accordingly, we have added more details to the description of the results shown in Figure 3C on page 5, line 183, highlighted in green). We have also included a sentence in the discussion that specifically emphasises the limitations of our study (see page 7, lines 342 to 344, highlighted in green).*

The lack of SYK phosphorylation in gp24-stimulated B cells cannot be confirmed by the absence of phosphorylated PLCg2, as there may be many other explanations for this. If anything, it can be stated that it is in line with lack of SYK phosphorylation.

Author Reply: *We agree with the reviewer and have modified our statement one page 5, lines 187-189: "We found that in line with the absence of detectable p-SYK induction, gp341-179 stimulation was accompanied by unaltered PLCγ2 (Y579) phosphorylation levels (Fig 3B)."*

The finding of increased DOK3 synthesis in B cells after 60 minutes of stimulation with gp34 (Fig. S5B) cannot explain why SYK phosphorylation is reduced at 5 minutes of stimulation.

Author Reply: We agree with the reviewer and have modified our statement on page 5, lines 191 to 193: “We observed an increase in DOK3 synthesis in gp34₁₋₁₇₉ stimulated B cells after 1 hour (Fig EV3B). Prolonged binding of gp34₁₋₁₇₉ may thereby result in more frequent association of DOK3 with Lyn, reducing SYK activity over time.”

The authors conclude that delayed internalization was caused by reduced Lyn activation in gp34-treated cells. There is no evidence that would support this, only an association was found. It also cannot be excluded that the internalization is independent of SYK, as it is 'normalized' between 6 and 12 hrs, and it remains possible that SYK undergoes very late activation, e.g. by a feedback loop.

Author Reply: We gently disagree, since we have not concluded that delayed internalization was caused by reduced Lyn activation in gp34-treated cells. We agree that SYK could undergo a very late activation and have modified our statement on page 5, lines 199 – 201, now phrasing: “Therefore, a delayed BCR internalization in gp34₁₋₁₇₉-treated B cells could be the result of alterations or delays in the activation of different pathways downstream of the BCR”.

Figure 4. Phosphorylation status of pAKT (S473) upon gp34 stimulation does not seem reproducible: Compared to anti-IgG it is significantly upregulated (Fig. 4A), reduced (Fig. 4B, histograms on the left) or equal (Fig. 4B, right).

Author Reply: Statistical analysis of the phosphorylation status of pAKT (S473) following stimulation with gp34 and anti-BCR reveals no significant differences (see Figures 4A and 4B).

Page 11, lines 329-333. Their findings suggest that the impact of gp34-stimulated IgG+ B cells on non-IgG B cells is due to TNF-alpha production, which may suppress the activation / differentiation of neighboring B cells. The finding is only an association, experiments in which TNF-alpha is neutralized, e.g. with antibodies, are required so substantiate the mechanism.

Author Reply: We agree. The data that were provided already in Suppl. Fig.5 escaped the reviewer's attention.

Page 11, lines 336-338. The authors state that SpA induced plasmablast formation and cell proliferation of both IgG+ and non-IgG+ B cells, unlike gp34. However, figure S9B-C does not show non-IgG+ B cells.

Author Reply: On page 6, lines 269-270 we note what Figures EV5 A-C show, i.e. that SpA induces plasmablast formation and cell proliferation of B cells (CD19+).

Page 13. Lines 373-377. The authors aim to investigate "how long gp34 can exert such anergic features on B cells". To this end, they treated B cell cultures with gp34 in the presence of CD40L/IL-21 for 3 days and then neutralized gp34 with specific antibodies. This experiment does not answer the research question. Rather, it address reversibility of the gp34 effects. To investigate how long gp34 can exert its effect, the setup should be different, involving stimulation with CD40L/IL-21 at several time points after gp34 exposure and washout.

Author Reply: We did not claim to investigate “how long gp34 can exert such anergic features on B cells” but tested the reversibility of the gp34 effect as rightly stated by the reviewer.

How do the authors define anergic B cells? The gp34-activated B cells do not show increased apoptosis susceptibility and induction of cell surface markers is only partially affected. Proliferative capacity should be checked.

Author Reply: *In the initial version of the manuscript, we referred to an anergic state of the IgG+ B cells following to contact with gp34. However, this has been avoided in the revised version. It seems that the reviewer's question refers to the previous version.*

MINOR COMMENTS:

- Figure 1: Description of the bottom part of panel C is missing.

Author Reply: *It seems that the reviewer's question refers to the previous version. We describe the figure now as follows: The top four panels show virions from wildtype HB5 and the bottom 2 panels represent virions from triple knockout (TKO) HB5 virions. Scale bars indicate 100 nm.*

- Page 11, line 312. The authors refer to B cell development, but this term is generally used for developing B cells in the bone marrow. B cell differentiation would be more correct.

Author Reply: *We are grateful for the reviewer's hint. It seems that the reviewer's question refers to the previous version of the ms. We had corrected this already.*

- The authors use parametric one-way or two-way ANOVA and t-tests that assume that variables are approximately normally distributed in each group analyzed. How do they know this is the case. It is more appropriate to use non-parametric statistical tests.

Author Reply: *We are grateful for the reviewer's hint. For samples size less than three, no statistics was performed. The authors used Kruskal Wallis test for non-parametric data sets. This has been indicated in the figure legends.*

Referee #3 (Remarks for Author):

the authors comprehensively addressed the concerns of the reviewers

We would like to thank the reviewers for re-evaluating the revised manuscript and his positive evaluation

23rd Dec 2025

Dear Prof. Hengel,

We are pleased to inform you that your manuscript is accepted for publication and is now being sent to our publisher to be included in the next available issue of EMBO Molecular Medicine.

You may qualify for financial assistance for your publication charges - either via a Springer Nature fully open access agreement or an EMBO initiative. Check your eligibility: <https://link.springer.com/journal/44321/how-to-publish-with-us>

Zeljko Durdevic
Senior Editor
EMBO Molecular Medicine

>>> Please note that it is EMBO Molecular Medicine policy for the transcript of the editorial process (containing referee reports and your response letter) to be published as an online supplement to each paper. If you do NOT want this, you will need to inform the Editorial Office via email immediately. More information is available here: <https://link.springer.com/partners/embo-press/editorial-policies#Peer%20review>